# Mass Spectrometric Identification of Metabolites after Magnetic-Pulse Treatment of Infected *Pyrus communis* L. Microplants

**DOI:** 10.3390/ijms242316776

**Published:** 2023-11-26

**Authors:** Mikhail Upadyshev, Bojidarka Ivanova, Svetlana Motyleva

**Affiliations:** 1Laboratory of Virology, Russian State Agrarian University—Moscow Timiryazev Agricultural Academy, Timiryazevskaya Str. 49, 127422 Moscow, Russia; upad8@mail.ru; 2Independent Researcher, 44139 Dortmund, Germany; bojidarka.ivanova@yahoo.com; 3Federal State Budgetary Scientific Institution “Federal Scientific Center of Legumes and Groat Crops”, Molodezhnaya Str. 10, 302502 Oryol, Russia

**Keywords:** mass spectrometric metabolomics, chromatography, *Pyrus communis* L., structural analysis, magnetic-pulse irradiation

## Abstract

The major goal of this study is to create a venue for further work on the effect of pulsed magnetic fields on plant metabolism. It deals with metabolite synthesis in the aforementioned conditions in microplants of *Pyrus communis* L. So far, there have been glimpses into the governing factors of plant biochemistry in vivo, and low-frequency pulsed magnestatic fields have been shown to induce additional electric currents in plant tissues, thus perturbing the value of cell membrane potential and causing the biosynthesis of new metabolites. In this study, sixty-seven metabolites synthesized in microplants within 3–72 h after treatment were identified and annotated. In total, thirty-one metabolites were produced. Magnetic-pulse treatment caused an 8.75-fold increase in the concentration of chlorogenic acid (RT = 8.33 ± 0.01_97_ min) in tissues and the perturbation of phenolic composition. Aucubin, which has antiviral and antistress biological activity, was identified as well. This study sheds light on the effect of magnetic fields on the biochemistry of low-molecular-weight metabolites of pear plants in vitro, thus providing in-depth metabolite analysis under optimized synthetic conditions. This study utilized high-resolution gas chromatography-mass spectrometry, metabolomics methods, stochastic dynamics mass spectrometry, quantum chemistry, and chemometrics, respectively. Stochastic dynamics uses the relationships between measurands and molecular structures of silylated carbohydrates, showing virtually identical mass spectra and comparable chemometrics parameters.

## 1. Introduction

Emerging environmental problems and social factors such as degradation processes in ecosystems, climate change, migration and demographic inequality have attracted public concern recently. This has caused a focus on innovations in science and technology regarding methods for the biofortification of crop plants. Despite ongoing efforts and scientific aspects, technical innovations and political and normative tasks focus on the restoration of damaged soils. There are many advances in technologies for the enrichment of plants with nutrients. These appear to be effective and low-cost approaches to treat the deficiency of micronutrients in crop plants, thus introducing these plants effectively into food chains. These methods are recommendable for soils that show a deficiency of micronutrients. Marked differences among approaches to biofortify crop plants can be observed.

Nevertheless, there is little work on the low-cost, environmentally friendly magnetic-pulse treatment of plants, which is capable of inducing a broad spectrum of in vitro and in vivo reactions [1,2,3,4,5,6]. External magnetic-pulse fields perturb the membrane potential in cells. A stimulatory or inhibitory effect is thus exerted on plant growth biochemical processes [1,2,3,4,5,6].

Therefore, we need to ask the following question: How does a magnetic-pulse field affect metabolite processes in vivo? This has not been addressed comprehensively so far, although plants have adapted to develop in the magnetic field of our planet. We should begin by highlighting that there are insufficient data on magnetic field effects on the metabolism of garden plants. This fact prevents us from obtaining an in-depth understanding of biochemical mechanisms related to the biological activity of metabolites in magnetic-pulse treatment conditions. Plants also use this process to recover from viruses. 

Furthermore, many methods are practically implemented to achieve the aforementioned purposes, amongst others, including thermotherapy, meristem culture in vitro, and chemotherapy. The former approach seems to be ineffective against thermostable viruses, while chemotherapy inhibits growth processes up to plant death. 

Conversely, magneto-therapy in vitro appears to be a promising recovery approach when looking at plants that are free from disease. It shows many advantages, including the following: (i) a favorable effect on plant growth; (ii) application to relatively large explants; (iii) the automation of plant treatment process; (iv) environmental friendliness compared with the ecotoxicity of commercially applied antiviral agents; (v) the safety to the operator, and more [6,7]. As has been assumed when looking at the mechanistic aspects of magnetic fields’ effects on free-from-disease plants that have recovered from viruses, nonspecific resistance develops in plants via the in vivo synthesis of phenols and other metabolites that have antiviral activity. To make an argument for this claim, so far, there have been few metabolomics studies devoted to plant biochemistry under magnetic-pulse treatment [6,7]. What is puzzling is the diversity of the new molecular structures of naturally occurring phenols. They have capabilities related to oxidation, isomerization, interconversion, or interaction with proteins, carbohydrates (CBs), metal ions, and more, depending on environmental conditions, including magnetic-pulse treatment. Various arguments provide reasons for supporting this view (below).

However, an in-depth understanding of the polyfunctional biological activity of plant metabolites, their importance in plant life, the relationships among environmental factors, and the biochemistry of plant metabolites is necessary. It is determined that new biologically active compounds are formed under certain environmental conditions and factors, such as magnetic-pulse treatment. The biological activity of many phenols, including chlorogenic (CA), gallic and salicylic acids, in forming plant resistance to abiotic and biotic factors is highlighted. Many such metabolites are involved in plant defense reactions during pathogenesis. A large number of them exhibit fungicidal, bactericidal, insecticidal, nematicidal, and antiviral activity. So far, it has been shown that magnetic-pulse treatment leads to a change in the phenolic composition of raspberry microplants, increasing the CA content in tissues by 33% comparing with the control. The biochemical composition of raspberry microplants was drastically affected under magnetic-pulse treatment conditions [8]. However, scarce data exist on the effect of magnetic fields on the biosynthesis of low-molecular-weight metabolites by microplants of *Pyrus communis* L.

In delineating the state of the art of this issue, this study determines new plant metabolites in an external magnetic-pulse field by looking at *Pyrus communis* L. microplants. It utilizes metabolomics gas chromatography coupled with mass spectrometry (GC-MS) and an enzyme-linked immunosorbent assay. Analyte identification and annotation involve not only MS library searching metabolomics methods, but also an innovative stochastic dynamics mass spectrometric method using Equations (1)–(6) (below).

Mass spectrometric database searching metabolomics methods are involved in analyte identification and annotation. The development of innovative exact methods for data-processing in MS measurands is of significant importance. The highlights focus on silylated carbohydrate metabolites. 

With regard to paper content, Section 2.4 schematically describes the metabolomic annotation of CBs. It is argued that robust chemometric methods of analysis of variance (ANOVA) struggle to achieve excellent-to-exact method performances when comparing measurands of the metabolites of microplants of *Pyrus communis* L. with the MS library of standard analytes. Section 2.5 illustrates how Equations (2), (3) and (6) play important roles in distinguishing between CB isomers. The virtual identity of CB fragmentation patterns causes comparable parameters of chemometrics. This fact challenges their unambiguous annotation via database searching methods for metabolomics. The dynamics of analyte concentrations in relation to the experimental conditions of magnetic-pulse measurements are examined as well.

## 2. Results

### 2.1. Chromatography and Mass Spectrometric Data

Chromatographic and MS data on the metabolites of *Pyrus communis* L. were used to identify and annotate eighty-nine analytes of alcoholic extracts of pear microplants. Thirty-two of them belonged to organic acids; thirty-four were CBs; five were phenolic compounds; five were amino acids, and two were their derivatives, respectively. Palmitic acid, N-acetylglucosamine, and nine analytes were also determined from other chemical groups (Table 1).

Naturally, organic acids and phenols have pronounced antioxidant effects [1,2,3,4,5,6]. Sugar alcohols, such as erythritol, arabitol, and ribitol, are products of CB metabolism. Differences in the biosynthesis of metabolites by microplants depending on the duration of the period after magnetic-pulse treatment were observed. Sixty-seven metabolites were biosynthesized in microplants at about t = 3 h after magnetic-pulse treatment, and 72 h later, in total, thirty-one metabolites were biosynthesized (Table 2).

The tabulated data indicate that magnetic-pulse treatment activates the biochemical synthesis of different metabolite classes, and the time duration of process is limited to several days. Pear microplants were infected with latent viruses. Therefore, we are able to propose the stimulation of nonspecific resistance under the action of magnetic-pulse treatment by means of the biosynthesis of metabolites, exhibiting antioxidant and antiviral biological activity. The functional relationship among the concentration of CA in tissues, the treatment mode, and the time duration after magnetic-pulse treatment is established. There is a decrease in the CA concentration in tissues 3 h after the magnetic-pulse treatment of explants (Figure 1).

At about 72 h after magnetic-pulse treatment (pulse frequency ω = 1–10 Hz, variant 4), an 8.75-fold increase in CA concentration is observed compared to without treatment (Figure 2). We obtained a recovery factor in the analytical method of R = 4.79 (see Equation (1) in work [9] and data in Appendix A). The CA biosynthesis in plant tissues is associated with the development of nonspecific resistance. Appendix A show chromatographic and MS data.

After 14 days, the CA concentration decreases to the control level. The temporal distribution of CA concentration levels in relation to the applied magnetic field shows complex dynamics (Figure 3 and Appendix A). The ANOVA data on analyte concentrations at ω = 1–10 and 51 Hz show a lack of statistically significant differences between datasets of values. The same can be said for measurands for the analyte concentration in relation to the applied magnetic field within a time duration t = 72–168 h. χ^2^/DoF data on the goodness-of-fit test by Cochran and Kolmogorov–Smirnov tests are also shown (see pages 35 and 118 [10].) The DoF denotes the number of degrees of freedom. It is equal to the number of curve points assessed statistically minus the amount of data, which is not fixed. These data allow us to evaluate the reliability of the fitting approach to the experimental pattern, due to the measurable variables.

The relationship between analyte concentration and the applied magnetic field energy at t = 72–168 h can be regarded as a continuous deterministic process [11] at a probability level (coefficient of determination r^2^ = 0.8685_6_–0.9912_5_). Conversely, the temporal distribution of the concentration with respect to the applied magnetic field energy is described by a nonlinear function, having a significant r^2^ parameter. These results assume that fluctuations in measurands depending on the experimental conditions should not be neglected. We have grounds for thinking that further systematic research efforts on the latter issue will allow us to explain the obtained relationships via stochastic kinetics [11].

Furthermore, we also examined the effect of magnetic-field treatment on the concentration dynamics of phenolcarboxylic acids, thus showing complex dynamic relations. At about 3 days after magnetic-field treatment (ω = 1–10 Hz, AMIS-8), pear explants exhibited increases in the concentration of gallic (2.4 times) and salicylic (28 times) acids. Magnetic-pulse treatment causes changes in the biosynthesis of CA and phenols in pear microplants. The maximum level of CA concentration in plants depends on the magnetic-pulse treatment regime. An enhancement of CA biosynthesis is observed 3 days after magnetic-pulse treatment exposure. It is acknowledged that CA participates in the phenolic defense complex of plants [12,13]. Depending on the experimental design of magnetic-pulse measurements, the biosynthesis of phenols in plants can be stimulated, thus involving them in the formation of nonspecific plant resistance to viruses.

The biosynthesis of monoterpene aucubin at about 3 h after magnetic-pulse treatment is also evidenced by MS metabolomics. It exhibits antiviral biological activity [14].

### 2.2. Mass Spectrometric Data on Aucubin

Adducts of aucubin ([M+Na]^+^ or [M+NH_4_]^+^) at *m*/*z* 269 [11] and 364 [15] have been observed, depending on the experimental MS conditions. The analyte fragmentation in positive polarity is associated with the cleavage of D-glucose (Glc) fragments [16,17], thus producing [M–162]^+^ ions at *m*/*z* 184 from charged iridoid glucoside [16]. The molecular [M]^+^ ions appear at *m*/*z* 346 [16,18]. A radical mechanism has been proposed [16]. Under FAB-MS ionization conditions and negative polarity, [M–1]^−^ molecular anions have been detected [19], while chemical ionization causes competitive H_2_O or NH_4_^+^ loss of [M+NH_4_]^+^ species [15,18]. Ions at *m*/*z* 346 of [M+NH_4_–H_2_O]^+^ of α- and β-C^6^-OH stereoisomers have also been assigned. The differences in the intensity values of ions of stereoisomers are important for the purposes of this study. The aglycone ([A+NH_4_]^+^, *m*/*z* 202) adduct is subsequently cleaved, thus producing two solvent water molecules. There are ions at *m*/*z* 184 and 166 of [A+NH_4_–H_2_O]^+^ and [A+NH_4_–2H_2_O]^+^. However, α- and β-C^6^-OH stereoisomers show insignificant differences in the intensity of the peak at *m*/*z* 184. Conversely, MS ions at *m*/*z* 166 and 346 ([M+NH_4_–H_2_O]^+^) distinguish between stereoisomers in the mixture. Nonetheless, Equation (2) is capable of determining species in the mixture, assessing the intensity of the peak at *m*/*z* 184 and quantifying the fluctuations in MS measurands [20,21,22]. Under ESI-MS conditions and negative or positive operation modes, aucubin reveals [M–H]^−^ or [M+H]^+^ ions at *m*/*z* 345 [23,24] or 347 [25]. The MS peak at *m*/*z* 405 can be assigned to the CH_3_COOH adduct ([M–H+CH_3_COOH]^−^). The formic acid adduct at *m*/*z* 391 ([M+HCOO^−^]^−^) has also been observed [26,27,28]. Collision-induced dissociation tandem mass spectrometry (CID-MS/MS) leads to the cleavage of D-glucose residue and water molecules, thus producing Y_0_^−^ ions (*m*/*z* 183 [M–H–Glc]^−^) and anions at *m*/*z* 165 ([M–H–Glc–H_2_O]^−^). The intramolecular rearrangement of Y_0_^−^ anions yields MS peaks at *m*/*z* 139 and 97 (Appendix A). MS identification in *Globularia alypum* L. and related species [29,30], *Cibotii rhizoma* [31], *Verbascum speciosum* [32], *Pedicularis* L. [24], *Veronica persica* [26], *Plantago species* [13,33] has been carried out. The analyte tends to stabilize adducts of ammonium or alkali metal cations, in addition to carboxylic acid ones. Both the radical and protonation mechanisms of ionization in aucubin have been observed. Substituents such as the COOH group at the C^4^ position of the aglycone fragment do not significantly affect major fragmentation reactions. There are competitive reactions of intramolecular cyclization with the participation of the C^4^-COOH group, however [23,34]. Looking at the MS data on aucubin, one should expect that the prediction of its MS spectrum does not represent a trivial task.

### 2.3. Mass Spectrometric Data on Chlorogenic Acid

Figure 4 depicts fragmentation reactions of chlorogenic acid. One might argue that its MS fragmentation processes have been extensively examined [35,36,37,38,39,40,41,42,43,44,45,46,47,48,49,50,51,52]. Thus, we only sketch out MS analysis of CA showing the intensity ratios of the peaks at *m*/*z* 191 and 179, thus quantitatively distinguishing not only among its positional 1-O-, 3-O-, 4-O-, and 5-O-isomers, but also between *trans*- and *cis*-isomers of the product 3-(3,4-dihydroxy-phenyl)-acrylic acid anions [35]. Ions at *m*/*z* 179 and 163 stabilize different tautomers. The recent application of Equation (2) to quantify tautomers of flavonoid-O-glycosides [22] revealed its ability to exactly quantify tautomers. Therefore, it determines the *cis*/*trans*-isomerization and tautomerism of 3-(3,4-dihydroxy-phenyl)-acrylic acid and 3-(3,4-dihydroxy-phenyl)-propenal anions as well as the position of the negative charge of 5-carboxy-2,3,5-trihydroxy-cyclohexanol anion based on theoretical intensity data on the peaks at *m*/*z* 179, 163 and 191. A negative operation MS mode is mainly used to quantify CA in foodstuffs. However, isomers of 3-O-, 4-O-, 5-O-caffeoylquinic acid show virtually identical *m*/*z* values of ions at *m*/*z* 353 (or 354 of [M]^−^ anion), 191, 179, 173 and 135, respectively [24,25,28,31,33,38,39]. Fragmentation reactions of [M+H]^+^ cations yield ions at *m*/*z* 337.1070, 181.0567, 163.0452, 135.0497, and 145.0337, respectively [49].

The abundance of the MS peak at *m*/*z* 163 depends on the experimental conditions. Ion mobility data assume neutral loss of the molecule 3-(3,4-dihydroxy-phenyl)-propa-1,2-dien-1-one [45]. ESI-MS/MS conditions allow for observing ions at *m*/*z* 161 [46,47]. Atmospheric pressure chemical ionization (APCI) MS shows peaks at *m*/*z* 354.9 and 163.1 [38]. A chemical diagram of ion 163__b_ is presented in [40], while that of ion 163__c_ is presented in [43]. The fragmentation pattern within *m*/*z* = 0–100 is assigned in [40,41,43]. Ref. [47] details the MS characteristics of CA isomers employing ultra-high-accuracy ion mobility MS data (see Table 1 in [47]).

### 2.4. Mass Spectrometric Data on Silylated Carbohydrate Monomers

The analysis herein solves problems caused by automated MS spectra library searching metabolomics methods. This issue can be highlighted by considering chromatograms and MS data on silylated CBs (Figure 5 and Appendix A; RTs = 18.15 and 25.59 min). Metabolomics assigns CB (A) to standard silylated D-gluconic acid.

However, MS fragmentation patterns on CB monomers and oligomers show that their structural analysis does not represent a trivial task, due to the virtual identity of the mass spectra of CBs. There is a lot of difficulty in this process due to a lack of regioselective chemical substitution; the presence of polydisperse CBs and molecular structures in the mixture; the diversity of isomers of CB monomers, oligomers, or polymers; and the capability of stabilizing linear and branched molecular structures, respectively. CBs show molecular conformational flexibility, intramolecular proton and charge transfer reactions, intramolecular rearrangement, and cyclization processes.

An important question is whether Equations (2), (3) or (6) can be successfully implemented in determining CBs. The results so far give us reason for thinking that both the GC- and (HP)LC-MS methods are capable of producing exact chemometrics and reproducible datasets in MS variables of CBs. 

What makes the use of Equations (2) and (3) appropriate in the case of the analysis of CBs (A) and (B) is that one can assign species precisely, which cannot be achieved by means of metabolomics methods. Using the chemometrics of not only *m*/*z* variables but also relative intensity (r.i.) data on the NIST standard spectrum of silylate D-gluconic acid, assessing MS peaks at *m*/*z* 45, 73, 103, 117, 129/133, 147, 189/191, 205, 217, 243/245, 271, 305/308, 319/320, 331/333, and 359/361 results in |r| = 0.9489 (CBs (A) and (B)), 0.9753 (CB (A) and the NIST standard) and 0.7714 (CB (B) and the NIST standard), respectively (Appendix A). Despite the fact that ANOVA tests show not significantly difference in *m*/*z* and intensity measurands (Table 3 and Appendix A), we cannot assign CB (B) to silylated D-gluconic acid. However, CB (A) is assigned to silylate D-gluconic acid at a statistical significance of |r| = 0.9488_5_ when examining GC-MS data on the NIST standard and at |r| = 0.9393_7_ (Appendix A) when correlating MS measurands of CB (A) with electron impact MS data based on the NIST standard for D-gluconic acid, 2,3,4,5,6-pentakis-O-(trimethylsilyl)-trimethylsilyl ester. Correlative analysis of the data on the latter analyte and CB (B) yields |r| = 0.9036_6_. Measurands of CB (B) according to the GC-MS and EI-MS NIST standard databases correlate mutually (|r| = 0.9925). The assignment of analyte product ions assumes the formation of δ-gluconolactone. CBs (A) and (B) MS variables show significant standard deviation (sd(yEr±)) and the mutual correlation of mesurands (|r| = 0.9489). ANOVA tests are used to assess whether variables are normally distributed or random. Histograms, probability plots, and the Shapiro–Wilk statistical method are used to assess the normality of distribution of measurands. When the plot follows a straight line, then variables are normally distributed. Conversely, when there is a lack of a linear plot, then the variables are random (Appendix A). The histogram and probability plots of CBs (A) and (B) show few points out of line. However, due to the relatively small number of measurands, whole datasets of variables are used. Shapiro–Wilk parameters assume a not normal distribution. This test is not robust. It is used with histogram and probability plots.

In assigning MS peaks of CB (B) via Equations (2), (3) and (6), we used the molecular structures of MS ions proposed on the basis of GC-MS analysis of products of γ-radiolysis of α-D-glucose, α-D-fructos, cellobiose, and their silylated derivatives, respectively [53,54,55,56,57,58,59,60,61,62,63,64,65,66]. Chemometrics of measurands of CBs (A) and (B), silylated 4-desoxy-glucose, and silylated α-2-deoxy-D-glucose (2DG) show |r| = 0.7753 when looking at the results from CB (A) and 4-desoxy-D-glucose (4DG); |r| = 0.9411_5_ when examining ions of CB (B) and 4DG; |r| = 0.9743_2_ when looking at data on 2DG and 4DG; and |r| = 0.9494_5_ when examining the variables of 2DG and silylated D-gluconic acid (Appendix A). The |r| = 0.9743_2_ is high when comparing with data on CB (B) measurands and silylated D-gluconic acid (Figure 6). ANOVA tests show a lack of significant statistical differences between variables of CB (B) and 4DG. The same can be said when looking at variables of 2DG and 4DG (Appendix A). 

Therefore, with a probability of |r| = 0.9411_5_, CB (B) can be assigned to silylated 4-desoxy-glucose, while the probability of assignment of the latter compound to silylated 2-desoxy-glucose is |r| = 0.9876 (Figure 7). Appendix A depicts linear relations among measurands of standards of silylated α-2-deoxy-D-glucose, D-glucose, and D-gluconic acid. There is a decrease in method performance (|r| = 0.8767_8_–0.7062_8_). The silylated CBs show an abundance of ions at *m*/*z* 73 [67,68,69], assigned to [Si(CH_3_)_3_]^+^ ions (below). 

A comparative theoretical analysis with cations of [SiH(CH_3_)_2_(CH_2_)]^+^ is conducted. The intensity data on MS peak at *m*/*z* 205 show marked differences in values across silylated monosaccharides. Correlating GC-MS data linearly on CBs (A) and (B) with standard MS spectra, a significant effect of intensity of cations on the |r| parameter is obtained. It is assigned to [C_8_H_21_O_2_Si_2_]^+^ ions produced via ^2,4^A cleavage of the CB skeleton, when there is C^4^-O-Si(CH_3_)_3_ (Appendix A) substitution. It can be assigned to product ion at *m*/*z* 217 via ^0,3^X cleavage, when there is a 4-desoxy-CB derivative (Figure 5). The assignment of CB (B) to silylated D-gluconic acid using intensity data on the peak at *m*/*z* 205 shows a probability value of |r| = 0.7714_3_. Due to the comparable |r| parameter in thet measurands of CB (B) and he spectra of standards of silylated CBs, in addition to the statistical similarity of experimental variables according to ANOVA tests, we are unable to unambiguously assign CB (B) to silylated α-2-deoxy-D-glucose or silylated 4-deoxy-D-glucose. Evaluating intensity data on MS ions at *m*/*z* 205, it can be assumed that CB (B) is 4DG, because ^2,4^A cleavage is a major fragmentation path of CB monomers. Its low abundance in the mass spectrum of CB (B) assumes that 4-deoxy-monosaccharide is substituted, rather than α-2-deoxy-D-glucose.

However, there is a difference in the |r| parameters in MS data on CB (B) and the NIST standards of 4DG and 2DG of Δ|r| = 0.0458_6_ (|r| = 0.9451_1_ and 0.9870_1_; Figure 6, Figure 7 and Appendix A). Owing to that MS peak at 361 together with the characteristics of the product ions of CBs observed in the spectra of silylated maltose and D-glucono-1,5- and 1-4-lacton shown above, we carried out chemometrics analysis to compare the intensity data of the peaks of CB (B) with GC-EI-Orbitrap MS/MS data [63] (Appendix A). Despite the virtually identical *m*/*z* data on CBs, we were unable to assign CB (B) to the lacton of D-gluconic acid (|r| = 0.678–0.5616_2_). In order to justify CB (B) assignment, the following subsections provide details on the application of Equations (2), (3) and (6) to this issue, highlighting their importance for distinguishing species in complex cases in the MS analysis of CBs. As the results from this subsection show, even the employment of robust chemometrics methods cannot provide unambiguous assignment of the MS spectrum of unknown compounds compared with measurable variables of standard samples from the MS spectrometry library and searching algorithms.

**Figure 5 ijms-24-16776-f005:**
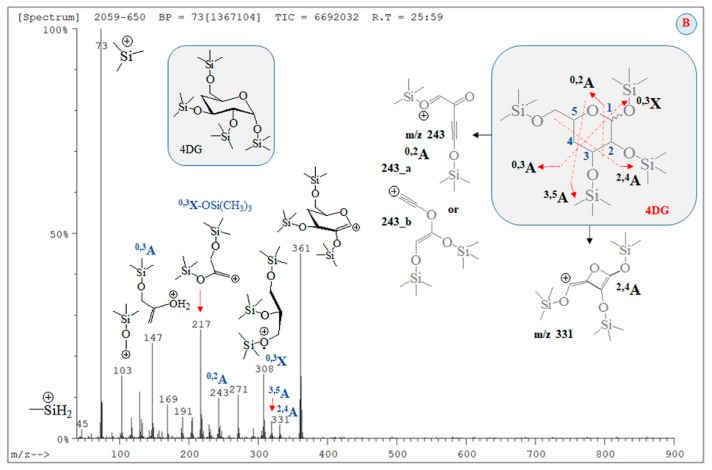
GC-MS spectrum of carbohydrates (**B**) (RT = 25.59 min; Appendix A) assigned to silylated 4-desoxy-glucose; chemical diagrams of product ions; assignment of carbohydrate species according to the nomenclature provided by Dommon and Costello [60], Stephens and coworkers [61], as well as Spina and coworkers [62], respectively.

**Figure 6 ijms-24-16776-f006:**
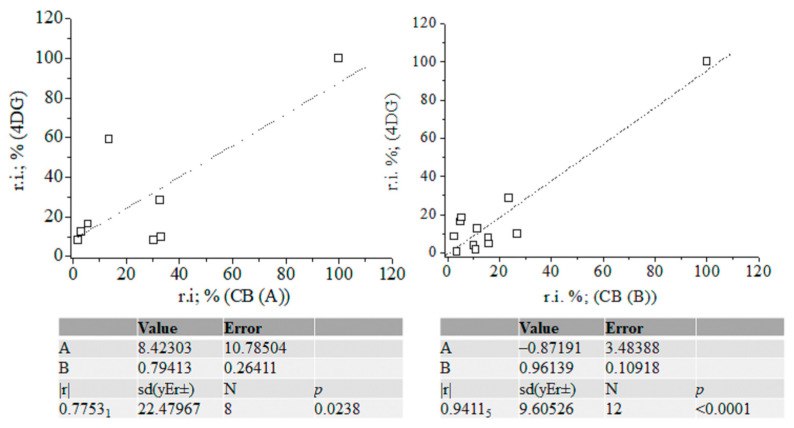
Linear correlation between relative intensity data (%) of fragmentation ions of 4-desoxy-glucose and carbohydrates (A) and (B); chemometrics.

**Figure 7 ijms-24-16776-f007:**
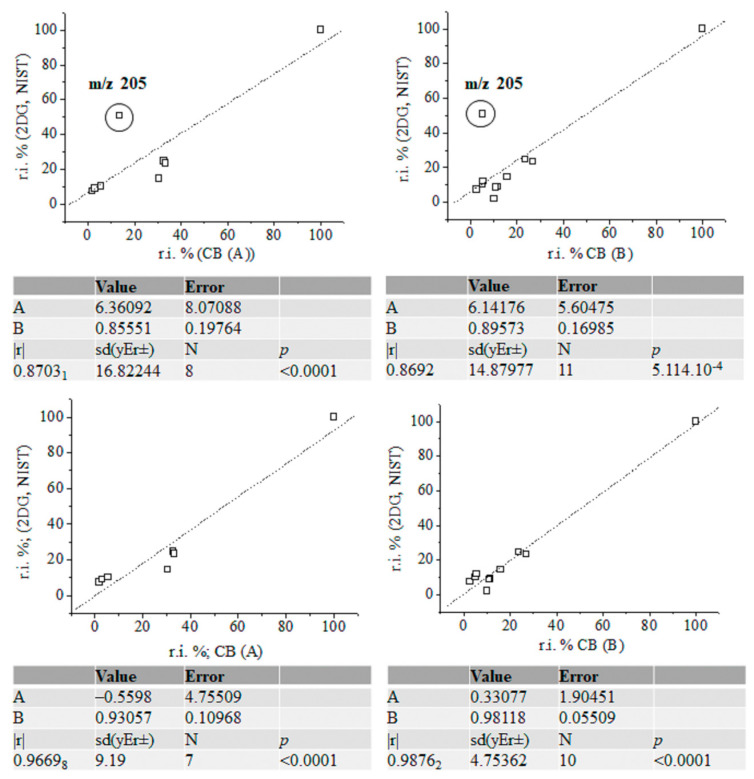
Linear correlation between relative intensity data (%) of fragmentation ions of silylated α-2-deoxy-D-glucose (NIST standard), carbohydrates (A) and (B) with and without accounting for the intensity data on ions at *m*/*z* 205; chemometrics.

### 2.5. Theoretical Stochastic Dynamics Mass Spectrometric and Quantum Chemical Data

This subsection deals with assigning ions of low-molecular-weight analytes in MS, not only employing annotation searching algorithms, but also Equations (1)–(6) in the data processing in MS measurands [20,21,22]. In the preceding subsections, we argued for the necessity of an in-depth analysis of the relationship between MS variables and the molecular structure of product ions. Depending on the experimental conditions in MS measurements, significant variation of ions and measurands can be observed. Charge transfer phenomena, tautomerism, and intramolecular rearrangement are highlighted, causing competitive mechanisms in MS cleavage processes. The analytes tend to form adducts with inorganic or solvent species, thus further complicating the MS fragmentation patterns. Considering the MS data on aucubin, CA and silylated CBs with which we began Section 2, we used a set of different correlation model equations assessing the relationship between MS measurands and molecular structures in analyte ions. Thus, herein, we provide the best explanation and prediction based on MS data for metabolites connecting between different theoretical methods. The results illustrate why this was necessary, via independent methods. Frequently, the established model equations do not describe or predict actual observable MS processes not only. This fact is due to the complexity of MS phenomena across ionization MS methods, analytical techniques, synthetic conditions, specific molecular properties, and chemical reactivity in analytes, respectively. 

To begin with, we utilized Formula (2) and its relation to Equation (3), as well as approximating Formula (6) of Equation (2). The D_QC_ data provided by Equation (3) were calculated using vibrational frequencies in GS and the saddle point (ν_i_^(0)^ and ν_i_^(s)^) of ions. Since the type of product ion depends on the experimental conditions, we examined the relationship, assessing a large number of ions, their tautomers, and adducts, via chemometrics. MM2-MD data on aucubin (Appendix A; Appendix A) show that the O-center of the aglycone-O-glucoside bond is the most preferred protonation position from the perspective of thermodynamics. Thus, the main fragmentation path of [M+H]^+^ ions shows cleavage of Glc residues. This is not applicable to iridoid glucoside [15,18]. Due to these reasons, this study accounts for both [M+NH_4_]^+^ and [M+Na]^+^ ions at *m*/*z* 364 and 369 [64,65]. Depending on the experimental conditions and molecular properties, NH_4_^+^ analyte adducts can be also determined by means of GC-MS [66]. The [M+NH_4_]^+^ and [M+Na]^+^ ions show a cleavage of Glc fragments [67]. MS peaks at *m*/*z* 263 and 283 have been used as chemical markers in determining aucubin in honey [68]. Molecular rearrangement of aucubigenin fragments is found when examining ions at *m*/*z* 184. The subsequent loss of the CO group produces a peak at *m*/*z* 333. In addition, iridoid glucosides form dimers depending on the substituents at the aucubigenin fragment [69]. They also tend to stabilize adducts. The cations [2M+NH_4_]^+^ and [2M+Na]^+^ were identified [57]. However, this study does not focus on analyte dimers. Figure 5 depicts chemical diagrams of species at *m*/*z* 167, 149, 131 and 121, their tautomers, and products of intramolecular rearrangement, while Appendix A provides details on GS and TS MD and the static computations. The vibrational modes, atomic coordinates of the ions, and their energetics are summarized in Appendix A. I_theor_ data provided by Equation (6) and the D_QC_ parameters are also calculated. The MS ion 167_C^6^_O^+^H_2_ (*m*/*z* 167) is characterized by intramolecular charge transfer and rearrangement reactions. The most stable is the 167_C^8^_O^+^H_2_ cation (Appendix A). The same is true for the cation at *m*/*z* 149 (149_C^10^-O^+^H_2_). It is described as a H_2_O solvate adduct of 7-methylene-4a, 7-dihydro-cyclopenta[c]pyran. This result agrees well with the observed MS peak at *m*/*z* 131 of the 7-methylene-7H-cyclopenta[c]pyranylium cation (131__a_; Figure 8). Its intramolecular rearrangement yields the most stable form, 131__b_, of the ion at *m*/*z* 131.

An exact linear relation between the D″_SD_ and D_QC_ parameters of Equations (2) and (3) [22] is established, looking at the experimental intensity data on MS ions and D_QC_ values. The D_QC_ value accounts for the energetics of ions or their free Gibbs energy. The intensity in MS ions accounts, in fact, for the energetics of ionic species. However, we computed the vibrations of ions at GSs and TSs. There is a lack of a direct relationship between measurable variables’ MS intensity and free Gibbs energy. Appendix A depict MM2 and DFT-BO data on MS in MS ions of CA. The most stable form of neutral CA has a quinoide-like structure, instead of an aromatic one. Depending on the negative charge position of [M–H]^−^ anions at *m*/*z* 353, the most stable form is the *trans*-353_C^4^_O^−^ anion among the examined series of ions, having a *trans* configuration of a 3-(3,4-dihydroxy-phenyl)-acrylic acid structural unit. Comparative analysis between the *cis* and *trans* forms of the same anion shows that the *cis* configuration is more stable, exhibiting differences in total energy, ΔE^TOT^ = |1.9148| kcal/mol. The difference in the energy of ions causes differences in experimental MS intensity data on species at *m*/*z* 353 depending on *cis*- or *trans*-isomers. This phenomenon is used to quantify these two isomers, examining intensity data on MS peaks at *m*/*z* 353 and those of the anions at *m*/*z* 191 [40]. Theoretical analysis of the anions at *m*/*z* 191 and 179 can be found. The most stable is the 191_C^5^O^−^ anion. Aromatic T1 tautomers of the ions at *m*/*z* 179 are more stable species compared with their T2 counterparts, which have a quinoide-like structure. The *cis* configuration shows lowest E^TOT^ energy (E^TOT^ = −21.46 kcal/mol.) The *trans*-isomer shows E^TOT^ = −18.58 kcal/mol. 

Correlative analysis between the experimental MS I^TOT^_av_ and theoretical I_theor_ parameters of Equation (6) in aucubin shows values of |r| = 0.8157–0.99 (Appendix A). The results from CA are based on the MS peaks at *m*/*z* 354, 353, 191, 179, 161, 135, and 128, respectively. GC-MS analysis of CA has been conducted using molecular ions at *m*/*z* 354.31 [52]. In particular, the intensity ratio of the peaks I^191^/I^179^ shows the following values, depending on the CA configuration: 2.36 (3-*cis*-), 18.59 (3-*trans*-), 46.76 (5-*trans*-), 6.30 (5-*cis*-), 8.30 (4-*cis*-), 113.7 (4-*trans*-) isomers, respectively. Importantly, the deviation from the exact |r| = 1 parameter is explained via the approximation of Equation (6) instead of employing Formula (2) exactly, thus neglecting fluctuations in measurands. Experimental data on the total average intensity values are assessed.

Appendix A depict data on the product ions of silylated 4-desoxy-glucose (Figure 5). We show the energetics and atomic coordinates of these species. Figure 9 and Table 4 show the chemometrics of the experimental intensity of ions at *m*/*z* 45, 73, 103, 147, 205/204, and 361/359 of CB (A), CB (B), 2DG, and 4DG, respectively. A value of |r| = 0.9963_6_ is obtained when looking at data on CB (B). There is a decrease in method performances when examining species of CB (A) (|r| = 0.9898_3_). The results from the standard GC-EI-MS/MS spectra of 2DG and 4DG show lower |r| parameters and comparable values. There is a difference in Δ|r| = 0.0107_8_, examining four common CB ions. The results underline that metabolomics searching algorithms are incapable of unambiguously distinguishing between 2DG and 4DG when looking at their standard MS spectra. In such cases, a comprehensive stable isotope labelling MS analysis within the framework of Formulas (2) and (6) is required. This process can precisely and reliably assign observable fragmentation ions to 3D molecular and electronic structures of the analytes. This task is required due to their virtual similarity when conventional MS methods and techniques are used. With a probability level of |r| = 0.9963_6_, the upshot of application of Equation (6) in order to assign the MS spectrum of CB (B) agrees well with the chemometric data presented above.

In a nutshell, we utilized Equation (6), making explicit remarks on the fact that it predicts the mass spectrum in analytes without assessing fluctuations in measurands. Therefore, Equation (6) is unable to determine positional stereoisomers in mixtures due to difference in the intensity values in MS peaks. Despite the excellent data provided by Equation (6), it can be argued that the exact quantitation of (stereo)isomers in aucubin, CA, and silylated carbohydrate monomers requires the employment in Equation (2). For the purposes of the identification of unknown analytes in the mixture, Equation (6) produces reliable analytical information from the perspective of chemometrics. However, it does not account for the subtle electronic effects of tautomers or fluctuations in measurands in MS ions of enantiomers. The I_theor_ data provided by Equation (6) correlated well with the GC-MS data, thus agreeing with the results reported so far [67,68,69,70]. The same is true for chlorogenic acid [35,50,51] and silylated 4-desoxy-glucose [58,59,60,61,62,63,64,65,66,67,68,69,70].

## 3. Discussion

This section begins with the puzzling question of how external magnetic field pulsed energy impacts upon biochemical reactions in living systems such as microplants. It is a known fact that there electric currents are induced in plant tissues, which perturb cell membrane potentials. There is thus an effect on biochemical reactions and plant metabolism [2,71]. The electromagnetic fields modify ion transport rates across the plasma membrane. They affect the dynamics of the cell membrane lipid protein structure. This process alters the permeability of the plasma membrane [2]. The latter cellular structure regulates molecular and ionic transport, among many other biological functions. The perturbation of membrane transport biological activity may alter cellular metabolic processes. It has been found that the application of external magnetic fields of about 50 Hz can increase amino acid uptake into plant roots in vivo. State-of-the-art studies into this issue evidenced that depending on the experimental design of magnetic-pulse measurements, there is crucial effect of magnetic pulsed energy on the defense mechanisms of plants. A modification to the plant metabolism and immune reactions against plant viruses has been proposed [6]. There are other approaches that can be used to protect plants against the negative effects of viruses, such as chemotherapy or thermotherapy. However, they suffer from the drawbacks detailed in [6]. Conversely, the elaborated approach to external physical factors shows a set of advantages. These include high technological efficiency, instrumentation portability, automation, low energy intensity, and more. It is also safe for humans. Due to these reasons, the development of a magnetic pulsed energy approach to improve plant metabolism is regarded as a prominent and innovative tool with a broad spectrum of applications in fundamental research and industry. So far, it has been argued that depending on the type of host plant and virus, not only plant growth is stimulated, but also that of pathogens [6,72,73,74,75,76,77,78]. Arguments have extended to the effect of magnetic-pulse treatment on biochemical reactions of plants and their metabolism, highlighting the treatment’s effects on both plant growth and pathogens depending on not only on the type of host plant, but also on the type of virus [6,78]. Therefore, plant metabolites depend crucially on the environmental conditions, type of plant and the type of virus, due to the obvious connection between cell growth and its biochemistry [6,75,76,77]. This raises the question of how external magnetic-pulse fields affect plant metabolism. To address this question reliably, we immediately face another question of how we can ensure that certain standardized measurement parameters are adhered to. The latter question is addressed by highlighting the elaboration of a magnetic-pulse treatment device at the Federal State Budgetary Scientific Organization ‘Federal Horticultural Center for Breeding, Agrotechnology and Nursery’ (patent of the Russian Federation No. 2652818) as a plausible and reliable approach to precisely studying plant metabolism upon magnetic-pulse-field exposure [6]. Its effectiveness has been proven already, despite the low number of results reported so far. Recently, the metabolites of virus-free raspberries microplants have been evaluated as well [7]. The current study first examines the effect of magnetic-pulse treatment on the metabolism of microplants of *Pyrus communis* L. It provides details on the dynamics of the concentration of chlorogenic acid in vitro, thus providing new insights into biochemical reactions upon magnetic pulsed energy [2,6,7,71,72,73,74,75,76,77,78].

Let us turn the reader’s attention to the fact that monoterpene aucubin has been identified and annotated at about 3h after magnetic-pulse treatment of microplants. This metabolite has proven antiviral biological activity. In discussing how magnetic-pulse treatment affects *Pyrus communis* L. metabolism, with significant prospects for broad interdisciplinary fields of fundamental research and many industrial branches, it is important to highlight that the enrichment or biofortification of crop plants with nutrients represent an effective and low-cost approach to facilitate plant introduction into the food chain [79,80,81,82,83,84,85,86,87,88,89,90]. It is recommended for soils, addressing the deficiency of micronutrients, thus highlighting its beneficial effects on plants as well as human and animal metabolism [79]. An enormous research effort has been concentrated on the development of methods for enriching plants with minerals such as ions of Zn, Fe, I, Se, Mg, and Ca [79,80,81,82,83,84,85,86,87,88,89,90]. The quality of plant raw materials, including their antimicrobial and antioxidant properties, crucially depends on the concentration levels of micronutrients. It, in turn, is affected by soil and climate, in addition to the conditions of food storage. The results from this study indicate that external magnetic-pulse treatment of plants affects the concentration levels of beneficial plant metabolites. With the effective increase in the levels of these metabolites in *Pyrus communis* L. plants, biological antioxidant and antiviral activity occurs. The external magnetic field may have beneficial effects on the humans health-promoting properties of these crops, in addition to the release of metabolite fertilizers that potentially regulate their growth and target activity. A magnetic field can cause an improvement in the characteristics of plant growth [90]. So far, only a few studies have evidenced that a magnetic field can influence the metabolite composition of plants and the analyte concentration. The same is true for data on the effect of magnetic fields on plant enzyme activation. Therefore, a comprehensive understanding of the molecular-level biochemical mechanisms of plant metabolism under the effect of a magnetic field remains far off. There is a need for further research efforts in order to implement this approach into the biofortification of crop plants. This study clearly shows magnetic-pulse treatment’s effectivity regarding the metabolite processes and enrichment of *Pyrus communis* L. metabolites with beneficial antiviral and antioxidant activity in humans. 

However, difficulties are faced in identifying and annotating plant metabolites by means of standard MS database searching algorithms used in metabolomics, highlighting the analysis of CBs and their silylated monosaccharides. It might be argued that statistical approaches to metabolomics do not provide sufficient method performance. They do not allow one to unambiguously and reliably annotate such analytes in plant tissue, due to the similarity of chemometric parameters and the acknowledged complexity of CB metabolites, as already demonstrated in previous subsections. Due to these reasons, a combination of theoretical methods can result in the reliable determination and structural assignment of plant metabolites, as has been exemplified by the current study. This study utilizes methods for MS metabolomics and stochastic dynamics mass spectrometry. It deals with quantitatively and structurally determining, in 3D, analytes via the chemometric assessment of the relationship D″_SD_ = *f* (D_QC_) of the parameters of Equations (2) and (3). Its predictive capability is extended further using Equation (6). This formula allows us to reliably elucidate the relations among MS measurable variables, molecular properties, and molecular structural parameters from the perspective of chemometrics. We obtained improved MS method performance when assigning unknown metabolites in plants. The application of the discussed model equations resulted in assigning silylated CB monomers and aucubin at performances of |r| = 0.9992_2_–0.9411_5_. We assessed a statistically representative set of fragmentation ions *m*/*z* 45, 73, 103, 117, 129/133, 147, 189/191, 205, 217, 243/245, 271, 305/308, 319/320, 331/333, and 359/361 when looking at silylated CBs as well as at *m*/*z* 167, 149, 131, and 121 when examining aucubin. 

## 4. Materials and Methods

### 4.1. Materials, Methods, (Bio)Synthesis, and Isolation

The quantitative definition of the chlorogenic acid was carried out by means of the High-Performance Liquid Chromatography (HPLC) method on a Knauer GmbH (Berlin, Germany) chromatograph with the Eurochrom HPLC Software (Version 3/05 P4). Acetonitrile composition eluent–0.03% TFC (trifluorouxic acid) was used at a ratio of 30:70. The detection in the samples was carried out at a 325 nm wavelength. *Pyrus communis* L. microplants (cv. Zagorievsky) were cultivated using a modified method based on Murashige and Skoog’s approach [91]. We added 6-benzylaminopurine with a concentration of 1 mg/L in standard conditions: photo-period, 16 h; T = 22–25 °C; and illumination, 3000 lux. Pear explants were infected with apple chlorotic leaf spot and apple stem grooving viruses. Diagnosis for viruses was conducted via enzyme-linked immunosorbent assays using Neogen kits (Lexington, UK). Enzyme-linked immunosorbent assay data were obtained by means of a StatFax 2100 spectrophotometer (Awareness Technology Inc., Palm City, FL, USA) at λ = 405 and 630 nm. Explants were treated with magnetic pulses via AMIS-8—analytical instrumentation elaborated at Federal State Budgetary Scientific Organization ‘Federal Horticultural Center for Breeding, Agrotechnology and Nursery’ (FSBSO ARHCBAN) (see Section 2.4). Microplants were extracted with 70% CH_3_CH_2_OH at about three and seventy-two hours after magnetic-pulse treatment. The extracts were filtered. Some of the extracts were used for HPLC analysis of chlorogenic acid. The quantitative definition of the chlorogenic acid was carried out using the High-Performance Liquid Chromatography (HPLC) method on a Knauer GmbH (Berlin, Germany) chromatograph with the Eurochrom HPLC Software (Version 3/05 P4). Acetonitrile composition eluent–0.03% TFC (trifluorouxic acid) was used at a the ratio of 30:70. The detection in the samples was carried out at a 325 nm wavelength. An aliquot of 800 µL was evaporated to dryness at T = 80 °C. Chemical substitution of metabolites was conducted using a silylating agent, *bis*(trimethylsilyl)trifluoroacetamide (see detail on [92]). Silylation in analytes was performed according to optimized synthetic methods based on the chemical substitution of test compounds, having a broad spectrum of physicochemical properties. The metabolites belonged to different chemical classes (consider silylated carbohydrates in Appendix A). The experimental design included the optimization of the experimental conditions of the chemical substitution, looking at the solvent type, silylation reagents, derivatization time, temperature, and more. The final parameters were determined on the basis of the efficiency of the synthetic method.

### 4.2. Metabolomics by Gas Chromatography-Mass Spectrometry

Microplants were dried in a hot air oven at T = 56 °C for 18 h. Then, they were powdered. Samples with a mass of 100 mg were extracted and place in 10 mL of CH_3_OH. These samples were then filtered and evaporated in the presence of a helium gas flow. Metabolomic analysis was carried out, utilizing a JMS-Q1050GC mass spectrometer coupled with gas-chromatography equipment (JEOL Ltd., Tokyo, Japan). A capillary column, DB-5HT (Agilent, Santa Clara, CA, USA), with a length of 30 m, an inner diameter of 0.25 mm, a film thickness of 0.52 μm and operating under helium was used. The experimental conditions of measurements involved temperature gradients within the range of T = 40–280 °C; an injector and interface temperature of T = 250 °C; and an ion source temperature of T = 200 °C. The column gas flow was 2.0 mL/min. A split-flow injection mode was used. The sample injected volume was 1–2 μL. Automated annotation of metabolites was performed using NIST-5 (National Institute of Standards and Technology, Gaithersburg, MD, USA) RTs and MS libraries (Appendix A). The scanning range was *m*/*z* 33–900 *m*/*z*. A credibility of metabolite analysis of 75–98% was achieved.

### 4.3. Analytical Procedure for Separation and Identification of Plant Metabolites

In discussing this issue, we shall focus on approaches to metabolomics, involving so-called targeted and untargeted analyses [93]. Hyphenated chromatographic methods coupled with MS allow the simultaneous identification and annotation of a large number of metabolites. The advantages of the GC method compared with the LC method have been highlighted [94,95,96]. The research tasks involve the chemical derivatization of metabolites. This study utilized the silylation approach, using trimethylsilyl ethers and GC-MS/MS analysis of substituted metabolites. This method is routinely used to identify and annotate low-molecular-weight analytes in plants, particularly highlighting CBs [93,94,95,96,97]. GC-MS yields a high reproducibility of chromatographic RTs. GC coupled with electron impact ionization MS has become a straightforward approach to annotate analytes via MS database searching algorithms, due to the extensive and highly reproducible analyte fragmentation patterns [94]. However, many silylated analytes are unstable [98]. This method is broadly used to examine unknown metabolites, lacking a suitable standard database MS spectrum. Despite its shortcomings, LC-MS offers good sensitivity in addition to a dynamic range. Comprehensive review articles devoted to the advantages and limitations of GC- and LC-MS methods can be found [96,99,100,101,102].

The MS methods are irreplaceable tools for many fields of analytical science, due to their superior method performances [20,103], showing (i) ultra-high-resolving-power accuracy, reliability, precision, reproducibility, selectivity, and specificity, in addition to (ii) an ability to determine both low-molecular-weight analytes and biomacromolecules. The instrumentation shows (iii) very low concentration limits of quantitation and detection ranging from fmol to attomole concentration levels; (iv) an ability to perform direct assays; (v) a flexible instrumentation scheme and hyphenated techniques coupling MS with chromatography, electrochemical methods, and more; (vi) portable and miniaturized instrumentation schemes allowing field analysis; (vii) adaptations to lab-on-chip technology; (viii) the ability to perform direct analyte imaging in biological samples; (ix) a continuous flow reaction monitoring operation mode; and (x) the ability to provide exact kinetics, ion mobility, thermodynamics, and diffusion analyte parameters, and more. The applications of MS methods encompass interdisciplinary research fields such as analytical chemistry, environmental research, petroleum chemistry, forensic chemistry, clinical diagnostics, medicine, pharmacy, biochemistry, toxicology, forensic medico-legal investigations, nuclear forensics, agricultural and food technology, geology, archaeology, and more. Among the fields of implementation of mass spectrometry, including molecular annotation, identification, and analyte determination, which have increased dramatically in scope more recently, metabolomics is developed as an omics tool, mainly using hyphenated chromatography, namely both GC and LC methods, coupled with mass spectrometry. It deals with the analysis of low-molecular-weight molecular metabolites having masses of up to 500 Da, which are synthesized as a result of cellular metabolism and biochemical reactions in vivo. Metabolomics provides an in-depth real-time understanding of biological processes. The key role of analytical mass spectrometric instrumentation in many aspects of molecular structural analysis is associated with the aforementioned method performances. Despite the fact that single-crystal X-ray diffraction represents the major tool for determining the 3D structures of biological molecules, its application requires relatively high amounts of metabolites, in addition to good samples of crystal growth. Often obtaining pure and good-quality single crystals of unknown analytes is a challenging research tast. In order to overcome the drawbacks of chemical crystallography, high accuracy tools for computational quantum chemistry have been developed, allowing us to obtain 3D molecular and electronic structures as well as the properties of metabolites. Molecular dynamics methods produce time-dependent data on the analyte structure and molecular properties in relation to experimental conditions. Computations of thermodynamics and kinetics parameters, binding affinity, ion mobility, catalytic activity, and diffusion in addition to 3D molecular structural parameters are routinely implemented in studies devoted to understanding biochemical processes in vitro and in vivo. There are also a set of robust experimental methods for molecular structural analysis detailing the metabolites involved into biochemical reactions, such as nuclear magnetic resonance and vibrational and circular dichroic spectroscopies. Despite their advantages,, there are often drawbacks to their widespread employment in the field of biochemistry, chiefly associated with the purity and amount of metabolites available. Conversely, the ability to obtain 3D molecular and electronic structural data on analytes via the complementarily application of mass spectrometry to quantum chemistry has been illustrated more recently in efforts to determine the isomers of molecules, complex cases of molecular electronic effects and tautomerism, and more [20,103]. 

### 4.4. Magnetic-Pulse Analysis of Plant Biochemistry

Magnetic-pulse treatment of samples was performed, using portable analytical instrumentation AMSI-8 elaborated at the Federal State Budgetary Scientific Organization ‘Federal Horticultural Center for Breeding, Agrotechnology and Nursery’ (FSBSO ARHCBAN). This is a patent of the Russian Federation No. 2652818. The instrumentation is designed for the automated programmable processing of garden planting materials. AMSI-8 operates in a periodic sequence of one- or multipolar magnetic induction pulses, both unmodulated and modulated, through attenuated oscillations in the ultra-low-frequency (HF) range in a frequency scanning mode with simultaneous additional synchronous irradiation with light pulses of certain wavelengths in the optical range in order to protect plants from viruses in vitro. It stimulates plant growth and development, causing an increase in productivity. The instrumentation scheme consists of an electronic unit, an inductor based on a multifilament flat spiral coil, and a light emitter of certain optical wavelengths [6,7]. It can be used to perform both laboratory and field analyses, thus having industrial-scale applications as well. This innovation shows a set of technical and economic advantages compared with commercial devices, such as: (i) ecological purity; (ii) a broad range of physical effects of pulsed magnetic-field treatment of plants without and with synchronous pulsed light irradiation of certain wavelengths in the optical range at all stages of plant development; (iii) a semiautomated operational mode; (iv) portability; (v) ease of operation; (vi) suitability for field analysis, and more. Table 5 lists the technical characteristics of AMSI-8. 

### 4.5. Theory/Computations

#### 4.5.1. Stochastic Dynamics Mass Spectrometric Approach

This study uses the recently developed stochastic dynamic Equations (1) and (2) [20,21,22], where the latter formula is derived from the former. D′_SD_ and D″_SD_ reflect the measurable variable intensity (I) of molecular and fragmentation product ions of analytes observed in mass spectrometric conditions of measurements, respectively.

(1)
$$D_{SD}^{tot}=\sum_{i}^{n}D_{SD}^{i}=\sum_{i}^{n}1.3194\;\times \;10^{-17}\;\times \;A^{i}\times \frac{\bar{I_{i}^{2}}-{\left(\bar{I_{i}}\right)}^{2}}{{\left(I_{i}-\bar{I_{i}}\right)}^{2}}$$


(2)
$$D_{SD}^{\prime \prime ,tot}=\sum_{i}^{n}D_{SD}^{\prime \prime ,i}=\sum_{i}^{n}2.6388\;\times \;10^{-17}\times \left(\bar{I_{i}^{2}}-{\left(\bar{I_{i}}\right)}^{2}\right)$$


(3)
$$D_{QC}=\frac{\prod_{i=1}^{3N}\nu_{i}^{0}}{\prod_{i=1}^{3N-1}\nu_{i}^{s}}\times e^{-\frac{\Delta H^{\#}}{R\times T}}$$


(4)
$$D_{SD,l}^{\prime \prime }+D_{SD,m}^{\prime \prime }=\left|r_{l,m}\right|\times \sqrt{\bar{I_{l,q}^{2}}-{\left(\bar{I_{l,q}}\right)}^{2}}\times \sqrt{\bar{I_{m,q}^{2}}-{\left(\bar{I_{m,q}}\right)}^{2}}$$


(5)
$$D_{SD}^{",q}\approx 2\times \frac{A_{D}^{q}}{A_{I}^{q}}\times I_{a\nu }^{TOT,q}$$


(6)
$$I_{SD}^{Theor}∼{\left(2.6388\;\times \;10^{-17}\times D_{QC}\right)}^{1/2}$$


Formula (2) overcomes one known drawback of classical quantitative MS-based methods, performing quantification of analytes in a solution. It provides exact analysis of condensed phases, looking at the quantitative criteria of chemometrics. This performance is associated with that of Equation (2), which opposes classical methods by quantifying the temporal variation in the measurand intensity pf MS peaks as an average over the span of the measurement time. It exactly quantifies the fluctuations in variables over the short span of the scan time. Formulas (1) and (2) are capable of determining the 3D molecular and electronic structures of analytes mass spectrometrically when Arrhenius’s Equation (3) is employed complementarily. Chemometrics can assess the relation D″_SD_ = *f*(D_QC_). Exact data-processing of MS variables with respect to the experimental factor collision energy (CE) within the framework of the same stochastic dynamic theory was obtained via Equation (4). This is derived from Equation (2), and it is valid for any of the two sets of measurable variables “*l*” and “*m*” related to the intensity values of the qth MS ion in the analyte within the framework of tandem mass spectrometric CID-MS^n^ measurements and n > 1. The correlation coefficient |r_l,m_| is obtained via intensity data on two sets of variables, “l” and “m”. After assessing the reliability of Equation (4), we achieved a value of |r| = 0.9999_9_ when studying the ESI-CID-MS^2^ to MS^7^ results from labetalol with respect to the CE value and the syringe-infused volume.

Furthermore, it has been found that the total average abundance (I^TOT^) parameters in MS peaks according to classical methods and the parameters D^i^_SD_ and D″_SD_ of Equations (1), (2) and (4) are mutually joined via Equation (5). Statistical data on A^D^ and A^I^ are obtained via the SineSqr approximation of functions D″_SD_ = *f* (CE) and I^TOT^ = *f* (CE).

So far, excellent-to-exact chemometric method performances of Equations (4) and (5) have been reported; however, few molecular examples have been examined. Therefore, in order to justify their ability to solve problems relating to the exact analysis of virtually identical MS fragmentation patterns, there is a need for further research efforts on testing the advantages and limitations of Equations (4) and (5). However, due to the latter formula, the experimental total average intensity at MS peaks is expressed in an indirect manner, which explains the fact that, frequently, there is a lack of exact |r| = 1 relations between I^TOT^_av_ and D″_SD_ parameters in MS ions.

For the purpose of the current study, we further extend the description of the functional relationship between experimental I^TOT^_AV_ and D″_SD_ data on MS ions and theoretical D_QC_ parameters, thus allowing us to predict MS fragmentation pattern in the analytes. 

Thus, owing to the fact that Equation (2) accounts for fluctuations in MS measurands and that the analysis so far has shown that <I^2^> > <I>^2^ as well as D″_SD_~D_QC_, Equations (2) and (3) can be written as Equation (6), where I_theor_ represents the approximated theoretical MS intensity value of the fragmentation MS peak of the qth ion under certain experimental MS conditions, within the framework of the discussed method. However, the fluctuations in the measurands are approximated, due to the fact that their absolute values are a priori unpredictable. The calculation tasks in Equation (3) are detailed in [20].

#### 4.5.2. Quantum Chemical Computations

The Gaussian, Dalton, and Gamess-US [104,105,106] program packages were used to compute data, which were visualized via GausView03 [107]. Ab initio and density functional theory (DFT) methods such as B3LYP, B3PW91 and ωB97X-D were utilized. Truhlar’s functional M06-2X was used [108,109]. The algorithm by Bernys determined GS. Stationary points at the potential energy surface (PES) were assessed by means of standard harmonic vibrational analysis. Basis sets such as cc-pVDZ by Dunning, 6-31++G (2d,2p) and quasirelativistic effective core pseudo potentials from Stuttgart–Dresden(–Bonn) (SDD, SDDAll) were used [110]. The ZPE and vibrational contributions accounted for up to a magnitude value of 0.3 eV. Species in the solution were examined via explicit super molecule and mixed-solvation approaches to PCM. The effect of ionic strength in the solution was accounted for via the IEF-PCM method. Merz–Kollman atomic radii and heavy-atom UFF topological models were used. The effect of pH was considered by computing data on neutral molecules, their cations, and their anions, respectively. MD computations were carried out by means of ab initio Born–Oppenheimer molecular dynamics, utilizing M062X functional and SDD or cc-pvDZ basis sets without considering periodic boundary conditions. The trajectories were integrated using a Hessian-based predictor-corrector approach with Hessian updating for each step in BO-PES. The step sizes were 0.3 and 0.25 amu^1/2^Bohr. The trajectory analysis stopped when (a) the centers of mass for a dissociating fragment was different at 15 Bohr, or (b) when the number of steps exceeded the given input parameter for the maximal number of points. The total energy was conserved during computations within at least 0.1 kcal/mol. The computations were performed by means of a fixed trajectory speed (*t* = 0.025 or fs) starting from initial velocities. The velocity Verlet and Bulirsch–Stoer integration approaches were used. 

Allinger’s molecular mechanics (MM) force field MM2 was used [111]. The low-order torsion terms were accounted for with higher priority than van der Waals interactions. The accuracy of this method compared with the experiment is 1.5 kJ·mol^−1^ of diamante [112], or 5.71.10^−4^ a.u. The differences in the heats of formation in alcohols and ethers are |0.04|–|6.02| kJ·mol^−1^.

The computation of Si-containing product ions accounted for scalar and spin-orbit relativistic contributions, in addition to total relativistic corrections. Molecular fragments containing heavy atoms exhibited relativistic effects playing an important role in the reliable description of the electronic structure of ions and their chemical bonding properties. The major relativistic effect in ^29^Si-containing ions cannot be regarded as a spin-orbit coupling one. Rather, it is a result of the scalar relativistic contraction of electronic s-shells [113,114]. Among the variety of relativistic approaches used to calculate the properties of heavy-atom-containing molecular systems, this study used the second-order Douglas–Kroll–Hess method (scf = tight units = bohr test int = dkh2 iop (3/93 = 1)) [115]. Correlative analysis of energetics and vibrational frequencies among theoretical methods was also performed.

The theoretical computations used, as well as the crystallographic data on iridoid glucoside aucubin as the input atomic coordinate parameters (CCDC 729042), were as reported previously [116].

#### 4.5.3. Chemometrics

The software R4Cal 0.1.12 Open Office STATISTICs for Windows 7 was used. The statistical significance was checked via a *t*-test. The model fit was determined via F-test. Analysis of variance tests were also used. The nonlinear fitting of experimental MS data was carried out by means of a searching method based on the Levenberg–Marquardt algorithm [117,118,119,120,121,122].

## Figures and Tables

**Figure 1 ijms-24-16776-f001:**
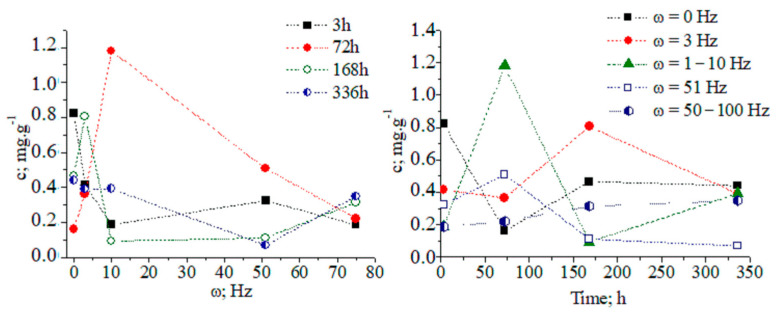
Concentration dynamics (c, (mg/g)) of chlorogenic acid in pear explants versus exposure time and frequency of magnetic treatment pulses (ω, (Hz)): 1–2–3; 2–51; 3–50–100; 4–1–10 Hz; and 5 − ω = 0 Hz (control).

**Figure 2 ijms-24-16776-f002:**
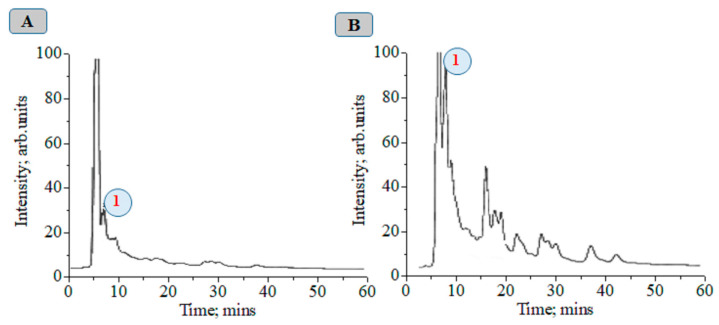
Chromatograms of plant alcoholic extracts at λ = 325 nm: 1—chlorogenic acid, control (**A**); 72 h after magnetic treatment with a frequency from 1 to 10 Hz (**B**).

**Figure 3 ijms-24-16776-f003:**
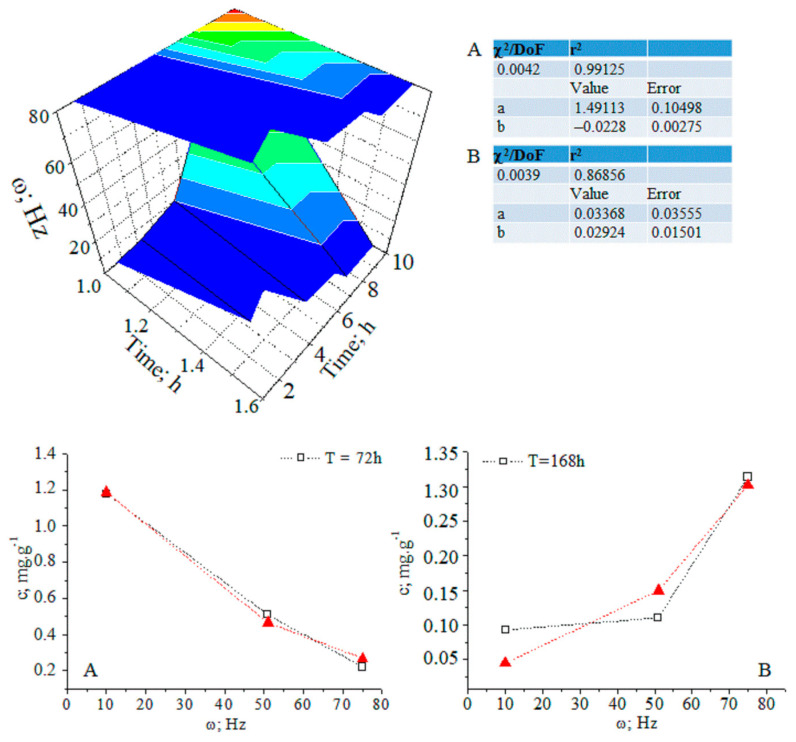
(**A**,**B**) Three-dimensional and two-dimensional plots of the temporal distribution of chlorogenic acid concentration versus time (h) and magnetic-pulse frequency (ω (Hz)); experimental data (black curve) and curve-fitted (red curve) relations of exponential function y = a.e^bx^; chemometrics.

**Figure 4 ijms-24-16776-f004:**
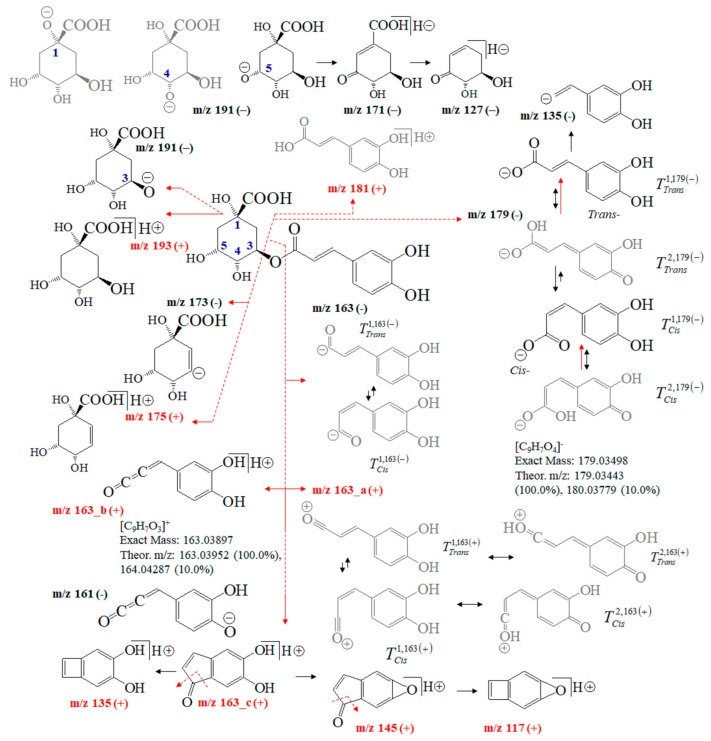
Chemical diagrams of chlorogenic acid and its product ions and isomers; theoretical *m*/*z* data and intensity ratio of the isotope shape of peaks; tautomer labeling, for example, T_cis_^2,179(−)^ denotes: Tautomeric form T^2^ in its *cis*-conformation of ion at *m*/*z* 179 and negative mass spectrometric polarity (−).

**Figure 8 ijms-24-16776-f008:**
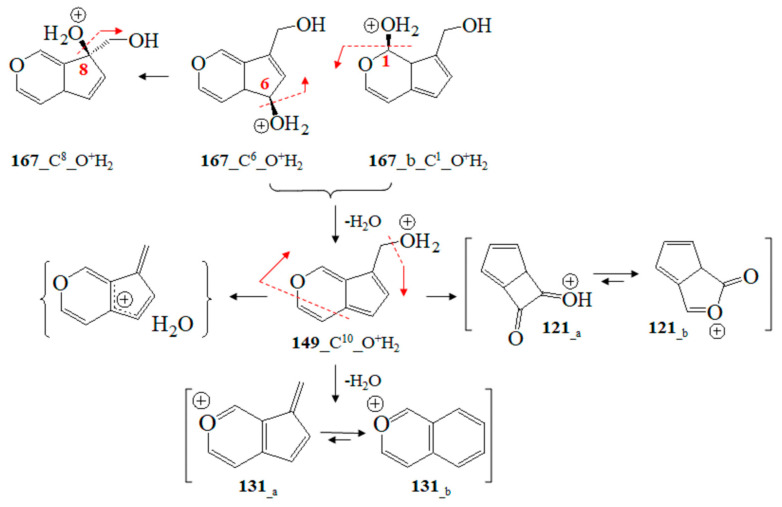
Chemical diagrams of GC-MS fragmentation ions at *m*/*z* 167, 149, 131, and 121 and charge transfer and rearrangement processes according to theoretical data.

**Figure 9 ijms-24-16776-f009:**
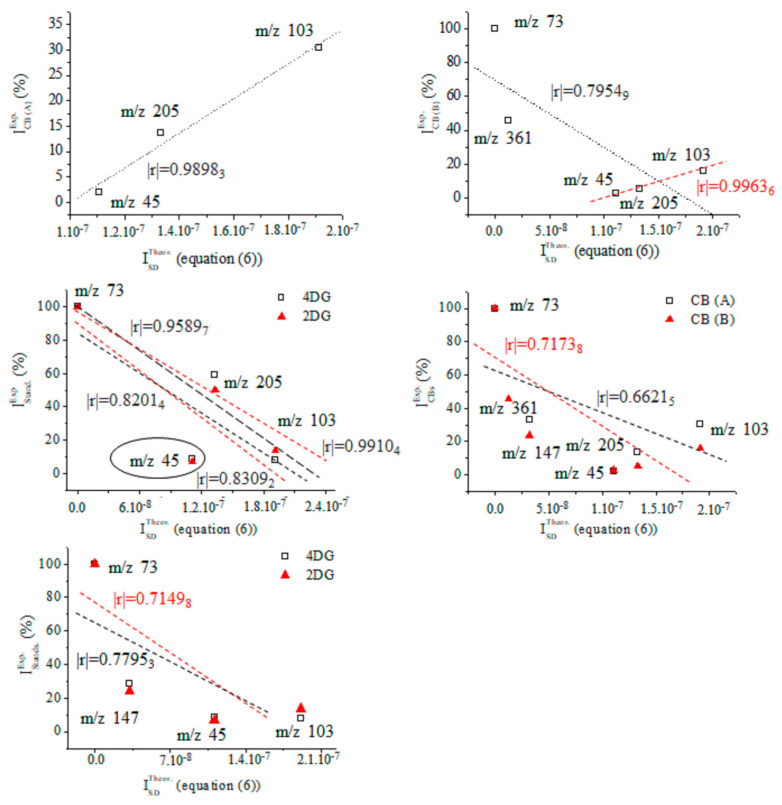
Chemometrics assessing relative intensity data on carbohydrates (A) and (B) as well as mass spectra of the standards of silylated 4- and 2-D-desoxy-glucose, and results from the theoretical intensity data provided by Equation (6).

**Table 1 ijms-24-16776-t001:** Metabolites identified in 70% CH_3_CH_2_OH extract of *Pyrus communis* L. microplants; RT denotes retention time.

N	RT; min	Connections
Organic acids		
1	9:37	Thioglycolic acid
2	10:23	Fumaric acid
3	10:24	Lactic acid
4	10:32	Acetic acid
5	10:39	Mercaptoacetic acid
6	11:34	Acetimidic acid
7	11:52	Allylacetic acid
8	12:28	Propanedioc acid
9	13:09	DL-malic (butanedioic) acid
10	13:19	Gluconic acid
11	14:26	Itaconic acid
12	14:30	Propanoic acid
13	14:58	Erythronic acid
14	14:59	Malonic acid
15	15:03	Succinic acid
16	15:09	Pentonic acid
17	16:33	Malic acid
18	16:56	Aspartic acid
19	16:59	Isocitric asid lactone
20	17:09	Monoamidoethylmalonic acid
21	17:51	Pentanoic acid
22	17:59	Erythro-pentonic acid
23	16:31	Succinic anhydride
24	18:06	Glutamic acid
25	18:12	Monoamidoethylmalonic acid
26	20:13	Schikimic acid
27	20:26	Citric acid
28	21:06	Quinic acid
29	21:18	Altronic acid
30	21:49	Acrylic acid
31	23:23	Coffeic acid
32	28:43	Galactaric acid
Phenolic compounds		
33	19:37	Benzoic acid
34	19:48	Cinnamic acid
35	21:05	Quinic acid
36	22:53	Ferulic acid
37	40:50	Chlorogenic acid
Fatty acids		
38	21:15	Hexadecanoic acid
Carbohydrates (CBs)		
39	17:02	Erythritol
40	18:19	Xylonic acid
41	18:36	D-(−)-Ribofuranose
42	19:22	D-Xylopyranose
43	19:26	D-Galactose
44	19:43	Gluconic acid, γ-lacton
45	19:43	L-(−)-Arabitol
46	19:43	Ribitol
47	20:11	D-(−)Tagatofuranose
48	20:36	D-(−)-Fructofuranose
49	20:51	D-(−)-Fructopyranose
50	20:51	1.5-Anhydroglucitol
51	20:53	Levoglucosan
52	21:01	Psicopyranose
53	21:13	Methylgalactoside
54	21:22	β-DL-arabinopyranose
55	21:17	Glyceryl-glycoside
56	21:31	Lactulose
57	21:32	Allopyranose
58	21:51	Inositol
59	22:03	Sorbitol
60	22:15	Allofuranose
61	22:17	β-D-Glucopyranose
62	22:26	Fructose
63	22:28	Talofuranose
64	23:13	D-Galacturonic acid
65	24:03	Hexopyranose
66	24:15	Lactose
67	30:03	Lactose
68	30:48	D-(+)-Turanose
69	31:02	Furanose
70	34:25	Dioxyfructose
71	35:55	Maltose
72	37:08	D-(+)-Cellobiose
Aminosugars		
73	19:22	N-acetil-glucosamin
Amino acids and their derivatives		
74	12:06	L-Leucine
75	12.53	L-Norvaline
76	14:48	Uracil-5-carboxylate decarboxylase
77	15:08	6-Azauracil
78	15:23	L-Treonin
79	16:48	L-Proline
80	17:20	Citrullin
Other connections		
81	12:51	Urea
82	13:43	Glycerol
83	13:50	Clycerol
84	16:00	2-Piperidone
85	17:41	Pirimidine
86	21:40	Octadecanamide
87	23:31	Myo-inositol
88	26:70	Glycerylglicoside
89	31:29	Aukubin

**Table 2 ijms-24-16776-t002:** Metabolites classes and the number of these classes identified in alcoholic extracts of pear microplants as a function of the period of time after magnetic-pulse treatment.

Metabolite	In Total	Number of Metabolites after Magnetic-Pulse Treatment	
		t ≥ 3 h	t ≥ 72 h
Organic acids	32	27	15
Carbohydrates	34	29	12
Amino acids and their derivatives	7	7	0
Phenolic compounds	5	3	3
Fatty acids	1	0	1
Aminosugars	1	1	0

**Table 3 ijms-24-16776-t003:** ANOVA data on datasets of variable intensity and *m*/*z* data on carbohydrates (A) and (B) (Figure 5 and Appendix A) (continued in Appendix A).

Intensity data on carbohydrates (A) and (B)					
Dataset	N	Mean	sd(yEr±)	se(yEr±)	
Data1_CBA	10	22.69459	30.22574	9.55822	
Data1_CBB	16	18.47876	24.44037	6.11009	
H_0_: The means of all selected datasets are equal					
H_1_: The means of one or more of the selected datasets are different					
Source	DoF	Sum of square	Mean square	F value	P value
Model	1	109.373548	109.373548	0.15277	0.69935
Error	24	17182.313	715.930471		
At the 0.05 level, the population means are not significantly different					
Means Comparison using the Bonferroni Test					
Dataset	Mean	Difference between means	Simultaneous Confidence Intervals		Significant at 0.05 Level
Data1_CBA	22.69459		Lower Limit	Upper Limit	
Data1_CBB	18.47876	4.21583	−18.04547	26.47712	No
Data on *m*/*z* of carbohydrates (A) and (B)					
Dataset	N	Mean	sd(yEr±)	se(yEr±)	
Data1_CBA	10	221.2	237.09623	74.97641	
Data1_CBB	16	201.875	97.17261	24.29315	
H_0_: The means of all selected datasets are equal					
H_1_: The means of one or more of the selected datasets are different					
Source	DoF	Sum of square	Mean square	F value	*p* value
Model	1	2298.18846	2298.18846	0.08517	0.77291
Error	24	647,569.350	26,982.0562		
At the 0.05 level, the population means are not significantly different					
Means Comparison using the Bonferroni Test					
Dataset	Mean	Difference between means	Simultaneous Confidence Intervals		Significant at 0.05 Level
Data1_CBA	221.2		Lower Limit	Upper Limit	
Data1_CBB	201.875	19.325	−117.3385	155.9885	No

**Table 4 ijms-24-16776-t004:** Experimental relative intensity data (r.i. (%)) of common product ions of carbohydrates (A) and (B) synthesized and isolated in *Pyrus communis* L. and GC-EI-MS/MS spectra of the standard samples of 4- (4DG) and 2-desoxy-D-glucose (2DG); theoretical intensity data provided by Equation (6) of 2D and 3D molecular structures of species of 4DG depicted in Figure 5.

*m*/*z*	I^Theor.^_SD_	r.i. [%]			
		CB (A)	CB (B)	4DG	2DG
45	1.11 × 10^−7^	2.00_6_	2.71_7_	8.39_7_	6.97
73	4.72 × 10^−12^	100.00	100.00	100.00	100.00
103	1.91 × 10^−7^	30.50_2_	15.83_8_	8.05_8_	13.69
147	3.22 × 10^−8^	32.75_6_	23.58_7_	28.43_4_	23.89
205	1.33 × 10^−7^	13.58_4_	5.31	-	-
361	1.24 × 10^−8^	-	45.68	-	-

**Table 5 ijms-24-16776-t005:** Technical characteristics of AMIS-8.

Frequency range of the scanning of unidirectional and multidirectional magnetic induction pulses (Hz)		0.10–150.00
Increase time of magnetic induction pulses (ms)		≤0.2
Decrease time of magnetic induction pulses (ms)		≤3.0
Duration of magnetic induction pulses at the 0.5 level of maximum amplitude (ms)		≤1.5
Magnetic induction pulse amplitude at frequency 20 Hz and the distance from the inductor surface PSI-1 (10–400 mm) along its central axis:	1. Without light pulse source (mTI)	11.0–0.25
	2. With light pulse source (mTI)	9.0–0.20
Exposure time (s)		10–1800
Working area of inductor PSI-1 (cm^2^)		1800
Power supply source—220 V, 50 Hz		
Continuous operation time (h)		0
Maximum power consumption for lower and upper values of magnetic induction pulse frequency range (V–A)		20 and 120

## Data Availability

The data presented in this study are available on request from the corresponding author.

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
