# Peer review of "Mass Spectrometric Identification of Metabolites after Magnetic-Pulse Treatment of Infected Pyrus communis L. Microplants"

_ijms, 2023, doi:10.3390/ijms242316776_

Round 1

Reviewer 1 Report

Comments and Suggestions for Authors 1) The conclusion highlights the identification of 89 metabolites from the virus infected micro-plants, but says nothing about the effect of magneto-therapy on plant recovery, a topic that was indicated on line 43. That chlorogenic acid enhances resistance is stated on line 53, but that is not the same as recovery. This discrepancy is emphasized by the control employed being virus infected microplants with no MF exposure according to the captions of Figures 1 and 2 and that the discussion section takes up the advantages of MPT but never addresses the degree of recovery after treatment by comparison with metabolites from healthy plants or virus diagnosis by ELISA mentioned in 4.2.   Since one of the aims of IJMS is "strong emphasis on molecular biology", and not just the analytical techniques needed to accomplish the goal, a clarification is needed in the introduction as to what the study intends to contribute to the plant-specific MPT field not already covered in references 6-8.   The claim on line 99 that there was a 14.5 increase in chlorogenic acid after 10 Hz treatment after 72 h relative to 0 Hz doesn't seem to be supported in Figure 1 where the concentration ratio is 1.2/0.18=6.7.   2)  That it is the analytical aspect which dominates the study is further emphasized in section 2 where only 2.1 deals with metabolites evolving from the microplants and the other three sections with fragmentation patterns.  It is pretty much impossible to determine from the section what new is being contributed to fragmentation patterns. Line 232 says that reliable MS protocols for quantifying chlorogenic acid in foods aren't relevant for the study. Line 401 claims agreement with previous data. What is needed in section 3 is a summary of accomplishments of the fragmentation work, such as distinguishing between positional stereo-isomers on line 390 and new metabolites on line 25, as compared to state of the art in the area. If the therapeutic aspect is to be pursued, then how the above MS accomplishments contributed to following recovery back toward healthy status, should be included in the discussion.    3)  Discussing references dealing with how magnetism affects Pyrus communis L. might strengthen the plant development topic, such as T Cebulak et al.,  Effect of abiotic stress factors on polyphenolic content in the skin and flesh of pear by UPLC‑PDA‑Q/TOF‑MS, European Food Research and Technology 245 (2019) 245 2715-2725, and A El-Nasr et al., Effect of Magnetite Nanoparticles (Fe3O4) as Nutritive Supplement on Pear Saplings, Middle East Journal of Applied Sciences 5 (2015) 777-785.   4) Experimental procedure with respect to silyation in 4.1 and 4.2 needs clarification. Section 4.1 says samples for 3 and 72 h analysis were extracted using 70% ethanol and silylated using BSATF. The procedure in 4.2 was extraction in methanol with silylation mentioned on line 484. Which samples and what analyses are associated with each procedure?   5) The long paragraphs need to be broken down into smaller ones. Section 3, for example, consists of only one.   6) Most of the equations in Appendix A are unreadable.   Comments on the Quality of English Language

Reading the test moves along although when facing the long paragraphs there is a feeling of do I really one to jump into this thing. There are often clumsy phrases, such as on line 15: There are scarce data on this thematic reported, which should read something like: There are few studies about this particular topic. Reading the text aloud can catch some of them, but it's pretty much impossible to correct all of them. 

Author Response

RESPONSE TO REVIEWER 1

Comments and Suggestions for Authors

REVIEWER 1: 1) The conclusion highlights the identification of 89 metabolites from the virus infected micro-plants, but says nothing about the effect of magneto-therapy on plant recovery, a topic that was indicated on line 43. That chlorogenic acid enhances resistance is stated on line 53, but that is not the same as recovery. This discrepancy is emphasized by the control employed being virus infected microplants with no MF exposure according to the captions of Figures 1 and 2 and that the discussion section takes up the advantages of MPT but never addresses the degree of recovery after treatment by comparison with metabolites from healthy plants or virus diagnosis by ELISA mentioned in 4.2.   Since one of the aims of IJMS is "strong emphasis on molecular biology", and not just the analytical techniques needed to accomplish the goal, a clarification is needed in the introduction as to what the study intends to contribute to the plant-specific MPT field not already covered in references 6-8.   The claim on line 99 that there was a 14.5 increase in chlorogenic acid after 10 Hz treatment after 72 h relative to 0 Hz doesn't seem to be supported in Figure 1 where the concentration ratio is 1.2/0.18=6.7.  

AUTHORS: The authors would like to thank for valuable Reviewer’s recommendations and remarks. According to shown above remarks, there is added recovery value (4.79) obtained according equation (1) of new cited reference [9]. In addition, there are shown chemometric parameters of chromatographic measurements (see new Figure A1.)

The chemometric determination of curve-areas of chromatographic data yields to 8.75-fold increasing in chlorogenic acid. The value is corrected in the revised version of the text.    

[9] Burns D.; Danzer K.; Townshend A. Use of the terms “recovery” and “apparent recovery” in analytical procedures. Pure Appl. Chem. 2002, 74, 22012205.

REVIEWER 1: 2)  That it is the analytical aspect which dominates the study is further emphasized in section 2 where only 2.1 deals with metabolites evolving from the microplants and the other three sections with fragmentation patterns.  It is pretty much impossible to determine from the section what new is being contributed to fragmentation patterns. Line 232 says that reliable MS protocols for quantifying chlorogenic acid in foods aren't relevant for the study. Line 401 claims agreement with previous data. What is needed in section 3 is a summary of accomplishments of the fragmentation work, such as distinguishing between positional stereo-isomers on line 390 and new metabolites on line 25, as compared to state of the art in the area. If the therapeutic aspect is to be pursued, then how the above MS accomplishments contributed to following recovery back toward healthy status, should be included in the discussion.   

AUTHORS: The content of the paper is designed for Special Issue [https://www.mdpi.com/journal/ijms/special_issues/P4R5D7UIZM], devoted to apply ‚mass spectrum‘ to field of metabolomics and udrestand comprehensively biochemical reactions in vitro and in vivo upon effect of magnetic field treatment. Due to this reason content focuses, mainly, on capability of mass spectrometry as robust analytical method for identification, annotation, and structural analysis of metabolites, despite, the fact that the major scope of the Journal focuses on biochemistry. Owing to the fact that, there is proven benefical biological function of aucubin, and there has been achieved perturbation of metabolism of chlorogenic acid upon external magnetic field treatment the focus of the old version of the manuscript was on mass spectrometric data on these species. However, from perspective of novelty, the fragmentation pattern of clorogenic acid is comprehensively examined by many authors. In fact, there is an in-depth analysis of fragmentation paths of the latter analyte (please, consider Table 1 in reference [59].) Particularly, the latter study looks at isomerism of clorogfenic acid. Also, MS data on stereoisomers of clorogenic acid have been in-depth discussed, in fact, in all cited studies by authors such as, for example [47-64]. Moreover, these are not fully representative references to the issue, because of our work does not represent exhaustive review-article. We only confirm known data on chlorogenic acid and aucubin via independent innovative mass spectrometric method based on stochastic dynamics (please, consider equations (1)-(6) and new sub-section 4.5.1.) The latter task aims at proving their validity and their adequate application to annotate unknown species, having comparable chemometric parameters using routine searching algorithms for data processing of mass spectrometric variables of metabolites and their assignment according to library parameters of mass spectra. For instance, these are carbohydrates. Thus, from perspective of novelty of fragmentation pattern there should be highlighted our analysis of salicylated carbohydrates, which mass spectrometric based annotation does not respresent trivial task due to comparable method performances. Please, see new sub section 2.4. Mass spectrometric data on silylated carbohydrate monomers and new Figures A2–A16. There are shown experimental chromatographic and mass spectrometric data on silylated carbohydrates assigned comprehensively via two independent theoretical methods, thus fitting content of the study to thematic issue of the Journal.

[59] Xie C.; Yu K.; Zhong D.; Yuan, T.; Ye, F.; Jarrell J.; Millar A.; Chen, X. Investigation of isomeric transformations of chlorogenic acid in buffers and biological matrixes by ultraperformance liquid chromatography coupled with hybrid quadrupole/ion mobility/orthogonal acceleration time-of-flight mass spectrometry. J. Agric. Food Chem. 2011, 59, 11078–11087.

REVIEWER 1: 3)  Discussing references dealing with how magnetism affects Pyrus communis L. might strengthen the plant development topic, such as T Cebulak et al.,  Effect of abiotic stress factors on polyphenolic content in the skin and flesh of pear by UPLC‑PDA‑Q/TOF‑MS, European Food Research and Technology 245 (2019) 245 2715-2725, and A El-Nasr et al., Effect of Magnetite Nanoparticles (Fe3O4) as Nutritive Supplement on Pear Saplings, Middle East Journal of Applied Sciences 5 (2015) 777-785.  

AUTHORS: There is presented new section ‚Discussion‘ of revised paper. It reflects Reviewer’s remarks and aforementioned contributions to other authors. Please, consider highlighted text of section 3. Discussion and new cited works [85–96]. The revised section ‚Discussion‘, we hope illustrates more clearly for Editors, Reviewers and potential Readers the impact and novelty of our study accounting not only for scope of special issue and mass spectrometric data, but also biochemical aspects of our results from metabolomics.  

REVIEWER 1: 4) Experimental procedure with respect to silyation in 4.1 and 4.2 needs clarification. Section 4.1 says samples for 3 and 72 h analysis were extracted using 70% ethanol and silylated using BSATF. The procedure in 4.2 was extraction in methanol with silylation mentioned on line 484. Which samples and what analyses are associated with each procedure?  

AUTHORS: Section ‚Experimental‘ of revised text is completely re-written. It details on synthesis of sylilated carbohydrates; experimental design of magnetic measurements of plants together with experimental parameters: detail on chromatographic measurements and annotation; advantages and limitation of GC-MS and LC-MS methods which are often used to purposes of metabolomics; brief description of strochastic dynamic theory, quantum chemical computations, and chemometrics, respectively. Relevant reference section is added, as well.   

REVIEWER 1: 5) The long paragraphs need to be broken down into smaller ones. Section 3, for example, consists of only one.  

AUTHORS: The text of the manuscript is fully re-written.

REVIEWER 1: 6) Most of the equations in Appendix A are unreadable.  

AUTHORS: The equation are written via text editor. Perhaps, due to conversion from doc to pdf format most of them are, thus, unreadable for the Reviewer. In order to exclude from such disadvantage, the revised text contains same formulas shown, however as a single embedded into the text figure.  

REVIEWER 1: Comments on the Quality of English Language

Reading the test moves along although when facing the long paragraphs there is a feeling of do I really one to jump into this thing. There are often clumsy phrases, such as on line 15: There are scarce data on this thematic reported, which should read something like: There are few studies about this particular topic. Reading the text aloud can catch some of them, but it's pretty much impossible to correct all of them. 

AUTHORS: The English is revised. The text of the manuscript, is in fact, rew

RESPONSE TO REVIEWER 1

Comments and Suggestions for Authors

REVIEWER 1: 1) The conclusion highlights the identification of 89 metabolites from the virus infected micro-plants, but says nothing about the effect of magneto-therapy on plant recovery, a topic that was indicated on line 43. That chlorogenic acid enhances resistance is stated on line 53, but that is not the same as recovery. This discrepancy is emphasized by the control employed being virus infected microplants with no MF exposure according to the captions of Figures 1 and 2 and that the discussion section takes up the advantages of MPT but never addresses the degree of recovery after treatment by comparison with metabolites from healthy plants or virus diagnosis by ELISA mentioned in 4.2.   Since one of the aims of IJMS is "strong emphasis on molecular biology", and not just the analytical techniques needed to accomplish the goal, a clarification is needed in the introduction as to what the study intends to contribute to the plant-specific MPT field not already covered in references 6-8.   The claim on line 99 that there was a 14.5 increase in chlorogenic acid after 10 Hz treatment after 72 h relative to 0 Hz doesn't seem to be supported in Figure 1 where the concentration ratio is 1.2/0.18=6.7.  

AUTHORS: The authors would like to thank for valuable Reviewer’s recommendations and remarks. According to shown above remarks, there is added recovery value (4.79) obtained according equation (1) of new cited reference [9]. In addition, there are shown chemometric parameters of chromatographic measurements (see new Figure A1.)

The chemometric determination of curve-areas of chromatographic data yields to 8.75-fold increasing in chlorogenic acid. The value is corrected in the revised version of the text.    

[9] Burns D.; Danzer K.; Townshend A. Use of the terms “recovery” and “apparent recovery” in analytical procedures. Pure Appl. Chem. 2002, 74, 22012205.

REVIEWER 1: 2)  That it is the analytical aspect which dominates the study is further emphasized in section 2 where only 2.1 deals with metabolites evolving from the microplants and the other three sections with fragmentation patterns.  It is pretty much impossible to determine from the section what new is being contributed to fragmentation patterns. Line 232 says that reliable MS protocols for quantifying chlorogenic acid in foods aren't relevant for the study. Line 401 claims agreement with previous data. What is needed in section 3 is a summary of accomplishments of the fragmentation work, such as distinguishing between positional stereo-isomers on line 390 and new metabolites on line 25, as compared to state of the art in the area. If the therapeutic aspect is to be pursued, then how the above MS accomplishments contributed to following recovery back toward healthy status, should be included in the discussion.   

AUTHORS: The content of the paper is designed for Special Issue [https://www.mdpi.com/journal/ijms/special_issues/P4R5D7UIZM], devoted to apply ‚mass spectrum‘ to field of metabolomics and udrestand comprehensively biochemical reactions in vitro and in vivo upon effect of magnetic field treatment. Due to this reason content focuses, mainly, on capability of mass spectrometry as robust analytical method for identification, annotation, and structural analysis of metabolites, despite, the fact that the major scope of the Journal focuses on biochemistry. Owing to the fact that, there is proven benefical biological function of aucubin, and there has been achieved perturbation of metabolism of chlorogenic acid upon external magnetic field treatment the focus of the old version of the manuscript was on mass spectrometric data on these species. However, from perspective of novelty, the fragmentation pattern of clorogenic acid is comprehensively examined by many authors. In fact, there is an in-depth analysis of fragmentation paths of the latter analyte (please, consider Table 1 in reference [59].) Particularly, the latter study looks at isomerism of clorogfenic acid. Also, MS data on stereoisomers of clorogenic acid have been in-depth discussed, in fact, in all cited studies by authors such as, for example [47-64]. Moreover, these are not fully representative references to the issue, because of our work does not represent exhaustive review-article. We only confirm known data on chlorogenic acid and aucubin via independent innovative mass spectrometric method based on stochastic dynamics (please, consider equations (1)-(6) and new sub-section 4.5.1.) The latter task aims at proving their validity and their adequate application to annotate unknown species, having comparable chemometric parameters using routine searching algorithms for data processing of mass spectrometric variables of metabolites and their assignment according to library parameters of mass spectra. For instance, these are carbohydrates. Thus, from perspective of novelty of fragmentation pattern there should be highlighted our analysis of salicylated carbohydrates, which mass spectrometric based annotation does not respresent trivial task due to comparable method performances. Please, see new sub section 2.4. Mass spectrometric data on silylated carbohydrate monomers and new Figures A2–A16. There are shown experimental chromatographic and mass spectrometric data on silylated carbohydrates assigned comprehensively via two independent theoretical methods, thus fitting content of the study to thematic issue of the Journal.

[59] Xie C.; Yu K.; Zhong D.; Yuan, T.; Ye, F.; Jarrell J.; Millar A.; Chen, X. Investigation of isomeric transformations of chlorogenic acid in buffers and biological matrixes by ultraperformance liquid chromatography coupled with hybrid quadrupole/ion mobility/orthogonal acceleration time-of-flight mass spectrometry. J. Agric. Food Chem. 2011, 59, 11078–11087.

REVIEWER 1: 3)  Discussing references dealing with how magnetism affects Pyrus communis L. might strengthen the plant development topic, such as T Cebulak et al.,  Effect of abiotic stress factors on polyphenolic content in the skin and flesh of pear by UPLC‑PDA‑Q/TOF‑MS, European Food Research and Technology 245 (2019) 245 2715-2725, and A El-Nasr et al., Effect of Magnetite Nanoparticles (Fe3O4) as Nutritive Supplement on Pear Saplings, Middle East Journal of Applied Sciences 5 (2015) 777-785.  

AUTHORS: There is presented new section ‚Discussion‘ of revised paper. It reflects Reviewer’s remarks and aforementioned contributions to other authors. Please, consider highlighted text of section 3. Discussion and new cited works [85–96]. The revised section ‚Discussion‘, we hope illustrates more clearly for Editors, Reviewers and potential Readers the impact and novelty of our study accounting not only for scope of special issue and mass spectrometric data, but also biochemical aspects of our results from metabolomics.  

REVIEWER 1: 4) Experimental procedure with respect to silyation in 4.1 and 4.2 needs clarification. Section 4.1 says samples for 3 and 72 h analysis were extracted using 70% ethanol and silylated using BSATF. The procedure in 4.2 was extraction in methanol with silylation mentioned on line 484. Which samples and what analyses are associated with each procedure?  

AUTHORS: Section ‚Experimental‘ of revised text is completely re-written. It details on synthesis of sylilated carbohydrates; experimental design of magnetic measurements of plants together with experimental parameters: detail on chromatographic measurements and annotation; advantages and limitation of GC-MS and LC-MS methods which are often used to purposes of metabolomics; brief description of strochastic dynamic theory, quantum chemical computations, and chemometrics, respectively. Relevant reference section is added, as well.   

REVIEWER 1: 5) The long paragraphs need to be broken down into smaller ones. Section 3, for example, consists of only one.  

AUTHORS: The text of the manuscript is fully re-written.

REVIEWER 1: 6) Most of the equations in Appendix A are unreadable.  

AUTHORS: The equation are written via text editor. Perhaps, due to conversion from doc to pdf format most of them are, thus, unreadable for the Reviewer. In order to exclude from such disadvantage, the revised text contains same formulas shown, however as a single embedded into the text figure.  

REVIEWER 1: Comments on the Quality of English Language

Reading the test moves along although when facing the long paragraphs there is a feeling of do I really one to jump into this thing. There are often clumsy phrases, such as on line 15: There are scarce data on this thematic reported, which should read something like: There are few studies about this particular topic. Reading the text aloud can catch some of them, but it's pretty much impossible to correct all of them. 

AUTHORS: The English is revised. The text of the manuscript, is in fact, rewritten, due to extended revision of almost all sections.

RESPONSE TO REVIEWER 1

Comments and Suggestions for Authors

REVIEWER 1: 1) The conclusion highlights the identification of 89 metabolites from the virus infected micro-plants, but says nothing about the effect of magneto-therapy on plant recovery, a topic that was indicated on line 43. That chlorogenic acid enhances resistance is stated on line 53, but that is not the same as recovery. This discrepancy is emphasized by the control employed being virus infected microplants with no MF exposure according to the captions of Figures 1 and 2 and that the discussion section takes up the advantages of MPT but never addresses the degree of recovery after treatment by comparison with metabolites from healthy plants or virus diagnosis by ELISA mentioned in 4.2.   Since one of the aims of IJMS is "strong emphasis on molecular biology", and not just the analytical techniques needed to accomplish the goal, a clarification is needed in the introduction as to what the study intends to contribute to the plant-specific MPT field not already covered in references 6-8.   The claim on line 99 that there was a 14.5 increase in chlorogenic acid after 10 Hz treatment after 72 h relative to 0 Hz doesn't seem to be supported in Figure 1 where the concentration ratio is 1.2/0.18=6.7.  

AUTHORS: The authors would like to thank for valuable Reviewer’s recommendations and remarks. According to shown above remarks, there is added recovery value (4.79) obtained according equation (1) of new cited reference [9]. In addition, there are shown chemometric parameters of chromatographic measurements (see new Figure A1.)

The chemometric determination of curve-areas of chromatographic data yields to 8.75-fold increasing in chlorogenic acid. The value is corrected in the revised version of the text.    

[9] Burns D.; Danzer K.; Townshend A. Use of the terms “recovery” and “apparent recovery” in analytical procedures. Pure Appl. Chem. 2002, 74, 22012205.

REVIEWER 1: 2)  That it is the analytical aspect which dominates the study is further emphasized in section 2 where only 2.1 deals with metabolites evolving from the microplants and the other three sections with fragmentation patterns.  It is pretty much impossible to determine from the section what new is being contributed to fragmentation patterns. Line 232 says that reliable MS protocols for quantifying chlorogenic acid in foods aren't relevant for the study. Line 401 claims agreement with previous data. What is needed in section 3 is a summary of accomplishments of the fragmentation work, such as distinguishing between positional stereo-isomers on line 390 and new metabolites on line 25, as compared to state of the art in the area. If the therapeutic aspect is to be pursued, then how the above MS accomplishments contributed to following recovery back toward healthy status, should be included in the discussion.   

AUTHORS: The content of the paper is designed for Special Issue [https://www.mdpi.com/journal/ijms/special_issues/P4R5D7UIZM], devoted to apply ‚mass spectrum‘ to field of metabolomics and udrestand comprehensively biochemical reactions in vitro and in vivo upon effect of magnetic field treatment. Due to this reason content focuses, mainly, on capability of mass spectrometry as robust analytical method for identification, annotation, and structural analysis of metabolites, despite, the fact that the major scope of the Journal focuses on biochemistry. Owing to the fact that, there is proven benefical biological function of aucubin, and there has been achieved perturbation of metabolism of chlorogenic acid upon external magnetic field treatment the focus of the old version of the manuscript was on mass spectrometric data on these species. However, from perspective of novelty, the fragmentation pattern of clorogenic acid is comprehensively examined by many authors. In fact, there is an in-depth analysis of fragmentation paths of the latter analyte (please, consider Table 1 in reference [59].) Particularly, the latter study looks at isomerism of clorogfenic acid. Also, MS data on stereoisomers of clorogenic acid have been in-depth discussed, in fact, in all cited studies by authors such as, for example [47-64]. Moreover, these are not fully representative references to the issue, because of our work does not represent exhaustive review-article. We only confirm known data on chlorogenic acid and aucubin via independent innovative mass spectrometric method based on stochastic dynamics (please, consider equations (1)-(6) and new sub-section 4.5.1.) The latter task aims at proving their validity and their adequate application to annotate unknown species, having comparable chemometric parameters using routine searching algorithms for data processing of mass spectrometric variables of metabolites and their assignment according to library parameters of mass spectra. For instance, these are carbohydrates. Thus, from perspective of novelty of fragmentation pattern there should be highlighted our analysis of salicylated carbohydrates, which mass spectrometric based annotation does not respresent trivial task due to comparable method performances. Please, see new sub section 2.4. Mass spectrometric data on silylated carbohydrate monomers and new Figures A2–A16. There are shown experimental chromatographic and mass spectrometric data on silylated carbohydrates assigned comprehensively via two independent theoretical methods, thus fitting content of the study to thematic issue of the Journal.

[59] Xie C.; Yu K.; Zhong D.; Yuan, T.; Ye, F.; Jarrell J.; Millar A.; Chen, X. Investigation of isomeric transformations of chlorogenic acid in buffers and biological matrixes by ultraperformance liquid chromatography coupled with hybrid quadrupole/ion mobility/orthogonal acceleration time-of-flight mass spectrometry. J. Agric. Food Chem. 2011, 59, 11078–11087.

REVIEWER 1: 3)  Discussing references dealing with how magnetism affects Pyrus communis L. might strengthen the plant development topic, such as T Cebulak et al.,  Effect of abiotic stress factors on polyphenolic content in the skin and flesh of pear by UPLC‑PDA‑Q/TOF‑MS, European Food Research and Technology 245 (2019) 245 2715-2725, and A El-Nasr et al., Effect of Magnetite Nanoparticles (Fe3O4) as Nutritive Supplement on Pear Saplings, Middle East Journal of Applied Sciences 5 (2015) 777-785.  

AUTHORS: There is presented new section ‚Discussion‘ of revised paper. It reflects Reviewer’s remarks and aforementioned contributions to other authors. Please, consider highlighted text of section 3. Discussion and new cited works [85–96]. The revised section ‚Discussion‘, we hope illustrates more clearly for Editors, Reviewers and potential Readers the impact and novelty of our study accounting not only for scope of special issue and mass spectrometric data, but also biochemical aspects of our results from metabolomics.  

REVIEWER 1: 4) Experimental procedure with respect to silyation in 4.1 and 4.2 needs clarification. Section 4.1 says samples for 3 and 72 h analysis were extracted using 70% ethanol and silylated using BSATF. The procedure in 4.2 was extraction in methanol with silylation mentioned on line 484. Which samples and what analyses are associated with each procedure?  

AUTHORS: Section ‚Experimental‘ of revised text is completely re-written. It details on synthesis of sylilated carbohydrates; experimental design of magnetic measurements of plants together with experimental parameters: detail on chromatographic measurements and annotation; advantages and limitation of GC-MS and LC-MS methods which are often used to purposes of metabolomics; brief description of strochastic dynamic theory, quantum chemical computations, and chemometrics, respectively. Relevant reference section is added, as well.   

REVIEWER 1: 5) The long paragraphs need to be broken down into smaller ones. Section 3, for example, consists of only one.  

AUTHORS: The text of the manuscript is fully re-written.

REVIEWER 1: 6) Most of the equations in Appendix A are unreadable.  

AUTHORS: The equation are written via text editor. Perhaps, due to conversion from doc to pdf format most of them are, thus, unreadable for the Reviewer. In order to exclude from such disadvantage, the revised text contains same formulas shown, however as a single embedded into the text figure.  

REVIEWER 1: Comments on the Quality of English Language

Reading the test moves along although when facing the long paragraphs there is a feeling of do I really one to jump into this thing. There are often clumsy phrases, such as on line 15: There are scarce data on this thematic reported, which should read something like: There are few studies about this particular topic. Reading the text aloud can catch some of them, but it's pretty much impossible to correct all of them. 

AUTHORS: The English is revised. The text of the manuscript, is in fact, rewritten, due to extended revision of almost all sections.

RESPONSE TO REVIEWER 1

Comments and Suggestions for Authors

REVIEWER 1: 1) The conclusion highlights the identification of 89 metabolites from the virus infected micro-plants, but says nothing about the effect of magneto-therapy on plant recovery, a topic that was indicated on line 43. That chlorogenic acid enhances resistance is stated on line 53, but that is not the same as recovery. This discrepancy is emphasized by the control employed being virus infected microplants with no MF exposure according to the captions of Figures 1 and 2 and that the discussion section takes up the advantages of MPT but never addresses the degree of recovery after treatment by comparison with metabolites from healthy plants or virus diagnosis by ELISA mentioned in 4.2.   Since one of the aims of IJMS is "strong emphasis on molecular biology", and not just the analytical techniques needed to accomplish the goal, a clarification is needed in the introduction as to what the study intends to contribute to the plant-specific MPT field not already covered in references 6-8.   The claim on line 99 that there was a 14.5 increase in chlorogenic acid after 10 Hz treatment after 72 h relative to 0 Hz doesn't seem to be supported in Figure 1 where the concentration ratio is 1.2/0.18=6.7.  

AUTHORS: The authors would like to thank for valuable Reviewer’s recommendations and remarks. According to shown above remarks, there is added recovery value (4.79) obtained according equation (1) of new cited reference [9]. In addition, there are shown chemometric parameters of chromatographic measurements (see new Figure A1.)

The chemometric determination of curve-areas of chromatographic data yields to 8.75-fold increasing in chlorogenic acid. The value is corrected in the revised version of the text.    

[9] Burns D.; Danzer K.; Townshend A. Use of the terms “recovery” and “apparent recovery” in analytical procedures. Pure Appl. Chem. 2002, 74, 22012205.

REVIEWER 1: 2)  That it is the analytical aspect which dominates the study is further emphasized in section 2 where only 2.1 deals with metabolites evolving from the microplants and the other three sections with fragmentation patterns.  It is pretty much impossible to determine from the section what new is being contributed to fragmentation patterns. Line 232 says that reliable MS protocols for quantifying chlorogenic acid in foods aren't relevant for the study. Line 401 claims agreement with previous data. What is needed in section 3 is a summary of accomplishments of the fragmentation work, such as distinguishing between positional stereo-isomers on line 390 and new metabolites on line 25, as compared to state of the art in the area. If the therapeutic aspect is to be pursued, then how the above MS accomplishments contributed to following recovery back toward healthy status, should be included in the discussion.   

AUTHORS: The content of the paper is designed for Special Issue [https://www.mdpi.com/journal/ijms/special_issues/P4R5D7UIZM], devoted to apply ‚mass spectrum‘ to field of metabolomics and udrestand comprehensively biochemical reactions in vitro and in vivo upon effect of magnetic field treatment. Due to this reason content focuses, mainly, on capability of mass spectrometry as robust analytical method for identification, annotation, and structural analysis of metabolites, despite, the fact that the major scope of the Journal focuses on biochemistry. Owing to the fact that, there is proven benefical biological function of aucubin, and there has been achieved perturbation of metabolism of chlorogenic acid upon external magnetic field treatment the focus of the old version of the manuscript was on mass spectrometric data on these species. However, from perspective of novelty, the fragmentation pattern of clorogenic acid is comprehensively examined by many authors. In fact, there is an in-depth analysis of fragmentation paths of the latter analyte (please, consider Table 1 in reference [59].) Particularly, the latter study looks at isomerism of clorogfenic acid. Also, MS data on stereoisomers of clorogenic acid have been in-depth discussed, in fact, in all cited studies by authors such as, for example [47-64]. Moreover, these are not fully representative references to the issue, because of our work does not represent exhaustive review-article. We only confirm known data on chlorogenic acid and aucubin via independent innovative mass spectrometric method based on stochastic dynamics (please, consider equations (1)-(6) and new sub-section 4.5.1.) The latter task aims at proving their validity and their adequate application to annotate unknown species, having comparable chemometric parameters using routine searching algorithms for data processing of mass spectrometric variables of metabolites and their assignment according to library parameters of mass spectra. For instance, these are carbohydrates. Thus, from perspective of novelty of fragmentation pattern there should be highlighted our analysis of salicylated carbohydrates, which mass spectrometric based annotation does not respresent trivial task due to comparable method performances. Please, see new sub section 2.4. Mass spectrometric data on silylated carbohydrate monomers and new Figures A2–A16. There are shown experimental chromatographic and mass spectrometric data on silylated carbohydrates assigned comprehensively via two independent theoretical methods, thus fitting content of the study to thematic issue of the Journal.

[59] Xie C.; Yu K.; Zhong D.; Yuan, T.; Ye, F.; Jarrell J.; Millar A.; Chen, X. Investigation of isomeric transformations of chlorogenic acid in buffers and biological matrixes by ultraperformance liquid chromatography coupled with hybrid quadrupole/ion mobility/orthogonal acceleration time-of-flight mass spectrometry. J. Agric. Food Chem. 2011, 59, 11078–11087.

REVIEWER 1: 3)  Discussing references dealing with how magnetism affects Pyrus communis L. might strengthen the plant development topic, such as T Cebulak et al.,  Effect of abiotic stress factors on polyphenolic content in the skin and flesh of pear by UPLC‑PDA‑Q/TOF‑MS, European Food Research and Technology 245 (2019) 245 2715-2725, and A El-Nasr et al., Effect of Magnetite Nanoparticles (Fe3O4) as Nutritive Supplement on Pear Saplings, Middle East Journal of Applied Sciences 5 (2015) 777-785.  

AUTHORS: There is presented new section ‚Discussion‘ of revised paper. It reflects Reviewer’s remarks and aforementioned contributions to other authors. Please, consider highlighted text of section 3. Discussion and new cited works [85–96]. The revised section ‚Discussion‘, we hope illustrates more clearly for Editors, Reviewers and potential Readers the impact and novelty of our study accounting not only for scope of special issue and mass spectrometric data, but also biochemical aspects of our results from metabolomics.  

REVIEWER 1: 4) Experimental procedure with respect to silyation in 4.1 and 4.2 needs clarification. Section 4.1 says samples for 3 and 72 h analysis were extracted using 70% ethanol and silylated using BSATF. The procedure in 4.2 was extraction in methanol with silylation mentioned on line 484. Which samples and what analyses are associated with each procedure?  

AUTHORS: Section ‚Experimental‘ of revised text is completely re-written. It details on synthesis of sylilated carbohydrates; experimental design of magnetic measurements of plants together with experimental parameters: detail on chromatographic measurements and annotation; advantages and limitation of GC-MS and LC-MS methods which are often used to purposes of metabolomics; brief description of strochastic dynamic theory, quantum chemical computations, and chemometrics, respectively. Relevant reference section is added, as well.   

REVIEWER 1: 5) The long paragraphs need to be broken down into smaller ones. Section 3, for example, consists of only one.  

AUTHORS: The text of the manuscript is fully re-written.

REVIEWER 1: 6) Most of the equations in Appendix A are unreadable.  

AUTHORS: The equation are written via text editor. Perhaps, due to conversion from doc to pdf format most of them are, thus, unreadable for the Reviewer. In order to exclude from such disadvantage, the revised text contains same formulas shown, however as a single embedded into the text figure.  

REVIEWER 1: Comments on the Quality of English Language

Reading the test moves along although when facing the long paragraphs there is a feeling of do I really one to jump into this thing. There are often clumsy phrases, such as on line 15: There are scarce data on this thematic reported, which should read something like: There are few studies about this particular topic. Reading the text aloud can catch some of them, but it's pretty much impossible to correct all of them. 

AUTHORS: The English is revised. The text of the manuscript, is in fact, rewritten, due to extended revision of almost all sections.

RESPONSE TO REVIEWER 1

Comments and Suggestions for Authors

REVIEWER 1: 1) The conclusion highlights the identification of 89 metabolites from the virus infected micro-plants, but says nothing about the effect of magneto-therapy on plant recovery, a topic that was indicated on line 43. That chlorogenic acid enhances resistance is stated on line 53, but that is not the same as recovery. This discrepancy is emphasized by the control employed being virus infected microplants with no MF exposure according to the captions of Figures 1 and 2 and that the discussion section takes up the advantages of MPT but never addresses the degree of recovery after treatment by comparison with metabolites from healthy plants or virus diagnosis by ELISA mentioned in 4.2.   Since one of the aims of IJMS is "strong emphasis on molecular biology", and not just the analytical techniques needed to accomplish the goal, a clarification is needed in the introduction as to what the study intends to contribute to the plant-specific MPT field not already covered in references 6-8.   The claim on line 99 that there was a 14.5 increase in chlorogenic acid after 10 Hz treatment after 72 h relative to 0 Hz doesn't seem to be supported in Figure 1 where the concentration ratio is 1.2/0.18=6.7.  

AUTHORS: The authors would like to thank for valuable Reviewer’s recommendations and remarks. According to shown above remarks, there is added recovery value (4.79) obtained according equation (1) of new cited reference [9]. In addition, there are shown chemometric parameters of chromatographic measurements (see new Figure A1.)

The chemometric determination of curve-areas of chromatographic data yields to 8.75-fold increasing in chlorogenic acid. The value is corrected in the revised version of the text.    

[9] Burns D.; Danzer K.; Townshend A. Use of the terms “recovery” and “apparent recovery” in analytical procedures. Pure Appl. Chem. 2002, 74, 22012205.

REVIEWER 1: 2)  That it is the analytical aspect which dominates the study is further emphasized in section 2 where only 2.1 deals with metabolites evolving from the microplants and the other three sections with fragmentation patterns.  It is pretty much impossible to determine from the section what new is being contributed to fragmentation patterns. Line 232 says that reliable MS protocols for quantifying chlorogenic acid in foods aren't relevant for the study. Line 401 claims agreement with previous data. What is needed in section 3 is a summary of accomplishments of the fragmentation work, such as distinguishing between positional stereo-isomers on line 390 and new metabolites on line 25, as compared to state of the art in the area. If the therapeutic aspect is to be pursued, then how the above MS accomplishments contributed to following recovery back toward healthy status, should be included in the discussion.   

AUTHORS: The content of the paper is designed for Special Issue [https://www.mdpi.com/journal/ijms/special_issues/P4R5D7UIZM], devoted to apply ‚mass spectrum‘ to field of metabolomics and udrestand comprehensively biochemical reactions in vitro and in vivo upon effect of magnetic field treatment. Due to this reason content focuses, mainly, on capability of mass spectrometry as robust analytical method for identification, annotation, and structural analysis of metabolites, despite, the fact that the major scope of the Journal focuses on biochemistry. Owing to the fact that, there is proven benefical biological function of aucubin, and there has been achieved perturbation of metabolism of chlorogenic acid upon external magnetic field treatment the focus of the old version of the manuscript was on mass spectrometric data on these species. However, from perspective of novelty, the fragmentation pattern of clorogenic acid is comprehensively examined by many authors. In fact, there is an in-depth analysis of fragmentation paths of the latter analyte (please, consider Table 1 in reference [59].) Particularly, the latter study looks at isomerism of clorogfenic acid. Also, MS data on stereoisomers of clorogenic acid have been in-depth discussed, in fact, in all cited studies by authors such as, for example [47-64]. Moreover, these are not fully representative references to the issue, because of our work does not represent exhaustive review-article. We only confirm known data on chlorogenic acid and aucubin via independent innovative mass spectrometric method based on stochastic dynamics (please, consider equations (1)-(6) and new sub-section 4.5.1.) The latter task aims at proving their validity and their adequate application to annotate unknown species, having comparable chemometric parameters using routine searching algorithms for data processing of mass spectrometric variables of metabolites and their assignment according to library parameters of mass spectra. For instance, these are carbohydrates. Thus, from perspective of novelty of fragmentation pattern there should be highlighted our analysis of salicylated carbohydrates, which mass spectrometric based annotation does not respresent trivial task due to comparable method performances. Please, see new sub section 2.4. Mass spectrometric data on silylated carbohydrate monomers and new Figures A2–A16. There are shown experimental chromatographic and mass spectrometric data on silylated carbohydrates assigned comprehensively via two independent theoretical methods, thus fitting content of the study to thematic issue of the Journal.

[59] Xie C.; Yu K.; Zhong D.; Yuan, T.; Ye, F.; Jarrell J.; Millar A.; Chen, X. Investigation of isomeric transformations of chlorogenic acid in buffers and biological matrixes by ultraperformance liquid chromatography coupled with hybrid quadrupole/ion mobility/orthogonal acceleration time-of-flight mass spectrometry. J. Agric. Food Chem. 2011, 59, 11078–11087.

REVIEWER 1: 3)  Discussing references dealing with how magnetism affects Pyrus communis L. might strengthen the plant development topic, such as T Cebulak et al.,  Effect of abiotic stress factors on polyphenolic content in the skin and flesh of pear by UPLC‑PDA‑Q/TOF‑MS, European Food Research and Technology 245 (2019) 245 2715-2725, and A El-Nasr et al., Effect of Magnetite Nanoparticles (Fe3O4) as Nutritive Supplement on Pear Saplings, Middle East Journal of Applied Sciences 5 (2015) 777-785.  

AUTHORS: There is presented new section ‚Discussion‘ of revised paper. It reflects Reviewer’s remarks and aforementioned contributions to other authors. Please, consider highlighted text of section 3. Discussion and new cited works [85–96]. The revised section ‚Discussion‘, we hope illustrates more clearly for Editors, Reviewers and potential Readers the impact and novelty of our study accounting not only for scope of special issue and mass spectrometric data, but also biochemical aspects of our results from metabolomics.  

REVIEWER 1: 4) Experimental procedure with respect to silyation in 4.1 and 4.2 needs clarification. Section 4.1 says samples for 3 and 72 h analysis were extracted using 70% ethanol and silylated using BSATF. The procedure in 4.2 was extraction in methanol with silylation mentioned on line 484. Which samples and what analyses are associated with each procedure?  

AUTHORS: Section ‚Experimental‘ of revised text is completely re-written. It details on synthesis of sylilated carbohydrates; experimental design of magnetic measurements of plants together with experimental parameters: detail on chromatographic measurements and annotation; advantages and limitation of GC-MS and LC-MS methods which are often used to purposes of metabolomics; brief description of strochastic dynamic theory, quantum chemical computations, and chemometrics, respectively. Relevant reference section is added, as well.   

REVIEWER 1: 5) The long paragraphs need to be broken down into smaller ones. Section 3, for example, consists of only one.  

AUTHORS: The text of the manuscript is fully re-written.

REVIEWER 1: 6) Most of the equations in Appendix A are unreadable.  

AUTHORS: The equation are written via text editor. Perhaps, due to conversion from doc to pdf format most of them are, thus, unreadable for the Reviewer. In order to exclude from such disadvantage, the revised text contains same formulas shown, however as a single embedded into the text figure.  

REVIEWER 1: Comments on the Quality of English Language

Reading the test moves along although when facing the long paragraphs there is a feeling of do I really one to jump into this thing. There are often clumsy phrases, such as on line 15: There are scarce data on this thematic reported, which should read something like: There are few studies about this particular topic. Reading the text aloud can catch some of them, but it's pretty much impossible to correct all of them. 

AUTHORS: The English is revised. The text of the manuscript, is in fact, rewritten, due to extended revision of almost all sections.

RESPONSE TO REVIEWER 1

Comments and Suggestions for Authors

REVIEWER 1: 1) The conclusion highlights the identification of 89 metabolites from the virus infected micro-plants, but says nothing about the effect of magneto-therapy on plant recovery, a topic that was indicated on line 43. That chlorogenic acid enhances resistance is stated on line 53, but that is not the same as recovery. This discrepancy is emphasized by the control employed being virus infected microplants with no MF exposure according to the captions of Figures 1 and 2 and that the discussion section takes up the advantages of MPT but never addresses the degree of recovery after treatment by comparison with metabolites from healthy plants or virus diagnosis by ELISA mentioned in 4.2.   Since one of the aims of IJMS is "strong emphasis on molecular biology", and not just the analytical techniques needed to accomplish the goal, a clarification is needed in the introduction as to what the study intends to contribute to the plant-specific MPT field not already covered in references 6-8.   The claim on line 99 that there was a 14.5 increase in chlorogenic acid after 10 Hz treatment after 72 h relative to 0 Hz doesn't seem to be supported in Figure 1 where the concentration ratio is 1.2/0.18=6.7.  

AUTHORS: The authors would like to thank for valuable Reviewer’s recommendations and remarks. According to shown above remarks, there is added recovery value (4.79) obtained according equation (1) of new cited reference [9]. In addition, there are shown chemometric parameters of chromatographic measurements (see new Figure A1.)

The chemometric determination of curve-areas of chromatographic data yields to 8.75-fold increasing in chlorogenic acid. The value is corrected in the revised version of the text.    

[9] Burns D.; Danzer K.; Townshend A. Use of the terms “recovery” and “apparent recovery” in analytical procedures. Pure Appl. Chem. 2002, 74, 22012205.

REVIEWER 1: 2)  That it is the analytical aspect which dominates the study is further emphasized in section 2 where only 2.1 deals with metabolites evolving from the microplants and the other three sections with fragmentation patterns.  It is pretty much impossible to determine from the section what new is being contributed to fragmentation patterns. Line 232 says that reliable MS protocols for quantifying chlorogenic acid in foods aren't relevant for the study. Line 401 claims agreement with previous data. What is needed in section 3 is a summary of accomplishments of the fragmentation work, such as distinguishing between positional stereo-isomers on line 390 and new metabolites on line 25, as compared to state of the art in the area. If the therapeutic aspect is to be pursued, then how the above MS accomplishments contributed to following recovery back toward healthy status, should be included in the discussion.   

AUTHORS: The content of the paper is designed for Special Issue [https://www.mdpi.com/journal/ijms/special_issues/P4R5D7UIZM], devoted to apply ‚mass spectrum‘ to field of metabolomics and udrestand comprehensively biochemical reactions in vitro and in vivo upon effect of magnetic field treatment. Due to this reason content focuses, mainly, on capability of mass spectrometry as robust analytical method for identification, annotation, and structural analysis of metabolites, despite, the fact that the major scope of the Journal focuses on biochemistry. Owing to the fact that, there is proven benefical biological function of aucubin, and there has been achieved perturbation of metabolism of chlorogenic acid upon external magnetic field treatment the focus of the old version of the manuscript was on mass spectrometric data on these species. However, from perspective of novelty, the fragmentation pattern of clorogenic acid is comprehensively examined by many authors. In fact, there is an in-depth analysis of fragmentation paths of the latter analyte (please, consider Table 1 in reference [59].) Particularly, the latter study looks at isomerism of clorogfenic acid. Also, MS data on stereoisomers of clorogenic acid have been in-depth discussed, in fact, in all cited studies by authors such as, for example [47-64]. Moreover, these are not fully representative references to the issue, because of our work does not represent exhaustive review-article. We only confirm known data on chlorogenic acid and aucubin via independent innovative mass spectrometric method based on stochastic dynamics (please, consider equations (1)-(6) and new sub-section 4.5.1.) The latter task aims at proving their validity and their adequate application to annotate unknown species, having comparable chemometric parameters using routine searching algorithms for data processing of mass spectrometric variables of metabolites and their assignment according to library parameters of mass spectra. For instance, these are carbohydrates. Thus, from perspective of novelty of fragmentation pattern there should be highlighted our analysis of salicylated carbohydrates, which mass spectrometric based annotation does not respresent trivial task due to comparable method performances. Please, see new sub section 2.4. Mass spectrometric data on silylated carbohydrate monomers and new Figures A2–A16. There are shown experimental chromatographic and mass spectrometric data on silylated carbohydrates assigned comprehensively via two independent theoretical methods, thus fitting content of the study to thematic issue of the Journal.

[59] Xie C.; Yu K.; Zhong D.; Yuan, T.; Ye, F.; Jarrell J.; Millar A.; Chen, X. Investigation of isomeric transformations of chlorogenic acid in buffers and biological matrixes by ultraperformance liquid chromatography coupled with hybrid quadrupole/ion mobility/orthogonal acceleration time-of-flight mass spectrometry. J. Agric. Food Chem. 2011, 59, 11078–11087.

REVIEWER 1: 3)  Discussing references dealing with how magnetism affects Pyrus communis L. might strengthen the plant development topic, such as T Cebulak et al.,  Effect of abiotic stress factors on polyphenolic content in the skin and flesh of pear by UPLC‑PDA‑Q/TOF‑MS, European Food Research and Technology 245 (2019) 245 2715-2725, and A El-Nasr et al., Effect of Magnetite Nanoparticles (Fe3O4) as Nutritive Supplement on Pear Saplings, Middle East Journal of Applied Sciences 5 (2015) 777-785.  

AUTHORS: There is presented new section ‚Discussion‘ of revised paper. It reflects Reviewer’s remarks and aforementioned contributions to other authors. Please, consider highlighted text of section 3. Discussion and new cited works [85–96]. The revised section ‚Discussion‘, we hope illustrates more clearly for Editors, Reviewers and potential Readers the impact and novelty of our study accounting not only for scope of special issue and mass spectrometric data, but also biochemical aspects of our results from metabolomics.  

REVIEWER 1: 4) Experimental procedure with respect to silyation in 4.1 and 4.2 needs clarification. Section 4.1 says samples for 3 and 72 h analysis were extracted using 70% ethanol and silylated using BSATF. The procedure in 4.2 was extraction in methanol with silylation mentioned on line 484. Which samples and what analyses are associated with each procedure?  

AUTHORS: Section ‚Experimental‘ of revised text is completely re-written. It details on synthesis of sylilated carbohydrates; experimental design of magnetic measurements of plants together with experimental parameters: detail on chromatographic measurements and annotation; advantages and limitation of GC-MS and LC-MS methods which are often used to purposes of metabolomics; brief description of strochastic dynamic theory, quantum chemical computations, and chemometrics, respectively. Relevant reference section is added, as well.   

REVIEWER 1: 5) The long paragraphs need to be broken down into smaller ones. Section 3, for example, consists of only one.  

AUTHORS: The text of the manuscript is fully re-written.

REVIEWER 1: 6) Most of the equations in Appendix A are unreadable.  

AUTHORS: The equation are written via text editor. Perhaps, due to conversion from doc to pdf format most of them are, thus, unreadable for the Reviewer. In order to exclude from such disadvantage, the revised text contains same formulas shown, however as a single embedded into the text figure.  

REVIEWER 1: Comments on the Quality of English Language

Reading the test moves along although when facing the long paragraphs there is a feeling of do I really one to jump into this thing. There are often clumsy phrases, such as on line 15: There are scarce data on this thematic reported, which should read something like: There are few studies about this particular topic. Reading the text aloud can catch some of them, but it's pretty much impossible to correct all of them. 

AUTHORS: The English is revised. The text of the manuscript, is in fact, rewritten, due to extended revision of almost all sections.

RESPONSE TO REVIEWER 1

Comments and Suggestions for Authors

REVIEWER 1: 1) The conclusion highlights the identification of 89 metabolites from the virus infected micro-plants, but says nothing about the effect of magneto-therapy on plant recovery, a topic that was indicated on line 43. That chlorogenic acid enhances resistance is stated on line 53, but that is not the same as recovery. This discrepancy is emphasized by the control employed being virus infected microplants with no MF exposure according to the captions of Figures 1 and 2 and that the discussion section takes up the advantages of MPT but never addresses the degree of recovery after treatment by comparison with metabolites from healthy plants or virus diagnosis by ELISA mentioned in 4.2.   Since one of the aims of IJMS is "strong emphasis on molecular biology", and not just the analytical techniques needed to accomplish the goal, a clarification is needed in the introduction as to what the study intends to contribute to the plant-specific MPT field not already covered in references 6-8.   The claim on line 99 that there was a 14.5 increase in chlorogenic acid after 10 Hz treatment after 72 h relative to 0 Hz doesn't seem to be supported in Figure 1 where the concentration ratio is 1.2/0.18=6.7.  

AUTHORS: The authors would like to thank for valuable Reviewer’s recommendations and remarks. According to shown above remarks, there is added recovery value (4.79) obtained according equation (1) of new cited reference [9]. In addition, there are shown chemometric parameters of chromatographic measurements (see new Figure A1.)

The chemometric determination of curve-areas of chromatographic data yields to 8.75-fold increasing in chlorogenic acid. The value is corrected in the revised version of the text.    

[9] Burns D.; Danzer K.; Townshend A. Use of the terms “recovery” and “apparent recovery” in analytical procedures. Pure Appl. Chem. 2002, 74, 22012205.

REVIEWER 1: 2)  That it is the analytical aspect which dominates the study is further emphasized in section 2 where only 2.1 deals with metabolites evolving from the microplants and the other three sections with fragmentation patterns.  It is pretty much impossible to determine from the section what new is being contributed to fragmentation patterns. Line 232 says that reliable MS protocols for quantifying chlorogenic acid in foods aren't relevant for the study. Line 401 claims agreement with previous data. What is needed in section 3 is a summary of accomplishments of the fragmentation work, such as distinguishing between positional stereo-isomers on line 390 and new metabolites on line 25, as compared to state of the art in the area. If the therapeutic aspect is to be pursued, then how the above MS accomplishments contributed to following recovery back toward healthy status, should be included in the discussion.   

AUTHORS: The content of the paper is designed for Special Issue [https://www.mdpi.com/journal/ijms/special_issues/P4R5D7UIZM], devoted to apply ‚mass spectrum‘ to field of metabolomics and udrestand comprehensively biochemical reactions in vitro and in vivo upon effect of magnetic field treatment. Due to this reason content focuses, mainly, on capability of mass spectrometry as robust analytical method for identification, annotation, and structural analysis of metabolites, despite, the fact that the major scope of the Journal focuses on biochemistry. Owing to the fact that, there is proven benefical biological function of aucubin, and there has been achieved perturbation of metabolism of chlorogenic acid upon external magnetic field treatment the focus of the old version of the manuscript was on mass spectrometric data on these species. However, from perspective of novelty, the fragmentation pattern of clorogenic acid is comprehensively examined by many authors. In fact, there is an in-depth analysis of fragmentation paths of the latter analyte (please, consider Table 1 in reference [59].) Particularly, the latter study looks at isomerism of clorogfenic acid. Also, MS data on stereoisomers of clorogenic acid have been in-depth discussed, in fact, in all cited studies by authors such as, for example [47-64]. Moreover, these are not fully representative references to the issue, because of our work does not represent exhaustive review-article. We only confirm known data on chlorogenic acid and aucubin via independent innovative mass spectrometric method based on stochastic dynamics (please, consider equations (1)-(6) and new sub-section 4.5.1.) The latter task aims at proving their validity and their adequate application to annotate unknown species, having comparable chemometric parameters using routine searching algorithms for data processing of mass spectrometric variables of metabolites and their assignment according to library parameters of mass spectra. For instance, these are carbohydrates. Thus, from perspective of novelty of fragmentation pattern there should be highlighted our analysis of salicylated carbohydrates, which mass spectrometric based annotation does not respresent trivial task due to comparable method performances. Please, see new sub section 2.4. Mass spectrometric data on silylated carbohydrate monomers and new Figures A2–A16. There are shown experimental chromatographic and mass spectrometric data on silylated carbohydrates assigned comprehensively via two independent theoretical methods, thus fitting content of the study to thematic issue of the Journal.

[59] Xie C.; Yu K.; Zhong D.; Yuan, T.; Ye, F.; Jarrell J.; Millar A.; Chen, X. Investigation of isomeric transformations of chlorogenic acid in buffers and biological matrixes by ultraperformance liquid chromatography coupled with hybrid quadrupole/ion mobility/orthogonal acceleration time-of-flight mass spectrometry. J. Agric. Food Chem. 2011, 59, 11078–11087.

REVIEWER 1: 3)  Discussing references dealing with how magnetism affects Pyrus communis L. might strengthen the plant development topic, such as T Cebulak et al.,  Effect of abiotic stress factors on polyphenolic content in the skin and flesh of pear by UPLC‑PDA‑Q/TOF‑MS, European Food Research and Technology 245 (2019) 245 2715-2725, and A El-Nasr et al., Effect of Magnetite Nanoparticles (Fe3O4) as Nutritive Supplement on Pear Saplings, Middle East Journal of Applied Sciences 5 (2015) 777-785.  

AUTHORS: There is presented new section ‚Discussion‘ of revised paper. It reflects Reviewer’s remarks and aforementioned contributions to other authors. Please, consider highlighted text of section 3. Discussion and new cited works [85–96]. The revised section ‚Discussion‘, we hope illustrates more clearly for Editors, Reviewers and potential Readers the impact and novelty of our study accounting not only for scope of special issue and mass spectrometric data, but also biochemical aspects of our results from metabolomics.  

REVIEWER 1: 4) Experimental procedure with respect to silyation in 4.1 and 4.2 needs clarification. Section 4.1 says samples for 3 and 72 h analysis were extracted using 70% ethanol and silylated using BSATF. The procedure in 4.2 was extraction in methanol with silylation mentioned on line 484. Which samples and what analyses are associated with each procedure?  

AUTHORS: Section ‚Experimental‘ of revised text is completely re-written. It details on synthesis of sylilated carbohydrates; experimental design of magnetic measurements of plants together with experimental parameters: detail on chromatographic measurements and annotation; advantages and limitation of GC-MS and LC-MS methods which are often used to purposes of metabolomics; brief description of strochastic dynamic theory, quantum chemical computations, and chemometrics, respectively. Relevant reference section is added, as well.   

REVIEWER 1: 5) The long paragraphs need to be broken down into smaller ones. Section 3, for example, consists of only one.  

AUTHORS: The text of the manuscript is fully re-written.

REVIEWER 1: 6) Most of the equations in Appendix A are unreadable.  

AUTHORS: The equation are written via text editor. Perhaps, due to conversion from doc to pdf format most of them are, thus, unreadable for the Reviewer. In order to exclude from such disadvantage, the revised text contains same formulas shown, however as a single embedded into the text figure.  

REVIEWER 1: Comments on the Quality of English Language

Reading the test moves along although when facing the long paragraphs there is a feeling of do I really one to jump into this thing. There are often clumsy phrases, such as on line 15: There are scarce data on this thematic reported, which should read something like: There are few studies about this particular topic. Reading the text aloud can catch some of them, but it's pretty much impossible to correct all of them. 

AUTHORS: The English is revised. The text of the manuscript, is in fact, rewritten, due to extended revision of almost all sections.

RESPONSE TO REVIEWER 1

Comments and Suggestions for Authors

REVIEWER 1: 1) The conclusion highlights the identification of 89 metabolites from the virus infected micro-plants, but says nothing about the effect of magneto-therapy on plant recovery, a topic that was indicated on line 43. That chlorogenic acid enhances resistance is stated on line 53, but that is not the same as recovery. This discrepancy is emphasized by the control employed being virus infected microplants with no MF exposure according to the captions of Figures 1 and 2 and that the discussion section takes up the advantages of MPT but never addresses the degree of recovery after treatment by comparison with metabolites from healthy plants or virus diagnosis by ELISA mentioned in 4.2.   Since one of the aims of IJMS is "strong emphasis on molecular biology", and not just the analytical techniques needed to accomplish the goal, a clarification is needed in the introduction as to what the study intends to contribute to the plant-specific MPT field not already covered in references 6-8.   The claim on line 99 that there was a 14.5 increase in chlorogenic acid after 10 Hz treatment after 72 h relative to 0 Hz doesn't seem to be supported in Figure 1 where the concentration ratio is 1.2/0.18=6.7.  

AUTHORS: The authors would like to thank for valuable Reviewer’s recommendations and remarks. According to shown above remarks, there is added recovery value (4.79) obtained according equation (1) of new cited reference [9]. In addition, there are shown chemometric parameters of chromatographic measurements (see new Figure A1.)

The chemometric determination of curve-areas of chromatographic data yields to 8.75-fold increasing in chlorogenic acid. The value is corrected in the revised version of the text.    

[9] Burns D.; Danzer K.; Townshend A. Use of the terms “recovery” and “apparent recovery” in analytical procedures. Pure Appl. Chem. 2002, 74, 22012205.

REVIEWER 1: 2)  That it is the analytical aspect which dominates the study is further emphasized in section 2 where only 2.1 deals with metabolites evolving from the microplants and the other three sections with fragmentation patterns.  It is pretty much impossible to determine from the section what new is being contributed to fragmentation patterns. Line 232 says that reliable MS protocols for quantifying chlorogenic acid in foods aren't relevant for the study. Line 401 claims agreement with previous data. What is needed in section 3 is a summary of accomplishments of the fragmentation work, such as distinguishing between positional stereo-isomers on line 390 and new metabolites on line 25, as compared to state of the art in the area. If the therapeutic aspect is to be pursued, then how the above MS accomplishments contributed to following recovery back toward healthy status, should be included in the discussion.   

AUTHORS: The content of the paper is designed for Special Issue [https://www.mdpi.com/journal/ijms/special_issues/P4R5D7UIZM], devoted to apply ‚mass spectrum‘ to field of metabolomics and udrestand comprehensively biochemical reactions in vitro and in vivo upon effect of magnetic field treatment. Due to this reason content focuses, mainly, on capability of mass spectrometry as robust analytical method for identification, annotation, and structural analysis of metabolites, despite, the fact that the major scope of the Journal focuses on biochemistry. Owing to the fact that, there is proven benefical biological function of aucubin, and there has been achieved perturbation of metabolism of chlorogenic acid upon external magnetic field treatment the focus of the old version of the manuscript was on mass spectrometric data on these species. However, from perspective of novelty, the fragmentation pattern of clorogenic acid is comprehensively examined by many authors. In fact, there is an in-depth analysis of fragmentation paths of the latter analyte (please, consider Table 1 in reference [59].) Particularly, the latter study looks at isomerism of clorogfenic acid. Also, MS data on stereoisomers of clorogenic acid have been in-depth discussed, in fact, in all cited studies by authors such as, for example [47-64]. Moreover, these are not fully representative references to the issue, because of our work does not represent exhaustive review-article. We only confirm known data on chlorogenic acid and aucubin via independent innovative mass spectrometric method based on stochastic dynamics (please, consider equations (1)-(6) and new sub-section 4.5.1.) The latter task aims at proving their validity and their adequate application to annotate unknown species, having comparable chemometric parameters using routine searching algorithms for data processing of mass spectrometric variables of metabolites and their assignment according to library parameters of mass spectra. For instance, these are carbohydrates. Thus, from perspective of novelty of fragmentation pattern there should be highlighted our analysis of salicylated carbohydrates, which mass spectrometric based annotation does not respresent trivial task due to comparable method performances. Please, see new sub section 2.4. Mass spectrometric data on silylated carbohydrate monomers and new Figures A2–A16. There are shown experimental chromatographic and mass spectrometric data on silylated carbohydrates assigned comprehensively via two independent theoretical methods, thus fitting content of the study to thematic issue of the Journal.

[59] Xie C.; Yu K.; Zhong D.; Yuan, T.; Ye, F.; Jarrell J.; Millar A.; Chen, X. Investigation of isomeric transformations of chlorogenic acid in buffers and biological matrixes by ultraperformance liquid chromatography coupled with hybrid quadrupole/ion mobility/orthogonal acceleration time-of-flight mass spectrometry. J. Agric. Food Chem. 2011, 59, 11078–11087.

REVIEWER 1: 3)  Discussing references dealing with how magnetism affects Pyrus communis L. might strengthen the plant development topic, such as T Cebulak et al.,  Effect of abiotic stress factors on polyphenolic content in the skin and flesh of pear by UPLC‑PDA‑Q/TOF‑MS, European Food Research and Technology 245 (2019) 245 2715-2725, and A El-Nasr et al., Effect of Magnetite Nanoparticles (Fe3O4) as Nutritive Supplement on Pear Saplings, Middle East Journal of Applied Sciences 5 (2015) 777-785.  

AUTHORS: There is presented new section ‚Discussion‘ of revised paper. It reflects Reviewer’s remarks and aforementioned contributions to other authors. Please, consider highlighted text of section 3. Discussion and new cited works [85–96]. The revised section ‚Discussion‘, we hope illustrates more clearly for Editors, Reviewers and potential Readers the impact and novelty of our study accounting not only for scope of special issue and mass spectrometric data, but also biochemical aspects of our results from metabolomics.  

REVIEWER 1: 4) Experimental procedure with respect to silyation in 4.1 and 4.2 needs clarification. Section 4.1 says samples for 3 and 72 h analysis were extracted using 70% ethanol and silylated using BSATF. The procedure in 4.2 was extraction in methanol with silylation mentioned on line 484. Which samples and what analyses are associated with each procedure?  

AUTHORS: Section ‚Experimental‘ of revised text is completely re-written. It details on synthesis of sylilated carbohydrates; experimental design of magnetic measurements of plants together with experimental parameters: detail on chromatographic measurements and annotation; advantages and limitation of GC-MS and LC-MS methods which are often used to purposes of metabolomics; brief description of strochastic dynamic theory, quantum chemical computations, and chemometrics, respectively. Relevant reference section is added, as well.   

REVIEWER 1: 5) The long paragraphs need to be broken down into smaller ones. Section 3, for example, consists of only one.  

AUTHORS: The text of the manuscript is fully re-written.

REVIEWER 1: 6) Most of the equations in Appendix A are unreadable.  

AUTHORS: The equation are written via text editor. Perhaps, due to conversion from doc to pdf format most of them are, thus, unreadable for the Reviewer. In order to exclude from such disadvantage, the revised text contains same formulas shown, however as a single embedded into the text figure.  

REVIEWER 1: Comments on the Quality of English Language

Reading the test moves along although when facing the long paragraphs there is a feeling of do I really one to jump into this thing. There are often clumsy phrases, such as on line 15: There are scarce data on this thematic reported, which should read something like: There are few studies about this particular topic. Reading the text aloud can catch some of them, but it's pretty much impossible to correct all of them. 

AUTHORS: The English is revised. The text of the manuscript, is in fact, rewritten, due to extended revision of almost all sections.

RESPONSE TO REVIEWER 1

Comments and Suggestions for Authors

REVIEWER 1: 1) The conclusion highlights the identification of 89 metabolites from the virus infected micro-plants, but says nothing about the effect of magneto-therapy on plant recovery, a topic that was indicated on line 43. That chlorogenic acid enhances resistance is stated on line 53, but that is not the same as recovery. This discrepancy is emphasized by the control employed being virus infected microplants with no MF exposure according to the captions of Figures 1 and 2 and that the discussion section takes up the advantages of MPT but never addresses the degree of recovery after treatment by comparison with metabolites from healthy plants or virus diagnosis by ELISA mentioned in 4.2.   Since one of the aims of IJMS is "strong emphasis on molecular biology", and not just the analytical techniques needed to accomplish the goal, a clarification is needed in the introduction as to what the study intends to contribute to the plant-specific MPT field not already covered in references 6-8.   The claim on line 99 that there was a 14.5 increase in chlorogenic acid after 10 Hz treatment after 72 h relative to 0 Hz doesn't seem to be supported in Figure 1 where the concentration ratio is 1.2/0.18=6.7.  

AUTHORS: The authors would like to thank for valuable Reviewer’s recommendations and remarks. According to shown above remarks, there is added recovery value (4.79) obtained according equation (1) of new cited reference [9]. In addition, there are shown chemometric parameters of chromatographic measurements (see new Figure A1.)

The chemometric determination of curve-areas of chromatographic data yields to 8.75-fold increasing in chlorogenic acid. The value is corrected in the revised version of the text.    

[9] Burns D.; Danzer K.; Townshend A. Use of the terms “recovery” and “apparent recovery” in analytical procedures. Pure Appl. Chem. 2002, 74, 22012205.

REVIEWER 1: 2)  That it is the analytical aspect which dominates the study is further emphasized in section 2 where only 2.1 deals with metabolites evolving from the microplants and the other three sections with fragmentation patterns.  It is pretty much impossible to determine from the section what new is being contributed to fragmentation patterns. Line 232 says that reliable MS protocols for quantifying chlorogenic acid in foods aren't relevant for the study. Line 401 claims agreement with previous data. What is needed in section 3 is a summary of accomplishments of the fragmentation work, such as distinguishing between positional stereo-isomers on line 390 and new metabolites on line 25, as compared to state of the art in the area. If the therapeutic aspect is to be pursued, then how the above MS accomplishments contributed to following recovery back toward healthy status, should be included in the discussion.   

AUTHORS: The content of the paper is designed for Special Issue [https://www.mdpi.com/journal/ijms/special_issues/P4R5D7UIZM], devoted to apply ‚mass spectrum‘ to field of metabolomics and udrestand comprehensively biochemical reactions in vitro and in vivo upon effect of magnetic field treatment. Due to this reason content focuses, mainly, on capability of mass spectrometry as robust analytical method for identification, annotation, and structural analysis of metabolites, despite, the fact that the major scope of the Journal focuses on biochemistry. Owing to the fact that, there is proven benefical biological function of aucubin, and there has been achieved perturbation of metabolism of chlorogenic acid upon external magnetic field treatment the focus of the old version of the manuscript was on mass spectrometric data on these species. However, from perspective of novelty, the fragmentation pattern of clorogenic acid is comprehensively examined by many authors. In fact, there is an in-depth analysis of fragmentation paths of the latter analyte (please, consider Table 1 in reference [59].) Particularly, the latter study looks at isomerism of clorogfenic acid. Also, MS data on stereoisomers of clorogenic acid have been in-depth discussed, in fact, in all cited studies by authors such as, for example [47-64]. Moreover, these are not fully representative references to the issue, because of our work does not represent exhaustive review-article. We only confirm known data on chlorogenic acid and aucubin via independent innovative mass spectrometric method based on stochastic dynamics (please, consider equations (1)-(6) and new sub-section 4.5.1.) The latter task aims at proving their validity and their adequate application to annotate unknown species, having comparable chemometric parameters using routine searching algorithms for data processing of mass spectrometric variables of metabolites and their assignment according to library parameters of mass spectra. For instance, these are carbohydrates. Thus, from perspective of novelty of fragmentation pattern there should be highlighted our analysis of salicylated carbohydrates, which mass spectrometric based annotation does not respresent trivial task due to comparable method performances. Please, see new sub section 2.4. Mass spectrometric data on silylated carbohydrate monomers and new Figures A2–A16. There are shown experimental chromatographic and mass spectrometric data on silylated carbohydrates assigned comprehensively via two independent theoretical methods, thus fitting content of the study to thematic issue of the Journal.

[59] Xie C.; Yu K.; Zhong D.; Yuan, T.; Ye, F.; Jarrell J.; Millar A.; Chen, X. Investigation of isomeric transformations of chlorogenic acid in buffers and biological matrixes by ultraperformance liquid chromatography coupled with hybrid quadrupole/ion mobility/orthogonal acceleration time-of-flight mass spectrometry. J. Agric. Food Chem. 2011, 59, 11078–11087.

REVIEWER 1: 3)  Discussing references dealing with how magnetism affects Pyrus communis L. might strengthen the plant development topic, such as T Cebulak et al.,  Effect of abiotic stress factors on polyphenolic content in the skin and flesh of pear by UPLC‑PDA‑Q/TOF‑MS, European Food Research and Technology 245 (2019) 245 2715-2725, and A El-Nasr et al., Effect of Magnetite Nanoparticles (Fe3O4) as Nutritive Supplement on Pear Saplings, Middle East Journal of Applied Sciences 5 (2015) 777-785.  

AUTHORS: There is presented new section ‚Discussion‘ of revised paper. It reflects Reviewer’s remarks and aforementioned contributions to other authors. Please, consider highlighted text of section 3. Discussion and new cited works [85–96]. The revised section ‚Discussion‘, we hope illustrates more clearly for Editors, Reviewers and potential Readers the impact and novelty of our study accounting not only for scope of special issue and mass spectrometric data, but also biochemical aspects of our results from metabolomics.  

REVIEWER 1: 4) Experimental procedure with respect to silyation in 4.1 and 4.2 needs clarification. Section 4.1 says samples for 3 and 72 h analysis were extracted using 70% ethanol and silylated using BSATF. The procedure in 4.2 was extraction in methanol with silylation mentioned on line 484. Which samples and what analyses are associated with each procedure?  

AUTHORS: Section ‚Experimental‘ of revised text is completely re-written. It details on synthesis of sylilated carbohydrates; experimental design of magnetic measurements of plants together with experimental parameters: detail on chromatographic measurements and annotation; advantages and limitation of GC-MS and LC-MS methods which are often used to purposes of metabolomics; brief description of strochastic dynamic theory, quantum chemical computations, and chemometrics, respectively. Relevant reference section is added, as well.   

REVIEWER 1: 5) The long paragraphs need to be broken down into smaller ones. Section 3, for example, consists of only one.  

AUTHORS: The text of the manuscript is fully re-written.

REVIEWER 1: 6) Most of the equations in Appendix A are unreadable.  

AUTHORS: The equation are written via text editor. Perhaps, due to conversion from doc to pdf format most of them are, thus, unreadable for the Reviewer. In order to exclude from such disadvantage, the revised text contains same formulas shown, however as a single embedded into the text figure.  

REVIEWER 1: Comments on the Quality of English Language

Reading the test moves along although when facing the long paragraphs there is a feeling of do I really one to jump into this thing. There are often clumsy phrases, such as on line 15: There are scarce data on this thematic reported, which should read something like: There are few studies about this particular topic. Reading the text aloud can catch some of them, but it's pretty much impossible to correct all of them. 

AUTHORS: The English is revised. The text of the manuscript, is in fact, rewritten, due to extended revision of almost all sections.

RESPONSE TO REVIEWER 1

Comments and Suggestions for Authors

REVIEWER 1: 1) The conclusion highlights the identification of 89 metabolites from the virus infected micro-plants, but says nothing about the effect of magneto-therapy on plant recovery, a topic that was indicated on line 43. That chlorogenic acid enhances resistance is stated on line 53, but that is not the same as recovery. This discrepancy is emphasized by the control employed being virus infected microplants with no MF exposure according to the captions of Figures 1 and 2 and that the discussion section takes up the advantages of MPT but never addresses the degree of recovery after treatment by comparison with metabolites from healthy plants or virus diagnosis by ELISA mentioned in 4.2.   Since one of the aims of IJMS is "strong emphasis on molecular biology", and not just the analytical techniques needed to accomplish the goal, a clarification is needed in the introduction as to what the study intends to contribute to the plant-specific MPT field not already covered in references 6-8.   The claim on line 99 that there was a 14.5 increase in chlorogenic acid after 10 Hz treatment after 72 h relative to 0 Hz doesn't seem to be supported in Figure 1 where the concentration ratio is 1.2/0.18=6.7.  

AUTHORS: The authors would like to thank for valuable Reviewer’s recommendations and remarks. According to shown above remarks, there is added recovery value (4.79) obtained according equation (1) of new cited reference [9]. In addition, there are shown chemometric parameters of chromatographic measurements (see new Figure A1.)

The chemometric determination of curve-areas of chromatographic data yields to 8.75-fold increasing in chlorogenic acid. The value is corrected in the revised version of the text.    

[9] Burns D.; Danzer K.; Townshend A. Use of the terms “recovery” and “apparent recovery” in analytical procedures. Pure Appl. Chem. 2002, 74, 22012205.

REVIEWER 1: 2)  That it is the analytical aspect which dominates the study is further emphasized in section 2 where only 2.1 deals with metabolites evolving from the microplants and the other three sections with fragmentation patterns.  It is pretty much impossible to determine from the section what new is being contributed to fragmentation patterns. Line 232 says that reliable MS protocols for quantifying chlorogenic acid in foods aren't relevant for the study. Line 401 claims agreement with previous data. What is needed in section 3 is a summary of accomplishments of the fragmentation work, such as distinguishing between positional stereo-isomers on line 390 and new metabolites on line 25, as compared to state of the art in the area. If the therapeutic aspect is to be pursued, then how the above MS accomplishments contributed to following recovery back toward healthy status, should be included in the discussion.   

AUTHORS: The content of the paper is designed for Special Issue [https://www.mdpi.com/journal/ijms/special_issues/P4R5D7UIZM], devoted to apply ‚mass spectrum‘ to field of metabolomics and udrestand comprehensively biochemical reactions in vitro and in vivo upon effect of magnetic field treatment. Due to this reason content focuses, mainly, on capability of mass spectrometry as robust analytical method for identification, annotation, and structural analysis of metabolites, despite, the fact that the major scope of the Journal focuses on biochemistry. Owing to the fact that, there is proven benefical biological function of aucubin, and there has been achieved perturbation of metabolism of chlorogenic acid upon external magnetic field treatment the focus of the old version of the manuscript was on mass spectrometric data on these species. However, from perspective of novelty, the fragmentation pattern of clorogenic acid is comprehensively examined by many authors. In fact, there is an in-depth analysis of fragmentation paths of the latter analyte (please, consider Table 1 in reference [59].) Particularly, the latter study looks at isomerism of clorogfenic acid. Also, MS data on stereoisomers of clorogenic acid have been in-depth discussed, in fact, in all cited studies by authors such as, for example [47-64]. Moreover, these are not fully representative references to the issue, because of our work does not represent exhaustive review-article. We only confirm known data on chlorogenic acid and aucubin via independent innovative mass spectrometric method based on stochastic dynamics (please, consider equations (1)-(6) and new sub-section 4.5.1.) The latter task aims at proving their validity and their adequate application to annotate unknown species, having comparable chemometric parameters using routine searching algorithms for data processing of mass spectrometric variables of metabolites and their assignment according to library parameters of mass spectra. For instance, these are carbohydrates. Thus, from perspective of novelty of fragmentation pattern there should be highlighted our analysis of salicylated carbohydrates, which mass spectrometric based annotation does not respresent trivial task due to comparable method performances. Please, see new sub section 2.4. Mass spectrometric data on silylated carbohydrate monomers and new Figures A2–A16. There are shown experimental chromatographic and mass spectrometric data on silylated carbohydrates assigned comprehensively via two independent theoretical methods, thus fitting content of the study to thematic issue of the Journal.

[59] Xie C.; Yu K.; Zhong D.; Yuan, T.; Ye, F.; Jarrell J.; Millar A.; Chen, X. Investigation of isomeric transformations of chlorogenic acid in buffers and biological matrixes by ultraperformance liquid chromatography coupled with hybrid quadrupole/ion mobility/orthogonal acceleration time-of-flight mass spectrometry. J. Agric. Food Chem. 2011, 59, 11078–11087.

REVIEWER 1: 3)  Discussing references dealing with how magnetism affects Pyrus communis L. might strengthen the plant development topic, such as T Cebulak et al.,  Effect of abiotic stress factors on polyphenolic content in the skin and flesh of pear by UPLC‑PDA‑Q/TOF‑MS, European Food Research and Technology 245 (2019) 245 2715-2725, and A El-Nasr et al., Effect of Magnetite Nanoparticles (Fe3O4) as Nutritive Supplement on Pear Saplings, Middle East Journal of Applied Sciences 5 (2015) 777-785.  

AUTHORS: There is presented new section ‚Discussion‘ of revised paper. It reflects Reviewer’s remarks and aforementioned contributions to other authors. Please, consider highlighted text of section 3. Discussion and new cited works [85–96]. The revised section ‚Discussion‘, we hope illustrates more clearly for Editors, Reviewers and potential Readers the impact and novelty of our study accounting not only for scope of special issue and mass spectrometric data, but also biochemical aspects of our results from metabolomics.  

REVIEWER 1: 4) Experimental procedure with respect to silyation in 4.1 and 4.2 needs clarification. Section 4.1 says samples for 3 and 72 h analysis were extracted using 70% ethanol and silylated using BSATF. The procedure in 4.2 was extraction in methanol with silylation mentioned on line 484. Which samples and what analyses are associated with each procedure?  

AUTHORS: Section ‚Experimental‘ of revised text is completely re-written. It details on synthesis of sylilated carbohydrates; experimental design of magnetic measurements of plants together with experimental parameters: detail on chromatographic measurements and annotation; advantages and limitation of GC-MS and LC-MS methods which are often used to purposes of metabolomics; brief description of strochastic dynamic theory, quantum chemical computations, and chemometrics, respectively. Relevant reference section is added, as well.   

REVIEWER 1: 5) The long paragraphs need to be broken down into smaller ones. Section 3, for example, consists of only one.  

AUTHORS: The text of the manuscript is fully re-written.

REVIEWER 1: 6) Most of the equations in Appendix A are unreadable.  

AUTHORS: The equation are written via text editor. Perhaps, due to conversion from doc to pdf format most of them are, thus, unreadable for the Reviewer. In order to exclude from such disadvantage, the revised text contains same formulas shown, however as a single embedded into the text figure.  

REVIEWER 1: Comments on the Quality of English Language

Reading the test moves along although when facing the long paragraphs there is a feeling of do I really one to jump into this thing. There are often clumsy phrases, such as on line 15: There are scarce data on this thematic reported, which should read something like: There are few studies about this particular topic. Reading the text aloud can catch some of them, but it's pretty much impossible to correct all of them. 

AUTHORS: The English is revised. The text of the manuscript, is in fact, rewritten, due to extended revision of almost all sections.

RESPONSE TO REVIEWER 1

Comments and Suggestions for Authors

REVIEWER 1: 1) The conclusion highlights the identification of 89 metabolites from the virus infected micro-plants, but says nothing about the effect of magneto-therapy on plant recovery, a topic that was indicated on line 43. That chlorogenic acid enhances resistance is stated on line 53, but that is not the same as recovery. This discrepancy is emphasized by the control employed being virus infected microplants with no MF exposure according to the captions of Figures 1 and 2 and that the discussion section takes up the advantages of MPT but never addresses the degree of recovery after treatment by comparison with metabolites from healthy plants or virus diagnosis by ELISA mentioned in 4.2.   Since one of the aims of IJMS is "strong emphasis on molecular biology", and not just the analytical techniques needed to accomplish the goal, a clarification is needed in the introduction as to what the study intends to contribute to the plant-specific MPT field not already covered in references 6-8.   The claim on line 99 that there was a 14.5 increase in chlorogenic acid after 10 Hz treatment after 72 h relative to 0 Hz doesn't seem to be supported in Figure 1 where the concentration ratio is 1.2/0.18=6.7.  

AUTHORS: The authors would like to thank for valuable Reviewer’s recommendations and remarks. According to shown above remarks, there is added recovery value (4.79) obtained according equation (1) of new cited reference [9]. In addition, there are shown chemometric parameters of chromatographic measurements (see new Figure A1.)

The chemometric determination of curve-areas of chromatographic data yields to 8.75-fold increasing in chlorogenic acid. The value is corrected in the revised version of the text.    

[9] Burns D.; Danzer K.; Townshend A. Use of the terms “recovery” and “apparent recovery” in analytical procedures. Pure Appl. Chem. 2002, 74, 22012205.

REVIEWER 1: 2)  That it is the analytical aspect which dominates the study is further emphasized in section 2 where only 2.1 deals with metabolites evolving from the microplants and the other three sections with fragmentation patterns.  It is pretty much impossible to determine from the section what new is being contributed to fragmentation patterns. Line 232 says that reliable MS protocols for quantifying chlorogenic acid in foods aren't relevant for the study. Line 401 claims agreement with previous data. What is needed in section 3 is a summary of accomplishments of the fragmentation work, such as distinguishing between positional stereo-isomers on line 390 and new metabolites on line 25, as compared to state of the art in the area. If the therapeutic aspect is to be pursued, then how the above MS accomplishments contributed to following recovery back toward healthy status, should be included in the discussion.   

AUTHORS: The content of the paper is designed for Special Issue [https://www.mdpi.com/journal/ijms/special_issues/P4R5D7UIZM], devoted to apply ‚mass spectrum‘ to field of metabolomics and udrestand comprehensively biochemical reactions in vitro and in vivo upon effect of magnetic field treatment. Due to this reason content focuses, mainly, on capability of mass spectrometry as robust analytical method for identification, annotation, and structural analysis of metabolites, despite, the fact that the major scope of the Journal focuses on biochemistry. Owing to the fact that, there is proven benefical biological function of aucubin, and there has been achieved perturbation of metabolism of chlorogenic acid upon external magnetic field treatment the focus of the old version of the manuscript was on mass spectrometric data on these species. However, from perspective of novelty, the fragmentation pattern of clorogenic acid is comprehensively examined by many authors. In fact, there is an in-depth analysis of fragmentation paths of the latter analyte (please, consider Table 1 in reference [59].) Particularly, the latter study looks at isomerism of clorogfenic acid. Also, MS data on stereoisomers of clorogenic acid have been in-depth discussed, in fact, in all cited studies by authors such as, for example [47-64]. Moreover, these are not fully representative references to the issue, because of our work does not represent exhaustive review-article. We only confirm known data on chlorogenic acid and aucubin via independent innovative mass spectrometric method based on stochastic dynamics (please, consider equations (1)-(6) and new sub-section 4.5.1.) The latter task aims at proving their validity and their adequate application to annotate unknown species, having comparable chemometric parameters using routine searching algorithms for data processing of mass spectrometric variables of metabolites and their assignment according to library parameters of mass spectra. For instance, these are carbohydrates. Thus, from perspective of novelty of fragmentation pattern there should be highlighted our analysis of salicylated carbohydrates, which mass spectrometric based annotation does not respresent trivial task due to comparable method performances. Please, see new sub section 2.4. Mass spectrometric data on silylated carbohydrate monomers and new Figures A2–A16. There are shown experimental chromatographic and mass spectrometric data on silylated carbohydrates assigned comprehensively via two independent theoretical methods, thus fitting content of the study to thematic issue of the Journal.

[59] Xie C.; Yu K.; Zhong D.; Yuan, T.; Ye, F.; Jarrell J.; Millar A.; Chen, X. Investigation of isomeric transformations of chlorogenic acid in buffers and biological matrixes by ultraperformance liquid chromatography coupled with hybrid quadrupole/ion mobility/orthogonal acceleration time-of-flight mass spectrometry. J. Agric. Food Chem. 2011, 59, 11078–11087.

REVIEWER 1: 3)  Discussing references dealing with how magnetism affects Pyrus communis L. might strengthen the plant development topic, such as T Cebulak et al.,  Effect of abiotic stress factors on polyphenolic content in the skin and flesh of pear by UPLC‑PDA‑Q/TOF‑MS, European Food Research and Technology 245 (2019) 245 2715-2725, and A El-Nasr et al., Effect of Magnetite Nanoparticles (Fe3O4) as Nutritive Supplement on Pear Saplings, Middle East Journal of Applied Sciences 5 (2015) 777-785.  

AUTHORS: There is presented new section ‚Discussion‘ of revised paper. It reflects Reviewer’s remarks and aforementioned contributions to other authors. Please, consider highlighted text of section 3. Discussion and new cited works [85–96]. The revised section ‚Discussion‘, we hope illustrates more clearly for Editors, Reviewers and potential Readers the impact and novelty of our study accounting not only for scope of special issue and mass spectrometric data, but also biochemical aspects of our results from metabolomics.  

REVIEWER 1: 4) Experimental procedure with respect to silyation in 4.1 and 4.2 needs clarification. Section 4.1 says samples for 3 and 72 h analysis were extracted using 70% ethanol and silylated using BSATF. The procedure in 4.2 was extraction in methanol with silylation mentioned on line 484. Which samples and what analyses are associated with each procedure?  

AUTHORS: Section ‚Experimental‘ of revised text is completely re-written. It details on synthesis of sylilated carbohydrates; experimental design of magnetic measurements of plants together with experimental parameters: detail on chromatographic measurements and annotation; advantages and limitation of GC-MS and LC-MS methods which are often used to purposes of metabolomics; brief description of strochastic dynamic theory, quantum chemical computations, and chemometrics, respectively. Relevant reference section is added, as well.   

REVIEWER 1: 5) The long paragraphs need to be broken down into smaller ones. Section 3, for example, consists of only one.  

AUTHORS: The text of the manuscript is fully re-written.

REVIEWER 1: 6) Most of the equations in Appendix A are unreadable.  

AUTHORS: The equation are written via text editor. Perhaps, due to conversion from doc to pdf format most of them are, thus, unreadable for the Reviewer. In order to exclude from such disadvantage, the revised text contains same formulas shown, however as a single embedded into the text figure.  

REVIEWER 1: Comments on the Quality of English Language

Reading the test moves along although when facing the long paragraphs there is a feeling of do I really one to jump into this thing. There are often clumsy phrases, such as on line 15: There are scarce data on this thematic reported, which should read something like: There are few studies about this particular topic. Reading the text aloud can catch some of them, but it's pretty much impossible to correct all of them. 

AUTHORS: The English is revised. The text of the manuscript, is in fact, rewritten, due to extended revision of almost all sections.

RESPONSE TO REVIEWER 1

Comments and Suggestions for Authors

REVIEWER 1: 1) The conclusion highlights the identification of 89 metabolites from the virus infected micro-plants, but says nothing about the effect of magneto-therapy on plant recovery, a topic that was indicated on line 43. That chlorogenic acid enhances resistance is stated on line 53, but that is not the same as recovery. This discrepancy is emphasized by the control employed being virus infected microplants with no MF exposure according to the captions of Figures 1 and 2 and that the discussion section takes up the advantages of MPT but never addresses the degree of recovery after treatment by comparison with metabolites from healthy plants or virus diagnosis by ELISA mentioned in 4.2.   Since one of the aims of IJMS is "strong emphasis on molecular biology", and not just the analytical techniques needed to accomplish the goal, a clarification is needed in the introduction as to what the study intends to contribute to the plant-specific MPT field not already covered in references 6-8.   The claim on line 99 that there was a 14.5 increase in chlorogenic acid after 10 Hz treatment after 72 h relative to 0 Hz doesn't seem to be supported in Figure 1 where the concentration ratio is 1.2/0.18=6.7.  

AUTHORS: The authors would like to thank for valuable Reviewer’s recommendations and remarks. According to shown above remarks, there is added recovery value (4.79) obtained according equation (1) of new cited reference [9]. In addition, there are shown chemometric parameters of chromatographic measurements (see new Figure A1.)

The chemometric determination of curve-areas of chromatographic data yields to 8.75-fold increasing in chlorogenic acid. The value is corrected in the revised version of the text.    

[9] Burns D.; Danzer K.; Townshend A. Use of the terms “recovery” and “apparent recovery” in analytical procedures. Pure Appl. Chem. 2002, 74, 22012205.

REVIEWER 1: 2)  That it is the analytical aspect which dominates the study is further emphasized in section 2 where only 2.1 deals with metabolites evolving from the microplants and the other three sections with fragmentation patterns.  It is pretty much impossible to determine from the section what new is being contributed to fragmentation patterns. Line 232 says that reliable MS protocols for quantifying chlorogenic acid in foods aren't relevant for the study. Line 401 claims agreement with previous data. What is needed in section 3 is a summary of accomplishments of the fragmentation work, such as distinguishing between positional stereo-isomers on line 390 and new metabolites on line 25, as compared to state of the art in the area. If the therapeutic aspect is to be pursued, then how the above MS accomplishments contributed to following recovery back toward healthy status, should be included in the discussion.   

AUTHORS: The content of the paper is designed for Special Issue [https://www.mdpi.com/journal/ijms/special_issues/P4R5D7UIZM], devoted to apply ‚mass spectrum‘ to field of metabolomics and udrestand comprehensively biochemical reactions in vitro and in vivo upon effect of magnetic field treatment. Due to this reason content focuses, mainly, on capability of mass spectrometry as robust analytical method for identification, annotation, and structural analysis of metabolites, despite, the fact that the major scope of the Journal focuses on biochemistry. Owing to the fact that, there is proven benefical biological function of aucubin, and there has been achieved perturbation of metabolism of chlorogenic acid upon external magnetic field treatment the focus of the old version of the manuscript was on mass spectrometric data on these species. However, from perspective of novelty, the fragmentation pattern of clorogenic acid is comprehensively examined by many authors. In fact, there is an in-depth analysis of fragmentation paths of the latter analyte (please, consider Table 1 in reference [59].) Particularly, the latter study looks at isomerism of clorogfenic acid. Also, MS data on stereoisomers of clorogenic acid have been in-depth discussed, in fact, in all cited studies by authors such as, for example [47-64]. Moreover, these are not fully representative references to the issue, because of our work does not represent exhaustive review-article. We only confirm known data on chlorogenic acid and aucubin via independent innovative mass spectrometric method based on stochastic dynamics (please, consider equations (1)-(6) and new sub-section 4.5.1.) The latter task aims at proving their validity and their adequate application to annotate unknown species, having comparable chemometric parameters using routine searching algorithms for data processing of mass spectrometric variables of metabolites and their assignment according to library parameters of mass spectra. For instance, these are carbohydrates. Thus, from perspective of novelty of fragmentation pattern there should be highlighted our analysis of salicylated carbohydrates, which mass spectrometric based annotation does not respresent trivial task due to comparable method performances. Please, see new sub section 2.4. Mass spectrometric data on silylated carbohydrate monomers and new Figures A2–A16. There are shown experimental chromatographic and mass spectrometric data on silylated carbohydrates assigned comprehensively via two independent theoretical methods, thus fitting content of the study to thematic issue of the Journal.

[59] Xie C.; Yu K.; Zhong D.; Yuan, T.; Ye, F.; Jarrell J.; Millar A.; Chen, X. Investigation of isomeric transformations of chlorogenic acid in buffers and biological matrixes by ultraperformance liquid chromatography coupled with hybrid quadrupole/ion mobility/orthogonal acceleration time-of-flight mass spectrometry. J. Agric. Food Chem. 2011, 59, 11078–11087.

REVIEWER 1: 3)  Discussing references dealing with how magnetism affects Pyrus communis L. might strengthen the plant development topic, such as T Cebulak et al.,  Effect of abiotic stress factors on polyphenolic content in the skin and flesh of pear by UPLC‑PDA‑Q/TOF‑MS, European Food Research and Technology 245 (2019) 245 2715-2725, and A El-Nasr et al., Effect of Magnetite Nanoparticles (Fe3O4) as Nutritive Supplement on Pear Saplings, Middle East Journal of Applied Sciences 5 (2015) 777-785.  

AUTHORS: There is presented new section ‚Discussion‘ of revised paper. It reflects Reviewer’s remarks and aforementioned contributions to other authors. Please, consider highlighted text of section 3. Discussion and new cited works [85–96]. The revised section ‚Discussion‘, we hope illustrates more clearly for Editors, Reviewers and potential Readers the impact and novelty of our study accounting not only for scope of special issue and mass spectrometric data, but also biochemical aspects of our results from metabolomics.  

REVIEWER 1: 4) Experimental procedure with respect to silyation in 4.1 and 4.2 needs clarification. Section 4.1 says samples for 3 and 72 h analysis were extracted using 70% ethanol and silylated using BSATF. The procedure in 4.2 was extraction in methanol with silylation mentioned on line 484. Which samples and what analyses are associated with each procedure?  

AUTHORS: Section ‚Experimental‘ of revised text is completely re-written. It details on synthesis of sylilated carbohydrates; experimental design of magnetic measurements of plants together with experimental parameters: detail on chromatographic measurements and annotation; advantages and limitation of GC-MS and LC-MS methods which are often used to purposes of metabolomics; brief description of strochastic dynamic theory, quantum chemical computations, and chemometrics, respectively. Relevant reference section is added, as well.   

REVIEWER 1: 5) The long paragraphs need to be broken down into smaller ones. Section 3, for example, consists of only one.  

AUTHORS: The text of the manuscript is fully re-written.

REVIEWER 1: 6) Most of the equations in Appendix A are unreadable.  

AUTHORS: The equation are written via text editor. Perhaps, due to conversion from doc to pdf format most of them are, thus, unreadable for the Reviewer. In order to exclude from such disadvantage, the revised text contains same formulas shown, however as a single embedded into the text figure.  

REVIEWER 1: Comments on the Quality of English Language

Reading the test moves along although when facing the long paragraphs there is a feeling of do I really one to jump into this thing. There are often clumsy phrases, such as on line 15: There are scarce data on this thematic reported, which should read something like: There are few studies about this particular topic. Reading the text aloud can catch some of them, but it's pretty much impossible to correct all of them. 

AUTHORS: The English is revised. The text of the manuscript, is in fact, rewritten, due to extended revision of almost all sections.

RESPONSE TO REVIEWER 1

Comments and Suggestions for Authors

REVIEWER 1: 1) The conclusion highlights the identification of 89 metabolites from the virus infected micro-plants, but says nothing about the effect of magneto-therapy on plant recovery, a topic that was indicated on line 43. That chlorogenic acid enhances resistance is stated on line 53, but that is not the same as recovery. This discrepancy is emphasized by the control employed being virus infected microplants with no MF exposure according to the captions of Figures 1 and 2 and that the discussion section takes up the advantages of MPT but never addresses the degree of recovery after treatment by comparison with metabolites from healthy plants or virus diagnosis by ELISA mentioned in 4.2.   Since one of the aims of IJMS is "strong emphasis on molecular biology", and not just the analytical techniques needed to accomplish the goal, a clarification is needed in the introduction as to what the study intends to contribute to the plant-specific MPT field not already covered in references 6-8.   The claim on line 99 that there was a 14.5 increase in chlorogenic acid after 10 Hz treatment after 72 h relative to 0 Hz doesn't seem to be supported in Figure 1 where the concentration ratio is 1.2/0.18=6.7.  

AUTHORS: The authors would like to thank for valuable Reviewer’s recommendations and remarks. According to shown above remarks, there is added recovery value (4.79) obtained according equation (1) of new cited reference [9]. In addition, there are shown chemometric parameters of chromatographic measurements (see new Figure A1.)

The chemometric determination of curve-areas of chromatographic data yields to 8.75-fold increasing in chlorogenic acid. The value is corrected in the revised version of the text.    

[9] Burns D.; Danzer K.; Townshend A. Use of the terms “recovery” and “apparent recovery” in analytical procedures. Pure Appl. Chem. 2002, 74, 22012205.

REVIEWER 1: 2)  That it is the analytical aspect which dominates the study is further emphasized in section 2 where only 2.1 deals with metabolites evolving from the microplants and the other three sections with fragmentation patterns.  It is pretty much impossible to determine from the section what new is being contributed to fragmentation patterns. Line 232 says that reliable MS protocols for quantifying chlorogenic acid in foods aren't relevant for the study. Line 401 claims agreement with previous data. What is needed in section 3 is a summary of accomplishments of the fragmentation work, such as distinguishing between positional stereo-isomers on line 390 and new metabolites on line 25, as compared to state of the art in the area. If the therapeutic aspect is to be pursued, then how the above MS accomplishments contributed to following recovery back toward healthy status, should be included in the discussion.   

AUTHORS: The content of the paper is designed for Special Issue [https://www.mdpi.com/journal/ijms/special_issues/P4R5D7UIZM], devoted to apply ‚mass spectrum‘ to field of metabolomics and udrestand comprehensively biochemical reactions in vitro and in vivo upon effect of magnetic field treatment. Due to this reason content focuses, mainly, on capability of mass spectrometry as robust analytical method for identification, annotation, and structural analysis of metabolites, despite, the fact that the major scope of the Journal focuses on biochemistry. Owing to the fact that, there is proven benefical biological function of aucubin, and there has been achieved perturbation of metabolism of chlorogenic acid upon external magnetic field treatment the focus of the old version of the manuscript was on mass spectrometric data on these species. However, from perspective of novelty, the fragmentation pattern of clorogenic acid is comprehensively examined by many authors. In fact, there is an in-depth analysis of fragmentation paths of the latter analyte (please, consider Table 1 in reference [59].) Particularly, the latter study looks at isomerism of clorogfenic acid. Also, MS data on stereoisomers of clorogenic acid have been in-depth discussed, in fact, in all cited studies by authors such as, for example [47-64]. Moreover, these are not fully representative references to the issue, because of our work does not represent exhaustive review-article. We only confirm known data on chlorogenic acid and aucubin via independent innovative mass spectrometric method based on stochastic dynamics (please, consider equations (1)-(6) and new sub-section 4.5.1.) The latter task aims at proving their validity and their adequate application to annotate unknown species, having comparable chemometric parameters using routine searching algorithms for data processing of mass spectrometric variables of metabolites and their assignment according to library parameters of mass spectra. For instance, these are carbohydrates. Thus, from perspective of novelty of fragmentation pattern there should be highlighted our analysis of salicylated carbohydrates, which mass spectrometric based annotation does not respresent trivial task due to comparable method performances. Please, see new sub section 2.4. Mass spectrometric data on silylated carbohydrate monomers and new Figures A2–A16. There are shown experimental chromatographic and mass spectrometric data on silylated carbohydrates assigned comprehensively via two independent theoretical methods, thus fitting content of the study to thematic issue of the Journal.

[59] Xie C.; Yu K.; Zhong D.; Yuan, T.; Ye, F.; Jarrell J.; Millar A.; Chen, X. Investigation of isomeric transformations of chlorogenic acid in buffers and biological matrixes by ultraperformance liquid chromatography coupled with hybrid quadrupole/ion mobility/orthogonal acceleration time-of-flight mass spectrometry. J. Agric. Food Chem. 2011, 59, 11078–11087.

REVIEWER 1: 3)  Discussing references dealing with how magnetism affects Pyrus communis L. might strengthen the plant development topic, such as T Cebulak et al.,  Effect of abiotic stress factors on polyphenolic content in the skin and flesh of pear by UPLC‑PDA‑Q/TOF‑MS, European Food Research and Technology 245 (2019) 245 2715-2725, and A El-Nasr et al., Effect of Magnetite Nanoparticles (Fe3O4) as Nutritive Supplement on Pear Saplings, Middle East Journal of Applied Sciences 5 (2015) 777-785.  

AUTHORS: There is presented new section ‚Discussion‘ of revised paper. It reflects Reviewer’s remarks and aforementioned contributions to other authors. Please, consider highlighted text of section 3. Discussion and new cited works [85–96]. The revised section ‚Discussion‘, we hope illustrates more clearly for Editors, Reviewers and potential Readers the impact and novelty of our study accounting not only for scope of special issue and mass spectrometric data, but also biochemical aspects of our results from metabolomics.  

REVIEWER 1: 4) Experimental procedure with respect to silyation in 4.1 and 4.2 needs clarification. Section 4.1 says samples for 3 and 72 h analysis were extracted using 70% ethanol and silylated using BSATF. The procedure in 4.2 was extraction in methanol with silylation mentioned on line 484. Which samples and what analyses are associated with each procedure?  

AUTHORS: Section ‚Experimental‘ of revised text is completely re-written. It details on synthesis of sylilated carbohydrates; experimental design of magnetic measurements of plants together with experimental parameters: detail on chromatographic measurements and annotation; advantages and limitation of GC-MS and LC-MS methods which are often used to purposes of metabolomics; brief description of strochastic dynamic theory, quantum chemical computations, and chemometrics, respectively. Relevant reference section is added, as well.   

REVIEWER 1: 5) The long paragraphs need to be broken down into smaller ones. Section 3, for example, consists of only one.  

AUTHORS: The text of the manuscript is fully re-written.

REVIEWER 1: 6) Most of the equations in Appendix A are unreadable.  

AUTHORS: The equation are written via text editor. Perhaps, due to conversion from doc to pdf format most of them are, thus, unreadable for the Reviewer. In order to exclude from such disadvantage, the revised text contains same formulas shown, however as a single embedded into the text figure.  

REVIEWER 1: Comments on the Quality of English Language

Reading the test moves along although when facing the long paragraphs there is a feeling of do I really one to jump into this thing. There are often clumsy phrases, such as on line 15: There are scarce data on this thematic reported, which should read something like: There are few studies about this particular topic. Reading the text aloud can catch some of them, but it's pretty much impossible to correct all of them. 

AUTHORS: The English is revised. The text of the manuscript, is in fact, rewritten, due to extended revision of almost all sections.

RESPONSE TO REVIEWER 1

Comments and Suggestions for Authors

REVIEWER 1: 1) The conclusion highlights the identification of 89 metabolites from the virus infected micro-plants, but says nothing about the effect of magneto-therapy on plant recovery, a topic that was indicated on line 43. That chlorogenic acid enhances resistance is stated on line 53, but that is not the same as recovery. This discrepancy is emphasized by the control employed being virus infected microplants with no MF exposure according to the captions of Figures 1 and 2 and that the discussion section takes up the advantages of MPT but never addresses the degree of recovery after treatment by comparison with metabolites from healthy plants or virus diagnosis by ELISA mentioned in 4.2.   Since one of the aims of IJMS is "strong emphasis on molecular biology", and not just the analytical techniques needed to accomplish the goal, a clarification is needed in the introduction as to what the study intends to contribute to the plant-specific MPT field not already covered in references 6-8.   The claim on line 99 that there was a 14.5 increase in chlorogenic acid after 10 Hz treatment after 72 h relative to 0 Hz doesn't seem to be supported in Figure 1 where the concentration ratio is 1.2/0.18=6.7.  

AUTHORS: The authors would like to thank for valuable Reviewer’s recommendations and remarks. According to shown above remarks, there is added recovery value (4.79) obtained according equation (1) of new cited reference [9]. In addition, there are shown chemometric parameters of chromatographic measurements (see new Figure A1.)

The chemometric determination of curve-areas of chromatographic data yields to 8.75-fold increasing in chlorogenic acid. The value is corrected in the revised version of the text.    

[9] Burns D.; Danzer K.; Townshend A. Use of the terms “recovery” and “apparent recovery” in analytical procedures. Pure Appl. Chem. 2002, 74, 22012205.

REVIEWER 1: 2)  That it is the analytical aspect which dominates the study is further emphasized in section 2 where only 2.1 deals with metabolites evolving from the microplants and the other three sections with fragmentation patterns.  It is pretty much impossible to determine from the section what new is being contributed to fragmentation patterns. Line 232 says that reliable MS protocols for quantifying chlorogenic acid in foods aren't relevant for the study. Line 401 claims agreement with previous data. What is needed in section 3 is a summary of accomplishments of the fragmentation work, such as distinguishing between positional stereo-isomers on line 390 and new metabolites on line 25, as compared to state of the art in the area. If the therapeutic aspect is to be pursued, then how the above MS accomplishments contributed to following recovery back toward healthy status, should be included in the discussion.   

AUTHORS: The content of the paper is designed for Special Issue [https://www.mdpi.com/journal/ijms/special_issues/P4R5D7UIZM], devoted to apply ‚mass spectrum‘ to field of metabolomics and udrestand comprehensively biochemical reactions in vitro and in vivo upon effect of magnetic field treatment. Due to this reason content focuses, mainly, on capability of mass spectrometry as robust analytical method for identification, annotation, and structural analysis of metabolites, despite, the fact that the major scope of the Journal focuses on biochemistry. Owing to the fact that, there is proven benefical biological function of aucubin, and there has been achieved perturbation of metabolism of chlorogenic acid upon external magnetic field treatment the focus of the old version of the manuscript was on mass spectrometric data on these species. However, from perspective of novelty, the fragmentation pattern of clorogenic acid is comprehensively examined by many authors. In fact, there is an in-depth analysis of fragmentation paths of the latter analyte (please, consider Table 1 in reference [59].) Particularly, the latter study looks at isomerism of clorogfenic acid. Also, MS data on stereoisomers of clorogenic acid have been in-depth discussed, in fact, in all cited studies by authors such as, for example [47-64]. Moreover, these are not fully representative references to the issue, because of our work does not represent exhaustive review-article. We only confirm known data on chlorogenic acid and aucubin via independent innovative mass spectrometric method based on stochastic dynamics (please, consider equations (1)-(6) and new sub-section 4.5.1.) The latter task aims at proving their validity and their adequate application to annotate unknown species, having comparable chemometric parameters using routine searching algorithms for data processing of mass spectrometric variables of metabolites and their assignment according to library parameters of mass spectra. For instance, these are carbohydrates. Thus, from perspective of novelty of fragmentation pattern there should be highlighted our analysis of salicylated carbohydrates, which mass spectrometric based annotation does not respresent trivial task due to comparable method performances. Please, see new sub section 2.4. Mass spectrometric data on silylated carbohydrate monomers and new Figures A2–A16. There are shown experimental chromatographic and mass spectrometric data on silylated carbohydrates assigned comprehensively via two independent theoretical methods, thus fitting content of the study to thematic issue of the Journal.

[59] Xie C.; Yu K.; Zhong D.; Yuan, T.; Ye, F.; Jarrell J.; Millar A.; Chen, X. Investigation of isomeric transformations of chlorogenic acid in buffers and biological matrixes by ultraperformance liquid chromatography coupled with hybrid quadrupole/ion mobility/orthogonal acceleration time-of-flight mass spectrometry. J. Agric. Food Chem. 2011, 59, 11078–11087.

REVIEWER 1: 3)  Discussing references dealing with how magnetism affects Pyrus communis L. might strengthen the plant development topic, such as T Cebulak et al.,  Effect of abiotic stress factors on polyphenolic content in the skin and flesh of pear by UPLC‑PDA‑Q/TOF‑MS, European Food Research and Technology 245 (2019) 245 2715-2725, and A El-Nasr et al., Effect of Magnetite Nanoparticles (Fe3O4) as Nutritive Supplement on Pear Saplings, Middle East Journal of Applied Sciences 5 (2015) 777-785.  

AUTHORS: There is presented new section ‚Discussion‘ of revised paper. It reflects Reviewer’s remarks and aforementioned contributions to other authors. Please, consider highlighted text of section 3. Discussion and new cited works [85–96]. The revised section ‚Discussion‘, we hope illustrates more clearly for Editors, Reviewers and potential Readers the impact and novelty of our study accounting not only for scope of special issue and mass spectrometric data, but also biochemical aspects of our results from metabolomics.  

REVIEWER 1: 4) Experimental procedure with respect to silyation in 4.1 and 4.2 needs clarification. Section 4.1 says samples for 3 and 72 h analysis were extracted using 70% ethanol and silylated using BSATF. The procedure in 4.2 was extraction in methanol with silylation mentioned on line 484. Which samples and what analyses are associated with each procedure?  

AUTHORS: Section ‚Experimental‘ of revised text is completely re-written. It details on synthesis of sylilated carbohydrates; experimental design of magnetic measurements of plants together with experimental parameters: detail on chromatographic measurements and annotation; advantages and limitation of GC-MS and LC-MS methods which are often used to purposes of metabolomics; brief description of strochastic dynamic theory, quantum chemical computations, and chemometrics, respectively. Relevant reference section is added, as well.   

REVIEWER 1: 5) The long paragraphs need to be broken down into smaller ones. Section 3, for example, consists of only one.  

AUTHORS: The text of the manuscript is fully re-written.

REVIEWER 1: 6) Most of the equations in Appendix A are unreadable.  

AUTHORS: The equation are written via text editor. Perhaps, due to conversion from doc to pdf format most of them are, thus, unreadable for the Reviewer. In order to exclude from such disadvantage, the revised text contains same formulas shown, however as a single embedded into the text figure.  

REVIEWER 1: Comments on the Quality of English Language

Reading the test moves along although when facing the long paragraphs there is a feeling of do I really one to jump into this thing. There are often clumsy phrases, such as on line 15: There are scarce data on this thematic reported, which should read something like: There are few studies about this particular topic. Reading the text aloud can catch some of them, but it's pretty much impossible to correct all of them. 

AUTHORS: The English is revised. The text of the manuscript, is in fact, rewritten, due to extended revision of almost all sections.

RESPONSE TO REVIEWER 1

Comments and Suggestions for Authors

REVIEWER 1: 1) The conclusion highlights the identification of 89 metabolites from the virus infected micro-plants, but says nothing about the effect of magneto-therapy on plant recovery, a topic that was indicated on line 43. That chlorogenic acid enhances resistance is stated on line 53, but that is not the same as recovery. This discrepancy is emphasized by the control employed being virus infected microplants with no MF exposure according to the captions of Figures 1 and 2 and that the discussion section takes up the advantages of MPT but never addresses the degree of recovery after treatment by comparison with metabolites from healthy plants or virus diagnosis by ELISA mentioned in 4.2.   Since one of the aims of IJMS is "strong emphasis on molecular biology", and not just the analytical techniques needed to accomplish the goal, a clarification is needed in the introduction as to what the study intends to contribute to the plant-specific MPT field not already covered in references 6-8.   The claim on line 99 that there was a 14.5 increase in chlorogenic acid after 10 Hz treatment after 72 h relative to 0 Hz doesn't seem to be supported in Figure 1 where the concentration ratio is 1.2/0.18=6.7.  

AUTHORS: The authors would like to thank for valuable Reviewer’s recommendations and remarks. According to shown above remarks, there is added recovery value (4.79) obtained according equation (1) of new cited reference [9]. In addition, there are shown chemometric parameters of chromatographic measurements (see new Figure A1.)

The chemometric determination of curve-areas of chromatographic data yields to 8.75-fold increasing in chlorogenic acid. The value is corrected in the revised version of the text.    

[9] Burns D.; Danzer K.; Townshend A. Use of the terms “recovery” and “apparent recovery” in analytical procedures. Pure Appl. Chem. 2002, 74, 22012205.

REVIEWER 1: 2)  That it is the analytical aspect which dominates the study is further emphasized in section 2 where only 2.1 deals with metabolites evolving from the microplants and the other three sections with fragmentation patterns.  It is pretty much impossible to determine from the section what new is being contributed to fragmentation patterns. Line 232 says that reliable MS protocols for quantifying chlorogenic acid in foods aren't relevant for the study. Line 401 claims agreement with previous data. What is needed in section 3 is a summary of accomplishments of the fragmentation work, such as distinguishing between positional stereo-isomers on line 390 and new metabolites on line 25, as compared to state of the art in the area. If the therapeutic aspect is to be pursued, then how the above MS accomplishments contributed to following recovery back toward healthy status, should be included in the discussion.   

AUTHORS: The content of the paper is designed for Special Issue [https://www.mdpi.com/journal/ijms/special_issues/P4R5D7UIZM], devoted to apply ‚mass spectrum‘ to field of metabolomics and udrestand comprehensively biochemical reactions in vitro and in vivo upon effect of magnetic field treatment. Due to this reason content focuses, mainly, on capability of mass spectrometry as robust analytical method for identification, annotation, and structural analysis of metabolites, despite, the fact that the major scope of the Journal focuses on biochemistry. Owing to the fact that, there is proven benefical biological function of aucubin, and there has been achieved perturbation of metabolism of chlorogenic acid upon external magnetic field treatment the focus of the old version of the manuscript was on mass spectrometric data on these species. However, from perspective of novelty, the fragmentation pattern of clorogenic acid is comprehensively examined by many authors. In fact, there is an in-depth analysis of fragmentation paths of the latter analyte (please, consider Table 1 in reference [59].) Particularly, the latter study looks at isomerism of clorogfenic acid. Also, MS data on stereoisomers of clorogenic acid have been in-depth discussed, in fact, in all cited studies by authors such as, for example [47-64]. Moreover, these are not fully representative references to the issue, because of our work does not represent exhaustive review-article. We only confirm known data on chlorogenic acid and aucubin via independent innovative mass spectrometric method based on stochastic dynamics (please, consider equations (1)-(6) and new sub-section 4.5.1.) The latter task aims at proving their validity and their adequate application to annotate unknown species, having comparable chemometric parameters using routine searching algorithms for data processing of mass spectrometric variables of metabolites and their assignment according to library parameters of mass spectra. For instance, these are carbohydrates. Thus, from perspective of novelty of fragmentation pattern there should be highlighted our analysis of salicylated carbohydrates, which mass spectrometric based annotation does not respresent trivial task due to comparable method performances. Please, see new sub section 2.4. Mass spectrometric data on silylated carbohydrate monomers and new Figures A2–A16. There are shown experimental chromatographic and mass spectrometric data on silylated carbohydrates assigned comprehensively via two independent theoretical methods, thus fitting content of the study to thematic issue of the Journal.

[59] Xie C.; Yu K.; Zhong D.; Yuan, T.; Ye, F.; Jarrell J.; Millar A.; Chen, X. Investigation of isomeric transformations of chlorogenic acid in buffers and biological matrixes by ultraperformance liquid chromatography coupled with hybrid quadrupole/ion mobility/orthogonal acceleration time-of-flight mass spectrometry. J. Agric. Food Chem. 2011, 59, 11078–11087.

REVIEWER 1: 3)  Discussing references dealing with how magnetism affects Pyrus communis L. might strengthen the plant development topic, such as T Cebulak et al.,  Effect of abiotic stress factors on polyphenolic content in the skin and flesh of pear by UPLC‑PDA‑Q/TOF‑MS, European Food Research and Technology 245 (2019) 245 2715-2725, and A El-Nasr et al., Effect of Magnetite Nanoparticles (Fe3O4) as Nutritive Supplement on Pear Saplings, Middle East Journal of Applied Sciences 5 (2015) 777-785.  

AUTHORS: There is presented new section ‚Discussion‘ of revised paper. It reflects Reviewer’s remarks and aforementioned contributions to other authors. Please, consider highlighted text of section 3. Discussion and new cited works [85–96]. The revised section ‚Discussion‘, we hope illustrates more clearly for Editors, Reviewers and potential Readers the impact and novelty of our study accounting not only for scope of special issue and mass spectrometric data, but also biochemical aspects of our results from metabolomics.  

REVIEWER 1: 4) Experimental procedure with respect to silyation in 4.1 and 4.2 needs clarification. Section 4.1 says samples for 3 and 72 h analysis were extracted using 70% ethanol and silylated using BSATF. The procedure in 4.2 was extraction in methanol with silylation mentioned on line 484. Which samples and what analyses are associated with each procedure?  

AUTHORS: Section ‚Experimental‘ of revised text is completely re-written. It details on synthesis of sylilated carbohydrates; experimental design of magnetic measurements of plants together with experimental parameters: detail on chromatographic measurements and annotation; advantages and limitation of GC-MS and LC-MS methods which are often used to purposes of metabolomics; brief description of strochastic dynamic theory, quantum chemical computations, and chemometrics, respectively. Relevant reference section is added, as well.   

REVIEWER 1: 5) The long paragraphs need to be broken down into smaller ones. Section 3, for example, consists of only one.  

AUTHORS: The text of the manuscript is fully re-written.

REVIEWER 1: 6) Most of the equations in Appendix A are unreadable.  

AUTHORS: The equation are written via text editor. Perhaps, due to conversion from doc to pdf format most of them are, thus, unreadable for the Reviewer. In order to exclude from such disadvantage, the revised text contains same formulas shown, however as a single embedded into the text figure.  

REVIEWER 1: Comments on the Quality of English Language

Reading the test moves along although when facing the long paragraphs there is a feeling of do I really one to jump into this thing. There are often clumsy phrases, such as on line 15: There are scarce data on this thematic reported, which should read something like: There are few studies about this particular topic. Reading the text aloud can catch some of them, but it's pretty much impossible to correct all of them. 

AUTHORS: The English is revised. The text of the manuscript, is in fact, rewritten, due to extended revision of almost all sections.

RESPONSE TO REVIEWER 1

Comments and Suggestions for Authors

REVIEWER 1: 1) The conclusion highlights the identification of 89 metabolites from the virus infected micro-plants, but says nothing about the effect of magneto-therapy on plant recovery, a topic that was indicated on line 43. That chlorogenic acid enhances resistance is stated on line 53, but that is not the same as recovery. This discrepancy is emphasized by the control employed being virus infected microplants with no MF exposure according to the captions of Figures 1 and 2 and that the discussion section takes up the advantages of MPT but never addresses the degree of recovery after treatment by comparison with metabolites from healthy plants or virus diagnosis by ELISA mentioned in 4.2.   Since one of the aims of IJMS is "strong emphasis on molecular biology", and not just the analytical techniques needed to accomplish the goal, a clarification is needed in the introduction as to what the study intends to contribute to the plant-specific MPT field not already covered in references 6-8.   The claim on line 99 that there was a 14.5 increase in chlorogenic acid after 10 Hz treatment after 72 h relative to 0 Hz doesn't seem to be supported in Figure 1 where the concentration ratio is 1.2/0.18=6.7.  

AUTHORS: The authors would like to thank for valuable Reviewer’s recommendations and remarks. According to shown above remarks, there is added recovery value (4.79) obtained according equation (1) of new cited reference [9]. In addition, there are shown chemometric parameters of chromatographic measurements (see new Figure A1.)

The chemometric determination of curve-areas of chromatographic data yields to 8.75-fold increasing in chlorogenic acid. The value is corrected in the revised version of the text.    

[9] Burns D.; Danzer K.; Townshend A. Use of the terms “recovery” and “apparent recovery” in analytical procedures. Pure Appl. Chem. 2002, 74, 22012205.

REVIEWER 1: 2)  That it is the analytical aspect which dominates the study is further emphasized in section 2 where only 2.1 deals with metabolites evolving from the microplants and the other three sections with fragmentation patterns.  It is pretty much impossible to determine from the section what new is being contributed to fragmentation patterns. Line 232 says that reliable MS protocols for quantifying chlorogenic acid in foods aren't relevant for the study. Line 401 claims agreement with previous data. What is needed in section 3 is a summary of accomplishments of the fragmentation work, such as distinguishing between positional stereo-isomers on line 390 and new metabolites on line 25, as compared to state of the art in the area. If the therapeutic aspect is to be pursued, then how the above MS accomplishments contributed to following recovery back toward healthy status, should be included in the discussion.   

AUTHORS: The content of the paper is designed for Special Issue [https://www.mdpi.com/journal/ijms/special_issues/P4R5D7UIZM], devoted to apply ‚mass spectrum‘ to field of metabolomics and udrestand comprehensively biochemical reactions in vitro and in vivo upon effect of magnetic field treatment. Due to this reason content focuses, mainly, on capability of mass spectrometry as robust analytical method for identification, annotation, and structural analysis of metabolites, despite, the fact that the major scope of the Journal focuses on biochemistry. Owing to the fact that, there is proven benefical biological function of aucubin, and there has been achieved perturbation of metabolism of chlorogenic acid upon external magnetic field treatment the focus of the old version of the manuscript was on mass spectrometric data on these species. However, from perspective of novelty, the fragmentation pattern of clorogenic acid is comprehensively examined by many authors. In fact, there is an in-depth analysis of fragmentation paths of the latter analyte (please, consider Table 1 in reference [59].) Particularly, the latter study looks at isomerism of clorogfenic acid. Also, MS data on stereoisomers of clorogenic acid have been in-depth discussed, in fact, in all cited studies by authors such as, for example [47-64]. Moreover, these are not fully representative references to the issue, because of our work does not represent exhaustive review-article. We only confirm known data on chlorogenic acid and aucubin via independent innovative mass spectrometric method based on stochastic dynamics (please, consider equations (1)-(6) and new sub-section 4.5.1.) The latter task aims at proving their validity and their adequate application to annotate unknown species, having comparable chemometric parameters using routine searching algorithms for data processing of mass spectrometric variables of metabolites and their assignment according to library parameters of mass spectra. For instance, these are carbohydrates. Thus, from perspective of novelty of fragmentation pattern there should be highlighted our analysis of salicylated carbohydrates, which mass spectrometric based annotation does not respresent trivial task due to comparable method performances. Please, see new sub section 2.4. Mass spectrometric data on silylated carbohydrate monomers and new Figures A2–A16. There are shown experimental chromatographic and mass spectrometric data on silylated carbohydrates assigned comprehensively via two independent theoretical methods, thus fitting content of the study to thematic issue of the Journal.

[59] Xie C.; Yu K.; Zhong D.; Yuan, T.; Ye, F.; Jarrell J.; Millar A.; Chen, X. Investigation of isomeric transformations of chlorogenic acid in buffers and biological matrixes by ultraperformance liquid chromatography coupled with hybrid quadrupole/ion mobility/orthogonal acceleration time-of-flight mass spectrometry. J. Agric. Food Chem. 2011, 59, 11078–11087.

REVIEWER 1: 3)  Discussing references dealing with how magnetism affects Pyrus communis L. might strengthen the plant development topic, such as T Cebulak et al.,  Effect of abiotic stress factors on polyphenolic content in the skin and flesh of pear by UPLC‑PDA‑Q/TOF‑MS, European Food Research and Technology 245 (2019) 245 2715-2725, and A El-Nasr et al., Effect of Magnetite Nanoparticles (Fe3O4) as Nutritive Supplement on Pear Saplings, Middle East Journal of Applied Sciences 5 (2015) 777-785.  

AUTHORS: There is presented new section ‚Discussion‘ of revised paper. It reflects Reviewer’s remarks and aforementioned contributions to other authors. Please, consider highlighted text of section 3. Discussion and new cited works [85–96]. The revised section ‚Discussion‘, we hope illustrates more clearly for Editors, Reviewers and potential Readers the impact and novelty of our study accounting not only for scope of special issue and mass spectrometric data, but also biochemical aspects of our results from metabolomics.  

REVIEWER 1: 4) Experimental procedure with respect to silyation in 4.1 and 4.2 needs clarification. Section 4.1 says samples for 3 and 72 h analysis were extracted using 70% ethanol and silylated using BSATF. The procedure in 4.2 was extraction in methanol with silylation mentioned on line 484. Which samples and what analyses are associated with each procedure?  

AUTHORS: Section ‚Experimental‘ of revised text is completely re-written. It details on synthesis of sylilated carbohydrates; experimental design of magnetic measurements of plants together with experimental parameters: detail on chromatographic measurements and annotation; advantages and limitation of GC-MS and LC-MS methods which are often used to purposes of metabolomics; brief description of strochastic dynamic theory, quantum chemical computations, and chemometrics, respectively. Relevant reference section is added, as well.   

REVIEWER 1: 5) The long paragraphs need to be broken down into smaller ones. Section 3, for example, consists of only one.  

AUTHORS: The text of the manuscript is fully re-written.

REVIEWER 1: 6) Most of the equations in Appendix A are unreadable.  

AUTHORS: The equation are written via text editor. Perhaps, due to conversion from doc to pdf format most of them are, thus, unreadable for the Reviewer. In order to exclude from such disadvantage, the revised text contains same formulas shown, however as a single embedded into the text figure.  

REVIEWER 1: Comments on the Quality of English Language

Reading the test moves along although when facing the long paragraphs there is a feeling of do I really one to jump into this thing. There are often clumsy phrases, such as on line 15: There are scarce data on this thematic reported, which should read something like: There are few studies about this particular topic. Reading the text aloud can catch some of them, but it's pretty much impossible to correct all of them. 

AUTHORS: The English is revised. The text of the manuscript, is in fact, rewritten, due to extended revision of almost all sections.

ritten, due to extended revision of almost all sections.

Reviewer 2 Report

Comments and Suggestions for Authors

A brief summary 

The article is devoted to the study of the effect of magnetic pulses with a frequency of 1-10 and 51 Hz on micro-plants of the pear Pyrus communis on the background of tissue infection with ACLSV and ASGV viruses. Metabolites were determined 3 hours and 72 hours after irradiation with magnetic pulses. Dozens of metabolites have been found. Magnetic-pulse treatment leads to a 14.5-fold increase of chlorogenic acid concentration in tissues and change phenolic composition of micro-plants.

Broad comments 

Abstract, Introduction and Conclusions are written more or less normally throughout the article. The rest of the text is overloaded with a huge amount of small and specific information, which, against the background of a large number of abbreviations and specific designations, is not perceived. All formulas in the “Stochastic dynamics mass spectrometric approach” section cannot be deciphered. An A3 table 10 sheets long is not suitable even for supporting matherials.

There is absolutely no information about the applied magnetic irradiation (except frequency).

It's better to write the article completely over again.

Specific comments 

When an abbreviation first appears, it must be deciphered:

Line 36 - MF

Line 58 – MPT

Line 62 - LMW

Line 73 – MS

Line 85 – MIO

Line 118 - MFT     and so on

Table 3 – DoF - ?

Title of Table 3 - mg.g-1 - what is the designation mg.g-1 separated by a dot?   mg/g ?

Author Response

RESPONSE TO REVIEWER 2

The authors would like to thank for the useful and creative recommendations by the Reviewer. They are considered in depth in the revised text of the study.

Comments and Suggestions for Authors

A brief summary 

The article is devoted to the study of the effect of magnetic pulses with a frequency of 1-10 and 51 Hz on micro-plants of the pear Pyrus communis on the background of tissue infection with ACLSV and ASGV viruses. Metabolites were determined 3 hours and 72 hours after irradiation with magnetic pulses. Dozens of metabolites have been found. Magnetic-pulse treatment leads to a 14.5-fold increase of chlorogenic acid concentration in tissues and change phenolic composition of micro-plants.

Broad comments 

REVIEWER 2: Abstract, Introduction and Conclusions are written more or less normally throughout the article. The rest of the text is overloaded with a huge amount of small and specific information, which, against the background of a large number of abbreviations and specific designations, is not perceived.

AUTHORS: The text of the manuscript is in fact re-written. The revised version is associated with all sections.

The number of abbreviations is reduced, significantly.

All formulas in the “Stochastic dynamics mass spectrometric approach” section cannot be deciphered.

AUTHORS: The formulas are written via text editor implemented into WORD. As a result from conversion from doc to pdf format of the paper, most probably, they are unreadable for the Reviewer. Their deciphering is given in revised sub-section ‘4.5.1. Stochastic dynamics mass spectrometric approach, briefly.’

   An A3 table 10 sheets long is not suitable even for supporting matherials.

AUTHORS: Table A3, containing theoretical frequency analysis in ground and transition states of species is removed from revised version of supporting information file.

There is absolutely no information about the applied magnetic irradiation (except frequency).

It's better to write the article completely over again.

AUTHORS: The revised manuscript contains, new sub-section ‘4.4. Magnetic pulse analysis of plant biochemistry’, focusing the Readers’ attention on experimental task of magnetic pulse treatment of plants, in addition to experimental parameters of measurements. Please, consider data on new Table 1. 

Specific comments 

When an abbreviation first appears, it must be deciphered:

Line 36 - MF

AUTHORS: It is carried out.

Line 58 – MPT

It is performed.

Line 62 - LMW

It is performed.

Line 73 – MS

It is performed.

Line 85 – MIO

It is performed.

Line 118 - MFT     and so on

AUTHORS: It is performed.

Section ‘Abbreviations and acronyms’ is removed from the paper. There remains only most frequently used abbreviations sucha s GC-MS, and more. They are deciphered in the text. 

Table 3 – DoF - ?

AUTHORS: Data on χ2/DoF are associated with Cochran and Kolmogorov-Smirnov goodness-of-fit tests. The ‚DoF‘ denotes number of degree of freedom. It is equal to number of curve points assessed statistically minus number of data, which are not fixed. The test data allow us to evaluate reliability of fitting approach to experimental pattern, due to measurable variables. Please, look at new reference [10] cited in the revised version of the text. There is added new sentnece in the plain text in the latter context (please, consider page 7 of templated version oft he revised text; rows 164–170.)

[10] P. Mueller (Ed.) Wahrscheinlichkeitsrechnung und Mathematische Statistik, De Gruyter, Berlin, Boston (2022) pp. 1–485.

Title of Table 3 - mg.g-1 - what is the designation mg.g-1 separated by a dot?   mg/g ?

AUTHORS: Since, mg.g-1 denotes unit of measurand analyte concentration. It is derived from equation defining analyte concentration measurand. The designation ‘mg.g-1’ is identical with ‘mg/g’ one. It is ‘mg’ multiplied by ‘g’ of degree ‘-1’. The revised version of the paper uses the latter designation according to Reviewer’s recommendations.    

Reviewer 3 Report

Comments and Suggestions for Authors

Comments to Authors

 This manuscript aimed at studying the “effect of the magnetic pulse field on the biochemical synthesis of a set of metabolites of Pyrus communis L. micro-plants using metabolomics gas-chromatography mass spectrometric approach”.  

Major Corrections:

1) In Sections 2.2 and 2.3 – It is better to add the representative and labeled “mass spectrum of aucubin and chlorogenic acid” indicating their fragmentation species as well in the “supporting information”.

2) I also recommend to run the 1H NMR Spectra of aucubin and chlorogenic acid fractions to confirm their structures. It will add more interesting information in the discussion section to correlate it with the mass spectrum data.

3) The discussion part of the “research article” is more representative of the “introduction part”. I think it is better to discuss the results of the effect of MPT on the metabolism of micro-plant of Pyrus communis L. The discussion of “exposure time of MPT on the growth of Pyrus communis L and other factors affecting the synthesis and concentration of metabolites” would be more appealing and interesting to the readers.

3) In continuation of the previous comment – In “Discussion Section” it is also important to correlate the medicinal importance of metabolites formed and increase in their concentration on magnetic pulse treatment of Pyrus communis L.

Minor Corrections:

1) Abstract Please mention reference for these sentences “Plants are sensitive toward external low-frequency pulsed magnetic fields. They induce additional electric currents in tissues and change value of cell membrane potential”.

2) Discussion   In Sentence “Despite, the fact that there have been developed a large number of effective approaches to prevent plants from negative effect of viruses such as, for example, chemotherapy, thermotherapy, and more, a lot of them are characterized by drawback (consider detail on [6]”. It is better to mention those drawbacks because it is very confusing whether we have to look for reference [6] within this article or we have to read the article of this reference.

3) In discussion line 20 - the sentence “but also ist pathogens” I am not able to understand the word “ist” Please clear this statement.

This research article provides in-details both the qualitative and quantitative description of new metabolites formed under exposure to magnetic pulse treatment on pear plant by using mass spectrometric methods of metabolomics, stochastic dynamics, and chemometrics. However, I think after the revision of the above-mentioned comments it will be better suitable for publication in this journal.

Comments on the Quality of English Language

Dear Authors minor editing of english language required for this research article.

Author Response

RESPONSE TO REVIEWER 3

The authors thank the useful Reviewer’s remarks and recommendation. They are completely reflected on the revised version of the paper.

Comments and Suggestions for Authors

Comments to Authors

 This manuscript aimed at studying the “effect of the magnetic pulse field on the biochemical synthesis of a set of metabolites of Pyrus communis L. micro-plants using metabolomics gas-chromatography mass spectrometric approach”.  

Major Corrections:

1) In Sections 2.2 and 2.3 – It is better to add the representative and labeled “mass spectrum of aucubin and chlorogenic acid” indicating their fragmentation species as well in the “supporting information”.

The two analytes chlorogenic acid and aucubin are comprehensively examined by many other authors. Please, look at reference section [47-64]. Work [59] details on mass spectra of isomers of chlorogenic acid. Owing to the fact that the study is designed for Special Issue [https://www.mdpi.com/journal/ijms/special_issues/P4R5D7UIZM] devoted to application of mass spectrum to annotate metabolites in plants the text concentrates on novelty. It focuses on mass spectrometric assignment of complex metabolites, having comparable chemometric method performances using routine method of metabolomics in addition to effect of magnetic pulse treatment on biochemical reactions. There is an interdisciplinary study, focusing however not only on new results from effect of magnetic field on biosynthesis of metabolites in plants, but also on application of mass pectrometry to plant metabolomics. The novelty of our study does not effect well-known mass spectrometric fragmentation patterns of chlogogenic acid and aucubin. Rather, it employs measurands to test empirically validity of the innovative stochastic dynamics equations (1)-(6); thus allowing their reliable application to assign mass spectra of complex analytes such as carbohydrates. Please consider experimental chromatographic and mass spectrometric data on sililated carbohydrates (new sub-section 2.4 in addition to new Figures A2-A16.) Thus, we hope that it become more clear for Editors, Reviewers and potential Readers the scope and novelty of our research which paper is designed to the scope of aforementioned Special Issue.      

2) I also recommend to run the 1H NMR Spectra of aucubin and chlorogenic acid fractions to confirm their structures. It will add more interesting information in the discussion section to correlate it with the mass spectrum data.

Advantages, limitations and, applications of mass spectrometry are briefly compared with those ones of robust methods for molecular sutructural analysis; for instance, single crystal X-ray diffraction, NMR, CD-spectroscopy and more. There is added new sub-section ‚4.3. Analytical procedure for separation and identification of plant metabolites.’ A particular advantage of mass pectrometry, among others, is its capability of providing exact quantunative and 3D structural analysis at analyte concentration levels of fmol-attomol range. These levels are beyond analyte concentration limits of detection, achieved by NMR and single crystal X-ray diffraction, operating with microgram pro milliliter concentration of the compounds.

3) The discussion part of the “research article” is more representative of the “introduction part”. I think it is better to discuss the results of the effect of MPT on the metabolism of micro-plant of Pyrus communis L. The discussion of “exposure time of MPT on the growth of Pyrus communis L and other factors affecting the synthesis and concentration of metabolites” would be more appealing and interesting to the readers.

Both the introductory section and section ‚Discussion‘ of the study are fully re-written.

3) In continuation of the previous comment – In “Discussion Section” it is also important to correlate the medicinal importance of metabolites formed and increase in their concentration on magnetic pulse treatment of Pyrus communis L.

Minor Corrections:

1) Abstract – Please mention reference for these sentences “Plants are sensitive toward external low-frequency pulsed magnetic fields. They induce additional electric currents in tissues and change value of cell membrane potential”.

It is performed. These are references [1–6].

2) Discussion –  In Sentence “Despite, the fact that there have been developed a large number of effective approaches to prevent plants from negative effect of viruses such as, for example, chemotherapy, thermotherapy, and more, a lot of them are characterized by drawback (consider detail on [6]”. It is better to mention those drawbacks because it is very confusing whether we have to look for reference [6] within this article or we have to read the article of this reference.

The section discussion is completely re-written, according to Reviewer’s recommendations. Since, there are similar remarks ny other Reviewers‘ there are discussed new references [85-96]. Please, consider the in-depth revised version of the study and the mentioned in this point section. 

3) In discussion line 20 - the sentence “but also ist pathogens” I am not able to understand the word “ist” Please clear this statement.

All typos are corrected, accordingly.

This research article provides in-details both the qualitative and quantitative description of new metabolites formed under exposure to magnetic pulse treatment on pear plant by using mass spectrometric methods of metabolomics, stochastic dynamics, and chemometrics. However, I think after the revision of the above-mentioned comments it will be better suitable for publication in this journal.

Comments on the Quality of English Language

Dear Authors minor editing of english language required for this research article.

The English is corrected, accordingly.

Reviewer 4 Report

Comments and Suggestions for Authors

Dear Authors,

Your manuscript titled "Effect of magnetically pulse impact on the composition of Pyrus communis L. metabolites¨ might be considered for publication at International Journal of Pyrus communis L. metabolites after major review by the authors. In general, several flaws were detected in it. Introduction section needs to show clearly the background and objectives of your research. Lack of information is observed in Material and Methods. Experimental design is needed. Results section needs to be checked carefully to show relevant data. Discussion section needs to be completed by comparing your work with previous ones in the literature. Conclusions section needs to be rewritten by avoiding to summarize your findings. Suggestions were given to you for addressing all these issues (see below).

Yours sincerely,

Reviewer.

L25-L26 Replace 'metab-olites' by 'meta-bolites'. 

L32-70 Explain clearly what was done by others before in this subject and which are the main objectives of the research conducted herein.

L33 Replace 'magnetic pulse field' by 'magnetic pulse field (MPF)'. 

L34-L35 Replace 'of cells' by 'in cells' and 'ef-fect' by 'e-ffect'.

L36 Replace 'MF' by 'MPF'.

L37 Replace 'MF' by 'MPF'.

L38-L39 Replace 'bi-ochemical' by 'bio-chemical'.

L39 Replace 'mechanisms of biological activity' by 'mechanisms related to biological activity'.

L47 Replace 'MF' by 'MPF'.

L56-L57 Replace 'bar-riers' by 'ba-rriers'.

L58 Replace 'MPT' by 'MPF'.

L61 Replace 'MF' by 'MPF'.

L62 Explain the meaning of LMW.

L66 Replace 'there is used enzyme linked' by 'it will be used enzyme linked to'.

L74-L75 Replace 'metab-olites' by 'meta-bolites'.

Table 1: Order all metabolites according to retention time by section groups.

L84 Replace 'MPT' by 'MPF'.

L85 Replace 'T=3h and T=72 h' by 't=3 h and t=72 h'.

L87 Replace 'as function of duration of period' by 'as function of period'.

L90 Replace 'MPT' by 'MPF'.

L90-L91 Replace 'oc-curring' by 'o-ccurring'.

L93 Replace 'MPT' by 'MPF'.

L96 Replace 'MPT' by 'MPF'.

L97 Replace 'MPT' by 'MPF'.

L99 Replace 'T=72h' by 't=72 h'.

L111 Replace 'T=72-168 h' by 't=72-168 h'.

L118 Replace 'MFT' by 'MPF'.

Table 3: Delete this table. Only comment its content in the text.

L122 Replace 'MFT' by 'MPF'.

L124 Replace 'MFT' by 'MPF'.

L126 Replace 'MFT' by 'MPF'.

L129 Replace 'MFT' by 'MPF'.

L131 Replace 'MFT' by 'MPF'.

L133 Replace 'MFT' by 'MPF'.

L179-L180 Use scientific nomenclature to cite all these species.

L185 Replace 'of analyte' by 'in analyte'.

L201-L202 Replace 'param-eters' by 'para-meters'.

L206 Replace 'of analytes' by 'in analytes'.

L207-L208 Replace 'avail-able' by 'avai-lable'.

L211-L212 Replace 're-spect' by 'res-pect'.

L223 Delete 'It is common sense'.

L233-L234 Replace 'cur-rent' by 'cu-rrent'.

L244-L245 Replace 'elec-tronic' by 'ele-ctronic'.

L280 Replace 'because of' by 'due to'.

L280-L281 Replace 'char-acterized' by 'cha-racterized'.

L281 Replace 'affect' by 'effect'.

L291 Describe the meaning of the acronym LMW.

L293-L294 Delete '(Consider detail on supporting information and relevant references, therein.)'.

L301-L302 Replace 'signif-icantly' by 'signi-ficantly'.

L303 Replace 'Due to,' by 'Due to'.

L303-L304 Replace 'cor-relation' by 'corre-lation'.

L305 Replace 'there is asked assigned molecular, respectively,' by 'they were'.

L306 Replace 'of analytes' by 'in analytes'.

L306-L307 Delete 'to account ... species as'.

L315 Replace 'of naturally' by 'in naturally'.

L316-L317 Replace 'ap-proximated' by 'appro-ximated'.

L317 Replace 'on equation' by 'of equation'.

L320-L321 Replace 'ad-ducts' by 'a-dducts'.

L336-L337 Replace 'ad-ducts' by 'a-dducts'.

L340-L341 Replace 'exam-ining' by 'exami-ning'.

L341-L342 Replace 'prod-ucts' by 'pro-ducts'.

L345-L346 Replace 'energet-ics' by 'energe-tics'.

L363-L364 Replace 'an-ion' by 'a-nion'.

L379-L380 Replace 'ex-amine' by 'exa-mine'.

L389-L390 Replace 'measur-ands' by 'measu-rands'.

L391-L392 Replace 'chal-lenges' by 'cha-llenges'.

L392-L393 Replace 'De-spite' by 'Des-pite'.

L394 Replace 'distinguishing of' by 'to distinguish between',

L397-L398 Replace 'analyt-ical' by 'analy-tical'.

L398 Replace 'of tests' by 'in tests'.

L410-L457 Discuss your results with previous research done by others. Discussion cannot be made by justification of the methodology used herein. It has to be based on evidences. Compare your findings with previous ones.

L414 Replace 'MPT' by 'MPF'.

L417-L418 Replace 'ex-ample' by 'e-xample'.

L425-L426 Replace 'meth-ods' by 'me-thods'.

L426 Replace 'for recovering of infectivity of plants' by a recover of infectivity in plants'.

L429 Replace 'Work' by 'Work done by'.

L430 Replace 'ist' by 'its'.

L431 Replace 'MPT' by 'MPF'.

L433 Replace 'Further:' by 'Furthermore,' 'MPT' by 'MPF'. 

L434 Replace 'depends' by 'depend'.

L438-L443 Delete 'Due to, ... , so far'.

L444 Replace 'MPT' by 'MPF'. 

L445 Replace 'affect on MPT' by 'effect of MPF'.

L447-L448 Replace 'meas-urements' by 'mea-surements'.

L448 Replace 'MPT' by 'MPF'. 

L448-L449 Replace 'en-ergy' by 'e-nergy'.

L449-L453 Delete 'The reported ... magnetic field'.

L453-L454 Replace 'doc-umented' by 'do-cumented'.

L453-L456 Justify this statement by using several cites from the literature.

L463 Explain the meaning of the acronyms 'ACLSV' and 'ASGV'.

L466-L467 Replace 'induc-tion' by 'indu-ction'.

L469 Replace 'MPT' by 'MPF'.

L471 Explain the meaning of the acronym 'BSTFA'.

L476 Replace 'it was' by 'were'.

L478 Replace 'col-umn' by 'co-lumn'.

L487-L488 Replace 'cred-ibility' by 'cre-dibility'.

L489-L497 Complete these sections by adding the corresponding text here.

L502 Replace 'respect the' by 'respect to the'.

L503 Replace 'There are' by 'There were'.

L503 Delete '(32) are' and '34'.

L504 Delete '(5)', '(5)' and '(2).

L505 Delete '(9).

L506-L507 Delete 'There are performed ... spectrometry'.

L499-L507 Rewrite Conclusions section by: 1) Namely the thesis; 2) Summary of main ideas; 3) Cover research gaps; 4) Make recommendations.

L670 Add '4.3.2'.

L685 Replace 'of a dissociating' by 'for a dissociating'.

L690 Replace 'of alcohols' by 'in alcohols'.

L693 Replace 'as reported' by 'were reported'.

L694 Add '4.3.3'.

L698 Add '4.3.1'.

L701 Clarify the formula. It's blurry and it's not understandable what is written.

L707 Replace 'of MS' by 'in mass spectrometry (MS)'.

L717 Clarify the formula. It's blurry and it's not understandable what is written.

L718 Replace 'and is' by 'and it is', 'to any two sets' by 'to any of the two sets' and 'of intensity' by 'related to intensity'.

L719 Replace 'of analyte' by 'in analyte'.

L720 Replace 'In assessing' by 'After assessing' and 'there' by ', it was'.

L722 Replace 'Furthermore' by 'Furthermore,'. 

L725 Clarify the formula. It's blurry and it's not understandable what is written.

L726 Replace 'So far, there are' by 'So far, although' and ', however,' by 'in'.

L728 Replace 'there are needed further' by 'there is a need of further'.

L729 Replace 'Despite, as' by 'However, due to'. Delete 'data' and 'is connected functionally with D¨sd, however'.

L737 Replace 'the fluctuations' by 'fluctuations' and 'because of' by 'due to'.

L738-L799 Add all these cites to references list.

L738 Clarify the formula. It's blurry and it's not understandable what is written.

L817 Replace 'of measurand' by 'for measurand'.

L828 Replace 'of aucubin' by 'in aucubin'.

L849 Replace 'of MS' by 'in MS'.

L852-866 Move Tables A1 and A2 to supplementary files.

Author Response

RESPONSE TO REVIEWER 4

The authors would like to thank for creative Reviewers‘ remarks and recommendations, regarding, the study. They are taken completely into consideration.

Comments and Suggestions for Authors

Dear Authors,

Your manuscript titled "Effect of magnetically pulse impact on the composition of Pyrus communis L. metabolites¨ might be considered for publication at International Journal of Pyrus communis L. metabolites after major review by the authors. In general, several flaws were detected in it. Introduction section needs to show clearly the background and objectives of your research.

AUTHORS: The introductory section is fully re-written. There is added motivation behind the study. We hope, that it illustrates more clearly for Editors, Reviewers, and potential Readers the impact of our study on metabolomics of plants upon external affect on magnetic field treatment in addition to methods of reliable assignment of unknown analytes based on mass pectrometry, thus fitting teh contributions of the study to the scope of the Special Issue of the Journal, because of the content of the paper is designed for Special Issue [https://www.mdpi.com/journal/ijms/special_issues/P4R5D7UIZM].   

Lack of information is observed in Material and Methods.

AUTHORS: Section ‚Materials and methods‘ of revised version of the manuscript, contains three new sub-sections:

4.2. Metabolomics by gas chromatography-mass spectrometry;

4.3. Analytical procedure for separation and identification of plant metabolites; and

4.4. Magnetic pulse analysis of plant biochemistry, respectively. They focuse on sinthesis of silylated metabolites from plant, obtained due to affect of magnetic pulse treatment; experimental design of magnetic measurements, in addition to experimental parameters (please, consider new Table 1;) as well as it deals with comparative analysis of GC-MS and LC-MS methods used to plant metabolomics.  

Experimental design is needed.

AUTHORS: Please, consider new sub-section 4.4 and data on Table 1. They detail on experimental design regarding magnetic pulse measurements. Also, section ‚4. Experimental includes‘ details on GC-MS measurements and synthesis of sililated carbohydrate metabolites extracted from the plant.  

Results section needs to be checked carefully to show relevant data.

AUTHORS: Section ‚Results‘ is reorganized. There is added new sub-section ‚3.4. Mass spectrometric data on silylated carbohydrate monomers’ detailing on relevant to the study mass spectrometric annotation and assignment of unknown silylated carbohydrates both employing routine database searching algorithms and stochastic dynamics mass spectrometric approach based on equations (1), (2), and (6). There are included new figures and tables, illustrating experimental chromatographic and mass spectrometric data of the analytes in addition to chemometrics evaluating reliability of the assignment. Please, see new Figures 5–7 and Table 5 (main text) in addition to data on Figures A8–A16 and Tables A1–A4 (appendix A.)  

Discussion section needs to be completed by comparing your work with previous ones in the literature.

AUTHORS: Section ‚Discussion‘ is completely re-written. There is discussed results from our study comparing analytical data on other authors. Also, there are included new references. Please, look at works [85-96].

Conclusions section needs to be rewritten by avoiding to summarize your findings.

AUTHORS: Section ‚Conclusion‘ is new one. It is designed according to Reviewer’s recommendations.

Suggestions were given to you for addressing all these issues (see below).

Yours sincerely,

Reviewer.

L25-L26 Replace 'metab-olites' by 'meta-bolites'. 

AUTHORS: It is performed.

L32-70 Explain clearly what was done by others before in this subject and which are the main objectives of the research conducted herein.

AUTHORS: New paragraph is added at the end of introductory section, in context of Reviewer’s remarks on the old version of the paper. It details briefly on analyte objects and methodology used to the study. 

L33 Replace 'magnetic pulse field' by 'magnetic pulse field (MPF)'. 

It is performed.

L34-L35 Replace 'of cells' by 'in cells' and 'ef-fect' by 'e-ffect'.

It is performed.

L36 Replace 'MF' by 'MPF'.

It is performed.

L37 Replace 'MF' by 'MPF'.

It is performed.

L38-L39 Replace 'bi-ochemical' by 'bio-chemical'.

It is performed.

L39 Replace 'mechanisms of biological activity' by 'mechanisms related to biological activity'.

It is performed.

L47 Replace 'MF' by 'MPF'.

It is performed.

L56-L57 Replace 'bar-riers' by 'ba-rriers'.

It is performed.

L58 Replace 'MPT' by 'MPF'.

It is performed.

L61 Replace 'MF' by 'MPF'.

It is performed.

L62 Explain the meaning of LMW.

It is done.

L66 Replace 'there is used enzyme linked' by 'it will be used enzyme linked to'.

This part oft he text is re-written.

L74-L75 Replace 'metab-olites' by 'meta-bolites'.

It is performed.

Table 1: Order all metabolites according to retention time by section groups.

Since, there is new Table 1, the old Table 1 becomes new Table 2. The data are presented according to the Reviewer’s remarks, shown above.

L84 Replace 'MPT' by 'MPF'.

It is performed.

L85 Replace 'T=3h and T=72 h' by 't=3 h and t=72 h'.

It is performed.

L87 Replace 'as function of duration of period' by 'as function of period'.

It is done.

L90 Replace 'MPT' by 'MPF'.

It is done.

L90-L91 Replace 'oc-curring' by 'o-ccurring'.

It is done.

L93 Replace 'MPT' by 'MPF'.

It is done.

L96 Replace 'MPT' by 'MPF'.

It is done.

L97 Replace 'MPT' by 'MPF'.

It is done.

L99 Replace 'T=72h' by 't=72 h'.

It is done.

L111 Replace 'T=72-168 h' by 't=72-168 h'.

It is done.

L118 Replace 'MFT' by 'MPF'.

It is done.

Table 3: Delete this table. Only comment its content in the text.

It is performed.

L122 Replace 'MFT' by 'MPF'.

It is performed.

L124 Replace 'MFT' by 'MPF'.

It is performed.

L126 Replace 'MFT' by 'MPF'.

It is performed.

L129 Replace 'MFT' by 'MPF'.

It is performed.

L131 Replace 'MFT' by 'MPF'.

It is performed.

L133 Replace 'MFT' by 'MPF'.

It is performed.

L179-L180 Use scientific nomenclature to cite all these species.

It is performed.

L185 Replace 'of analyte' by 'in analyte'.

It is performed.

L201-L202 Replace 'param-eters' by 'para-meters'.

It is performed.

L206 Replace 'of analytes' by 'in analytes'.

It is performed.

L207-L208 Replace 'avail-able' by 'avai-lable'.

It is performed.

L211-L212 Replace 're-spect' by 'res-pect'.

It is performed.

L223 Delete 'It is common sense'.

This sentense is re-written, completely.

L233-L234 Replace 'cur-rent' by 'cu-rrent'.

It is performed.

L244-L245 Replace 'elec-tronic' by 'ele-ctronic'.

It is performed.

L280 Replace 'because of' by 'due to'.

It is performed.

L280-L281 Replace 'char-acterized' by 'cha-racterized'.

It is performed.

L281 Replace 'affect' by 'effect'.

It is performed.

L291 Describe the meaning of the acronym LMW.

It is performed.

L293-L294 Delete '(Consider detail on supporting information and relevant references, therein.)'.

It is performed.

L301-L302 Replace 'signif-icantly' by 'signi-ficantly'.

It is performed.

L303 Replace 'Due to,' by 'Due to'.

It is carried out.

L303-L304 Replace 'cor-relation' by 'corre-lation'.

It is carried out.

L305 Replace 'there is asked assigned molecular, respectively,' by 'they were'.

The sentence is re-written.

L306 Replace 'of analytes' by 'in analytes'.

It is carried out.

L306-L307 Delete 'to account ... species as'.

The sentence is re-written.

L315 Replace 'of naturally' by 'in naturally'.

It is carried out.

L316-L317 Replace 'ap-proximated' by 'appro-ximated'.

It is carried out.

L317 Replace 'on equation' by 'of equation'.

It is carried out.

L320-L321 Replace 'ad-ducts' by 'a-dducts'.

It is carried out.

L336-L337 Replace 'ad-ducts' by 'a-dducts'.

It is carried out.

L340-L341 Replace 'exam-ining' by 'exami-ning'.

It is carried out.

L341-L342 Replace 'prod-ucts' by 'pro-ducts'.

It is carried out.

L345-L346 Replace 'energet-ics' by 'energe-tics'.

It is carried out.

L363-L364 Replace 'an-ion' by 'a-nion'.

It is carried out.

L379-L380 Replace 'ex-amine' by 'exa-mine'.

It is carried out.

L389-L390 Replace 'measur-ands' by 'measu-rands'.

It is carried out.

L391-L392 Replace 'chal-lenges' by 'cha-llenges'.

It is carried out.

L392-L393 Replace 'De-spite' by 'Des-pite'.

It is carried out.

L394 Replace 'distinguishing of' by 'to distinguish between',

The sentence is re-written.

L397-L398 Replace 'analyt-ical' by 'analy-tical'.

It is carried out.

L398 Replace 'of tests' by 'in tests'.

It is done.

L410-L457 Discuss your results with previous research done by others. Discussion cannot be made by justification of the methodology used herein. It has to be based on evidences. Compare your findings with previous ones.

Section ‚Discussion‘of the revised manuscript compares our analysis with those ones provided by other authors. There are cited and discussed new references [85-96].

L414 Replace 'MPT' by 'MPF'.

It is carried out.

L417-L418 Replace 'ex-ample' by 'e-xample'.

It is carried out.

L425-L426 Replace 'meth-ods' by 'me-thods'.

It is carried out.

L426 Replace 'for recovering of infectivity of plants' by a recover of infectivity in plants'.

This sentence is re-written, fully.

L429 Replace 'Work' by 'Work done by'.

This sentence is re-written, completely.

L430 Replace 'ist' by 'its'.

It is carried out.

L431 Replace 'MPT' by 'MPF'.

It is carried out.

L433 Replace 'Further:' by 'Furthermore,' 'MPT' by 'MPF'. 

It is carried out.

L434 Replace 'depends' by 'depend'.

It is carried out.

L438-L443 Delete 'Due to, ... , so far'.

It is carried out.

L444 Replace 'MPT' by 'MPF'. 

It is done.

L445 Replace 'affect on MPT' by 'effect of MPF'.

It is done.

L447-L448 Replace 'meas-urements' by 'mea-surements'.

It is done.

L448 Replace 'MPT' by 'MPF'. 

It is done.

L448-L449 Replace 'en-ergy' by 'e-nergy'.

It is done.

L449-L453 Delete 'The reported ... magnetic field'.

The sentence is fully re-written.

L453-L454 Replace 'doc-umented' by 'do-cumented'.

It is done.

L453-L456 Justify this statement by using several cites from the literature.

The sentence is re-written.

L463 Explain the meaning of the acronyms 'ACLSV' and 'ASGV'.

It is done.

L466-L467 Replace 'induc-tion' by 'indu-ction'.

It is done.

L469 Replace 'MPT' by 'MPF'.

It is done.

L471 Explain the meaning of the acronym 'BSTFA'.

It is done.

L476 Replace 'it was' by 'were'.

The sentence is re-written, fully.

L478 Replace 'col-umn' by 'co-lumn'.

It is done.

L487-L488 Replace 'cred-ibility' by 'cre-dibility'.

It is done.

L489-L497 Complete these sections by adding the corresponding text here.

Sub-section 3.4 Theory/computations is fully revised.

L502 Replace 'respect the' by 'respect to the'.

It is done.

L503 Replace 'There are' by 'There were'.

It is done.

L503 Delete '(32) are' and '34'.

It is carried out.

L504 Delete '(5)', '(5)' and '(2).

It is carried out.

L505 Delete '(9).

It is carried out.

L506-L507 Delete 'There are performed ... spectrometry'.

The sentence is re-written.

L499-L507 Rewrite Conclusions section by: 1) Namely the thesis; 2) Summary of main ideas; 3) Cover research gaps; 4) Make recommendations.

The conclusion section is re-written.

L670 Add '4.3.2'.

It is carried out in the plain text, according to the recommendations by the Reviewer.

L685 Replace 'of a dissociating' by 'for a dissociating'.

It is done.

L690 Replace 'of alcohols' by 'in alcohols'.

It is done.

L693 Replace 'as reported' by 'were reported'.

It is done.

L694 Add '4.3.3'.

It is done.

L698 Add '4.3.1'.

It is done.

L701 Clarify the formula. It's blurry and it's not understandable what is written.

There is discussed briefly in new sub-section these equations. Please, consider sub-section 4.5.1. Stochastic dynamics mass spectrometric approach.

L707 Replace 'of MS' by 'in mass spectrometry (MS)'.

It is done.

L717 Clarify the formula. It's blurry and it's not understandable what is written.

There is added new brief sub-section, clarifying the equatiuons. Please, look at sub-section 4.5.1. Stochastic dynamics mass spectrometric approach.

L718 Replace 'and is' by 'and it is', 'to any two sets' by 'to any of the two sets' and 'of intensity' by 'related to intensity'.

It is done.

L719 Replace 'of analyte' by 'in analyte'.

It is done.

L720 Replace 'In assessing' by 'After assessing' and 'there' by ', it was'.

It is done.

L722 Replace 'Furthermore' by 'Furthermore,'. 

It is done.

L725 Clarify the formula. It's blurry and it's not understandable what is written.

There is added new brief sub-section, clarifying all formulas used to this study. Please, consider sub-section 4.5.1. Stochastic dynamics mass spectrometric approach.

L726 Replace 'So far, there are' by 'So far, although' and ', however,' by 'in'.

It is done.

L728 Replace 'there are needed further' by 'there is a need of further'.

It is done.

L729 Replace 'Despite, as' by 'However, due to'. Delete 'data' and 'is connected functionally with D¨sd, however'.

It is done.

L737 Replace 'the fluctuations' by 'fluctuations' and 'because of' by 'due to'.

It is done.

L738-L799 Add all these cites to references list.

It is carried out.

L738 Clarify the formula. It's blurry and it's not understandable what is written.

The equations are written by means of text editor. Due to conversion from doc to pdf format of the paper most probably of them are unreadable for the Reviewer. The revised version of the manuscript contains the same equations shown, however, as figure embedded. In addition, there is added new brief sub-section detailing on the formulas used to this study. Please, consider sub-section 4.5.1. Stochastic dynamics mass spectrometric approach.

L817 Replace 'of measurand' by 'for measurand'.

It is carried out.

L828 Replace 'of aucubin' by 'in aucubin'.

It is carried out.

L849 Replace 'of MS' by 'in MS'.

It is performed.

L852-866 Move Tables A1 and A2 to supplementary files.

It is done.

Round 2

Reviewer 1 Report

Comments and Suggestions for Authors 1)  The intention of the Burn's reference is to "avoid confusion", but the confusion here is that line 62 uses "recovery" in the context of a sick pear plant returning to healthy status after magnetic pulse treatment.    "Recovery" is next seen on line 150 where the ratio of the chromatograms for CA in infected plants after 72 hr after no treatment or 10 Hz  is given as 4.8. This is not a measure of how much recovery to healthy has been achieved by a sick plant after treatment since no healthy plant has been measured on. The issue is now behind us, the manuscript is not about sick plants becoming healthy after magnetic pulse treatment.   2)  The title should be changed to what the manuscript is about, such as "Mass spectrometric identification of metabolites after magnetic pulse treatment of infected Pyrus communis L. microplants"    The special issue is "devoted to recent achievements and the application of mass spectrometric methods to the field of biochemistry." The magnetic pulse treatment is only a means to cause changes in metabolite composition and thus has a secondary role in the manuscript.   3)  Section 3 should be about what new is being presented about the capability of mass spectrometry to identify metabolites, which raises the following questions:   --The conclusion of the extremely long first paragraph on line 548 is that it is important to be able to determine the concentration of chlorogenic acid. This seems to be a routine analyte, so the paragraph doesn't seem to contribute to the subject of the manuscript. --Next up on 551 is monoterpene ancubin, but it is only mentioned for antiviral activity with no details about what is being presented new for its MS analysis. Then the text returns to praise of magnetic pulse treatment, the secondary subject of the manuscript, on line 552. --Aucubin returns on line 606 but no comparisons with other studies such as reference 78 is included. How the information on lines 598-613 specifically contributed to experimental results, such as on line 190, should be added to the discussion. In other words, how is the work reported advancing the field past just monitoring selected m/z by identifying new metobilites.    Author instructions describe a conclusion section as "This section is not mandatory but can be added to the manuscript if the discussion is unusually long or complex." The new conclusion is what's unusually long and complex. It should be reduced to a maximum of 15 lines containing the most important conclusions, not detailed results, from the study.   4) Author instructions for materials and methods are "They should be described with sufficient detail to allow others to replicate and build on published results." Section 4.3 should concentrate on describing what analytical methods were used, and not discuss experimental procedure as if it were some sort of issue.   5)  Section 2.1 contains examples of paragraphs with appropriate length having unified subject content. Section 2.2 is inappropriate, one paragraph for the whole section. All paragraphs should be broken up according to the 2.1 model. Anything over 15 lines should be viewed with suspicion and evaluated break up.   There are too many references, as stated in point 2 of the response it's not a review article which is being presented. Pick out the best ones that support particular issues without blanketing the area with them. It doesn't require, for example, 14 references to support fluctuations in measurands as on line 215. Cite the most relevant and delete the rest.     Comments on the Quality of English Language

They claim extensive re-writing but things didn't get better. I'd be satisfied if they just get rid of the long paragraphs.

Author Response

RESPONSE TO REVIEWER 1

Comments and Suggestions for Authors

REVIEWER 1: 1)  The intention of the Burn's reference is to "avoid confusion", but the confusion here is that line 62 uses "recovery" in the context of a sick pear plant returning to healthy status after magnetic pulse treatment.    "Recovery" is next seen on line 150 where the ratio of the chromatograms for CA in infected plants after 72 hr after no treatment or 10 Hz  is given as 4.8. This is not a measure of how much recovery to healthy has been achieved by a sick plant after treatment since no healthy plant has been measured on. The issue is now behind us, the manuscript is not about sick plants becoming healthy after magnetic pulse treatment.

AUTHORS: The authors would like to thank for the useful Reviewer’s comments on term ‚Recovery‘ from perspective of virology. We have corrected the value reflecting in mind ‚Recovery‘ looking at standard procedures for validation used to analytical chemistry and treating results from measurands or variables of properties of chemical compounds obtained on the base on analytical instrumentation response.   

 REVIEWER 1: 2) The title should be changed to what the manuscript is about, such as "Mass spectrometric identification of metabolites after magnetic pulse treatment of infected Pyrus communis L. microplants"    The special issue is "devoted to recent achievements and the application of mass spectrometric methods to the field of biochemistry." The magnetic pulse treatment is only a means to cause changes in metabolite composition and thus has a secondary role in the manuscript.  

AUTHORS: The title of the study is correcting using the Reviewer’s one.

REVIEWER 1: 3)  Section 3 should be about what new is being presented about the capability of mass spectrometry to identify metabolites, which raises the following questions:   --The conclusion of the extremely long first paragraph on line 548 is that it is important to be able to determine the concentration of chlorogenic acid. This seems to be a routine analyte, so the paragraph doesn't seem to contribute to the subject of the manuscript. --Next up on 551 is monoterpene ancubin, but it is only mentioned for antiviral activity with no details about what is being presented new for its MS analysis. Then the text returns to praise of magnetic pulse treatment, the secondary subject of the manuscript, on line 552. --Aucubin returns on line 606 but no comparisons with other studies such as reference 78 is included. How the information on lines 598-613 specifically contributed to experimental results, such as on line 190, should be added to the discussion. In other words, how is the work reported advancing the field past just monitoring selected m/z by identifying new metobilites.    Author instructions describe a conclusion section as "This section is not mandatory but can be added to the manuscript if the discussion is unusually long or complex." The new conclusion is what's unusually long and complex. It should be reduced to a maximum of 15 lines containing the most important conclusions, not detailed results, from the study.

Section ‚Conclusion‘ is removed owing to Reviewer’s comments on the text. On the one hand it is not mandatory, as the Reviewer has highlighted. On the other hand, section ‚Discussion‘ as length is relatively short text. Moreover, there are data from interdisciplinary research. In such cases there is unable to restrict ourselves toward what is most important from all new data on these different disciplines, which broadly depends on audience.   

REVIEWER 1: 4) Author instructions for materials and methods are "They should be described with sufficient detail to allow others to replicate and build on published results." Section 4.3 should concentrate on describing what analytical methods were used, and not discuss experimental procedure as if it were some sort of issue.  

AUTHORS: Section 4.3 is corrected according to Reviewer comments on. The authors would like to mention that it was designed on the base on Editors‘ remarks on the fact that there has not been specified, why is used GC-MS. The authors hope that newly, revised, version of the manuscript underlines, on the one hand, the fact that GC-MS is routinely used to substituted carbohydrates. On the otehr hand, it shows, why mass spectrometry is method of choice for plant metabolomics and research on plant biochemistry.   

5)  Section 2.1 contains examples of paragraphs with appropriate length having unified subject content. Section 2.2 is inappropriate, one paragraph for the whole section. All paragraphs should be broken up according to the 2.1 model. Anything over 15 lines should be viewed with suspicion and evaluated break up.   There are too many references, as stated in point 2 of the response it's not a review article which is being presented. Pick out the best ones that support particular issues without blanketing the area with them. It doesn't require, for example, 14 references to support fluctuations in measurands as on line 215. Cite the most relevant and delete the rest.    

AUTHORS: English is revised. There are short sentences, excluding from large complex phrases. The corrections are highlighted in red.

Comments on the Quality of English Language

They claim extensive re-writing but things didn't get better. I'd be satisfied if they just get rid of the long paragraphs.

AUTHORS: The authors hope that the current version of the text is more clearly written for Editors, Reviewers, and potential Readers.

Reviewer 2 Report

Comments and Suggestions for Authors

In the new edition of the manuscript, the authors took into account some of the reviewer’s comments. The authors significantly changed the text of the manuscript. In result the length of the text has almost doubled (18 to 34 pages, the number of references from 54 to 137). Whether such an increase in the text of the article is acceptable must be decided by the editors.

The new version of the manuscript requires a number of mandatory corrections:

1. All formulas and notations in the text must be typed in Word->Inset->Formula (or “Alt” plus “=” keys). In the manuscript all formulas are inserted into the text in the form of images. In this case, the indexes are not read.

2. Figures must meet the requirements of the MDPI journals for figures (>300 dpi, >1100 dpi in width).

3. APPENDIX A and APPENDIX B must be transferred to Supplementary Materials. This is a separate file that is not part of the text of the article and is mentioned in the article as a link to a resource on the Internet.

4. In the new version of this paper, Supplementary contains many Origin files. You should export the graphics from Origin, sign the resulting drawings and thus form materials for APPENDIX C in Supplementary Materials. There should also be accompanying text explaining these drawings.

5. Authors cite their works 20 times (14.6% of the total number of 137 references). This is inappropriate self-citations by authors.

6. In the new version, “Supplementary” contains many Origin files. Authors should export the graphics from these Origin files, sign the resulting drawings and thus obtain the APPENDIX C materials in Supplementary Materials. There should also be accompanying text explaining these drawings.

In summary, I would like to say that the article as a whole is suitable for publication. But the authors deviate very far from the IJMS “rules for article authors”. The editors of IJMS will have to solve this problem.

Author Response

RESPONSE TO REVIEWER 2

Comments and Suggestions for Authors

In the new edition of the manuscript, the authors took into account some of the reviewer’s comments. The authors significantly changed the text of the manuscript. In result the length of the text has almost doubled (18 to 34 pages, the number of references from 54 to 137). Whether such an increase in the text of the article is acceptable must be decided by the editors.

AUTHORS: Due to Reviewers’ recommendations largest part of supporting information section together with reference section, therein, from the first version of the paper was moved to plain text. This causes for increasing in length of the manuscript, however, it is not new for the Reviewers. Despite, as can be seen, due to new creative recommendations part of this text and references are excluded from the current new revised version of the paper. In fact, only section 4.3., 2.4., and 4.4. are new ones, because of there has been requested experimental chromatographic and mass spectrometric data, in addition to detail on advantages and limitations of GC-MS and LC-MS approaches as well as detail on the magnetic pulse experiments together with experimental parameters. Despite, these new sub-sections are corrected in the current version of the paper, accordingly. The authors hope that the current length of the manuscript is appropriated. It is reduced to 30 pages. 

The new version of the manuscript requires a number of mandatory corrections:

  1. All formulas and notations in the text must be typed in Word->Inset->Formula (or “Alt” plus “=” keys). In the manuscript all formulas are inserted into the text in the form of images. In this case, the indexes are not read.

AUTHORS: It is performed. The *.doc version of the newly revised manuscript contains equations written in Equation Editor 3 for Word 2019 according to the Reviewer’s recommendations. 

  1. Figures must meet the requirements of the MDPI journals for figures (>300 dpi, >1100 dpi in width).

AUTHORS: It is performed.

  1. APPENDIX A and APPENDIX B must be transferred to Supplementary Materials. This is a separate file that is not part of the text of the article and is mentioned in the article as a link to a resource on the Internet.

AUTHORS: It is performed.

  1. In the new version of this paper, Supplementary contains many Origin files. You should export the graphics from Origin, sign the resulting drawings and thus form materials for APPENDIX C in Supplementary Materials. There should also be accompanying text explaining these drawings.

AUTHORS: It is performed. Please, consider new section Appendix C.

  1. Authors cite their works 20 times (14.6% of the total number of 137 references). This is inappropriate self-citations by authors.

The self-citation is restricted up to four references. Please, consider the revised version of section ‘References’ of the paper.

  1. In the new version, “Supplementary” contains many Origin files. Authors should export the graphics from these Origin files, sign the resulting drawings and thus obtain the APPENDIX C materials in Supplementary Materials. There should also be accompanying text explaining these drawings.

AUTHORS: It is performed. Please, consider new section Appendix C.

In summary, I would like to say that the article as a whole is suitable for publication. But the authors deviate very far from the IJMS “rules for article authors”. The editors of IJMS will have to solve this problem.

AUTHORS: The authors hope that the current newly revised version of the paper balances between past and current contributions to all co-authors, despite, interdisciplinarity of the study. Thematically, the Special Issue itself is interdisciplinary one, balancing between pure disciplines of biochemistry and application of ‘’mass spectrum’’ to solve problems of identification and annotation of plant metabolites and biochemical processes in vitro and in vivo. However, new subsection 4.3 shows mass spectrometry is method of choice for a set of interdisciplinary research fields. Given that, contributions to develop methodologically mass spectrometry have impact on many research fields. The same can be said for methodological developments of biochemistry approaches themselves. In addition, the authors hope that with the signed figures (appendix C) expressing theoretical quantum chemical mass spectrometric part of the study and chemometrics, the research tasks of the three co-authors are clearly highlighted for Editors, Reviewers, and potential readers. Moreover, new sub-section 4.4 underlines that there is applied, in fact, patented method for magnetic pulse treatment of plants developed from Research Institution of Russian Federation. It is shown in section ‘Discussion’, as well.        

Reviewer 3 Report

Comments and Suggestions for Authors

All the corrections have been made successfully. The article has been accepted in its present form.

Author Response

RESPONSE TO REVIEWER 3

Comments and Suggestions for Authors

All the corrections have been made successfully. The article has been accepted in its present form.

AUTHORS: The authors would like to thank for the Reviewer for his, respectively, her spend time in evaluating the results from the study and their presentation.

Reviewer 4 Report

Comments and Suggestions for Authors

Dear Authors,

Your manuscript is now suitable for publication in present form.

Yours sincerely,

Reviewer.

Author Response

RESPONSE TO REVIEWER 4

Comments and Suggestions for Authors

Dear Authors,

Your manuscript is now suitable for publication in present form.

Yours sincerely,

Reviewer.

AUTHORS: The authors would like to thank for the Reviewer for his, respectively, her spend time in evaluating the results from the study and their presentation.

Round 3

Reviewer 1 Report

Comments and Suggestions for Authors

1)  "Recovery" is mentioned on lines 61, 65, 146 and 865. It is now clear that the first two refer to becoming free of a disease and the second two to analytical chemistry.  

5)  The first paragraph in the introduction starts on line 34 and ends on line 107. That makes it longer than the suspicion check point of 15 lines and, not surprisingly, it's the only paragraph in the introduction.

It would make the introduction look a lot more appealing if the paragraph were broken down at lines 46 ("on" in what will be the first sentence in the second paragraph should be deleted), 56, 64, 78, and 94.

The only paragraph in section 2.4 starts on line 256 and ends on line 344. It should be broken up on lines 268, 287, 309, 316, and 333.

Comments on the Quality of English Language

Comments are mandatory, but I don't know what to say. Pauses at the new paragraphs should help readers.

Author Response

Comments and Suggestions for Authors

The authors would like to thank for the valuable and very useful Reviewer’s comments on the revised text of the paper. They are completely taken into account.

1)  "Recovery" is mentioned on lines 61, 65, 146 and 865. It is now clear that the first two refer to becoming free of a disease and the second two to analytical chemistry.  

AUTHORS: The distinction between the two terms is highlighted in newly, revised, text of the paper, showing recovery of free of disease plants (line 61 and 65) and recovery factor in analytical method (line 146.) Thus, the authors hope that the text avoids potential misunderstandings.

5)  The first paragraph in the introduction starts on line 34 and ends on line 107. That makes it longer than the suspicion check point of 15 lines and, not surprisingly, it's the only paragraph in the introduction.

AUTHORS: The introductory paragraph is splitted into a set of small-length paragraphs.

It would make the introduction look a lot more appealing if the paragraph were broken down at lines 46 ("on" in what will be the first sentence in the second paragraph should be deleted), 56, 64, 78, and 94.

AUTHORS: It is performad. In addition, the sentences of the shown lines are re-written. The authors hope is that their current versions are more clear for Editors, Reviewers, and potential Readers.

The only paragraph in section 2.4 starts on line 256 and ends on line 344. It should be broken up on lines 268, 287, 309, 316, and 333.

AUTHORS: It is performed. In addition, some sentenses are re-phrased, thus, the authors hope is that they become more clearly written.  

Comments on the Quality of English Language

Comments are mandatory, but I don't know what to say. Pauses at the new paragraphs should help readers.
